# Synthesis and HPLC-ECD Study of Cytostatic Condensed *O,N*-Heterocycles Obtained from 3-Aminoflavanones

**DOI:** 10.3390/biom10101462

**Published:** 2020-10-20

**Authors:** Ádám Szappanos, Attila Mándi, Katalin Gulácsi, Erika Lisztes, Balázs István Tóth, Tamás Bíró, Anita Kónya-Ábrahám, Attila Kiss-Szikszai, Attila Bényei, Sándor Antus, Tibor Kurtán

**Affiliations:** 1Department of Organic Chemistry, University of Debrecen, P. O. Box 400, 4002 Debrecen, Hungary; szappanos.adam@gmail.com (Á.S.); mandi.attila@science.unideb.hu (A.M.); gulacsi.katalin@science.unideb.hu (K.G.); dulryc@unideb.hu (A.K.-Á.); kiss.attila@science.unideb.hu (A.K.-S.); antus.sandor@science.unideb.hu (S.A.); 2Doctoral School of Chemistry, University of Debrecen, Egyetem tér 1, 4032 Debrecen, Hungary; 3Department of Physiology, University of Debrecen, 4012 Debrecen, Hungary; lisztes.erika@med.unideb.hu (E.L.); toth.istvan@med.unideb.hu (B.I.T.); 4Department of Immunology, University of Debrecen, 4032 Debrecen, Hungary; biro.tamas@med.unideb.hu; 5Department of Physical Chemistry, University of Debrecen, 4032 Debrecen, Hungary; benyei.attila@science.unideb.hu

**Keywords:** neber rearrangement, 3-aminoflavanones, antiproliferative activity, TDDFT-ECD calculations, HPLC-ECD, 3-(*N*-chloroacetylamino)-flavan-4-ol, thiazole-condensed 2*H*-chromene, pyrrole-condensed 2*H*-chromene, lamellarin analogues

## Abstract

Racemic chiral *O,N*-heterocycles containing 2-arylchroman or 2-aryl-2*H*-chromene subunit condensed with morpholine, thiazole, or pyrrole moieties at the C-3-C-4 bond were synthesized with various substitution patterns of the aryl group by the cyclization of *cis*- or *trans*-3-aminoflavanone analogues. The 3-aminoflavanone precursors were obtained in a Neber rearrangement of oxime tosylates of flavanones, which provided the *trans* diastereomer as the major product and enabled the isolation of both the *cis*- and *trans*-diastereomers. The *cis*- and *trans*-aminoflavanones were utilized to prepare three diastereomers of 5-aryl-chromeno[4,3-b][1,4]oxazines. Antiproliferative activity of the condensed heterocycles and precursors was evaluated against A2780 and WM35 cancer cell lines. For a 3-(*N*-chloroacetylamino)-flavan-4-ol derivative, showing structural analogy with acyclic acid ceramidase inhibitors, 0.15 μM, 3.50 μM, and 6.06 μM IC_50_ values were measured against A2780, WM35, and HaCat cell lines, and apoptotic mechanism was confirmed. Low micromolar IC_50_ values down to 2.14 μM were identified for the thiazole- and pyrrole-condensed 2*H*-chromene derivatives. Enantiomers of the condensed heterocycles were separated by HPLC using chiral stationary phase, HPLC-ECD spectra were recorded and TDDFT-ECD calculations were performed to determine the absolute configuration and solution conformation. Characteristic ECD transitions of the separated enantiomers were correlated with the absolute configuration and effect of substitution pattern on the HPLC elution order was determined.

## 1. Introduction

The 3-aminoflavanone scaffold **1** is considered an efficient building block for the preparation of condensed chiral *O,N*-heterocycles **2**-**4** (Scheme 1), which contain a 2-arylchroman or 2-aryl-2*H*-chromene moiety fused at the C-3−C-4 bond with azine- or azole-type heterocycles. The 3-aminoflavanone derivatives can be obtained by the Neber rearrangement [1] of the oxime tosylate derivative **5**, readily available from flavanones **6** in two steps. The Neber rearrangement involves the conversion of the oxime tosylate of a ketone to a reactive 2*H*-azirine intermediate in the presence of an alkoxide base, the ring-opening of which produces an α-aminoketone [2,3]. Although oxime tosylate of flavanone *rac*-**E** were converted to 3-aminoflavanone in a Neber reaction as early as 1959 [4] and isolation of *trans*-3-aminoflavanone *rac*-**C** were reported [5,6], the synthetic potential of 3-aminoflavanones for the preparation of condensed *O*,*N*-heterocycles has been underutilized (Scheme 1). Only the preparation of oxazoline- and imidazole-condensed derivatives *rac*-**A** and *rac*-**B** was reported with a few examples. Moreover, in this work the diastereoselectivity and side-products of the Neber reaction in the presence of an inherent C-2 chirality center of flavanones were studied further by modifying the reaction conditions and isolation of the diastereomeric products. Asymmetric organocatalytic Neber reactions of oxime tosylates producing optically active 2*H*-azirine derivatives have been recently reported, in which the reaction conditions are adjusted to stop the transformation at the stage of the 2*H*-azirine intermediates [7,8,9].

The condensed *O,N*-heterocyclic target molecules **2**-**4** of the recent work contain a morpholine, thiazole, or pyrrole unit fused at the C-3−C-4 bond of the 2-arylchroman or 2*H*-chromene skeleton and each of them are represented by 7 analogues differing in the C-2 aryl substituents. A literature survey showed that analogous *O,N*-heterocycles with condensed morpholine, pyrrole, and thiazole subunits received great attention because of their remarkable pharmacological activities such as tau protein kinase 1 (TPK1) inhibitory activity of **7** [10], dopamine D_3_ receptor agonist activity of **8** [11], ion channel modulatory activity of **9** [12], interleukin-2 (IL-2) inhibitory activity of **10** [13], topoisomerase I inhibitory activity of **11** [14], and antibacterial activity of **12** [15] (Figure 1). The synthetic derivative **11** is a simplified analogue of natural lemallarins, cytotoxic marine natural products with potent topoisomerase I inhibitory activity isolated from molluscs, ascidians, and sponges [16,17].

In this work, the synthesis of the target heterocycles **2**-**4** is carried out through the 3-aminoflavanone derivatives and their antiproliferative activity was tested on WM35 melanoma and A2780 ovarian human cancer cell lines by MTT assay. Low micromolar IC_50_ values could be measured for several chiral racemic *O,N*-heterocyclic derivatives, which prompted us to separate the enantiomers with chiral HPLC, record the online HPLC-ECD spectra and determine the absolute configuration by TDDFT-ECD calculations.

## 2. Materials and Methods

### 2.1. Chemicals

Melting points were determined on a Kofler hot-stage apparatus and are uncorrected. The NMR spectra were recorded on Bruker Aspect 3000 (^1^H: 360 MHz, ^13^C: 90 MHz) and Bruker Avance II 400 (^1^H: 400 MHz; ^13^C: 100 MHz) spectrometers using TMS as internal standard. Chemical shifts were reported as ppm and ^3^*J*_H,H_ coupling constants in Hz. Chiral HPLC separation of *rac*-**20a-g**, *rac*-**23a-g**, *rac*-**3a-g,** and *rac*-**4a-g** were performed on a JASCO HPLC system with Chiralpak-IA column (5 μm, 150 × 4.6 mm, hexane/2-propanol 80:20, 90:10 eluent, respectively, 1 mL min^−1^ flow rate) or Chiralpak-IC column (5 μm, 250 × 4.6 mm, hexane/2-propanol 70:30 eluent, respectively, 1 mL min^−1^ flow rate) and HPLC-ECD spectra were recorded in stopped-flow mode on a JASCO *J*-810 electronic circular dichroism spectropolarimeter equipped with a 10 mm HPLC flow cell. ECD ellipticity (ϕ) values were not corrected for concentration. For an HPLC-ECD spectrum, three consecutive scans were recorded and averaged with 2 nm bandwidth, 1 s response, and standard sensitivity. The HPLC-ECD spectrum of the eluent recorded in the same way was used as background. The concentration of the injected sample was set so that the HT value did not exceed 500 V in the HT channel down to 230 nm. IR spectra were recorded on a JASCO FT/IR-4100 spectrometer and absorption bands are presented as wavenumber in cm^−1^. Electrospay quadrupole time-of-flight HRMS measurements were performed with a MicroTOF-Q type QqTOF MS or maXis II UHR ESI- QTOF MS instrument equipped with an ESI source from Bruker (Bruker Daltoniks, Bremen, Germany).

### 2.2. General Procedure for the Synthesis of Tosyl Oxime Analogues (***5a***-***g***)

Oxime derivative **16a-g** (12.54 mmol) and Et_3_N (2.11 mL, 15.04 mmol) were dissolved in anhydrous CH_2_Cl_2_ (50 mL) under inert atmosphere. At room temperature, *p*-toluenesulfonyl chloride (15.04 mmol) was added to the solution. The mixture was refluxed for 3 h. Extraction with water, drying over MgSO_4_ and concentration under reduced pressure afforded the crude product as orange oil. The oil was triturated with cold hexane, which resulted in the pure product.

*N-{[(4-methylphenyl)sulfonyl]oxy}-2-phenyl-2,3-dihydro-4H-chromen-4-imine* (**5a**). White crystals, yield 87%, mp 142–143 °C; ^1^H-NMR (400 MHz, CDCl_3_) δ: 2.43 (s, 3H, CH_3_), 2.75 (dd, *J* = 17.2, 12.4 Hz, 1H, 3-H_a_), 3.44 (dd, *J* = 17.2, 2.8 Hz, 1H, 3-H_b_), 5.01 (dd, *J* = 12.4, 2.8 Hz, 1H, 2-H), 6.94 (m, 2H, 6-H, 8-H), 7.34 (m, 8H, 7-H, 2′-H, 3′-H, 4′-H, 5′-H, 6′-H, 3″-H, 5″-H) 7.81 (dd, *J* = 8.0, 1.6 Hz, 1H, 5-H), 7.92 (d, 2H, 2″-H, 6″-H); ^13^C-NMR (100 MHz, CDCl_3_): δ: 21.8 (C-CH_3_), 31.9 (C-3), 76.8 (C-2), 115.6 (C-4a), 118.3 (C-8), 121.8 (C-6), 125.1 (C-5), 126.2 (C-2′, C-6′), 128.9 (C-3′, C-4′, C-5′), 129.1 (C-3″, C-5″), 129.7 (C-2″, C-6″), 132.6 (C-1″), 133.5 (C-7), 138.7 (C-1′), 145.3 (C-4″), 157.2 (C-4), 157.9 (C-8a); HRMS (ESI) calcd. for C_22_H_19_NaNO_4_S [M+Na]^+^ 416.093; found 416.093.

*N-{[(4-methylphenyl)sulfonyl]oxy}-2-(4-methoxyphenyl)-2,3-dihydro-4H-chromen-4-imine* (**5b**). Off-white crystals, yield 93%, mp 165–167 °C; ^1^H-NMR (400 MHz, CDCl_3_) δ: 2.44 (s, 3H, CH_3_), 2.78 (dd, *J* = 17.6, 12.8 Hz, 1H, 3-H_a_), 3.42 (dd, *J* = 17.6, 2.8 Hz, 1H, 3-H_b_), 3.81 (s, 3H, OCH_3_), 4.97 (dd, *J* = 12.8, 2.8 Hz, 1H, 2-H), 6.91 (m, 4 H, 6-H, 8-H, 3′-H, 5′-H), 7.32 (m, 5H, 7-H, 2′-H, 6′-H, 3″-H, 5″-H), 7.81 (dd, *J* = 8.0, 1.2 Hz, 1H, 5-H), 7.93 (d, 2H, 2″-H, 6″-H); ^13^C-NMR (100 MHz, CDCl_3_) δ: 21.8 (C-CH_3_), 31.6 (C-3), 55.4 (C-OCH_3_), 76.5 (C-2), 114.2 (C-3′, C-5′) 115.5 (C-4a), 118.3 (C-8), 121.7 (C-6), 125.1 (C-5), 127.7 (C-2′, C-6′), 129.1 (C-3″, C-5″), 129.7 (C-2″, C-6″), 130.7 (C-1′), 132.6 (C-1″), 133.5 (C-7), 145.3 (C-4″), 157.5 (C-4), 158.0 (C-8a), 160.0 (C-4′); HRMS (ESI) calcd. for C_23_H_21_NaNO_5_S [M+Na]^+^ 446.104; found 446.105.

*N-{[(4-methylphenyl)sulfonyl]oxy}-2-(3,4-dimethoxyphenyl)-2,3-dihydro-4H-chromen-4-imine* (**5c**). Off-white crystals, yield 92%, mp 148–150 °C; ^1^H-NMR (400 MHz, CDCl_3_) δ: 2.44 (s, 3H, CH_3_), 2.82 (dd, *J* = 17.6, 12.4 Hz, 1H, 3-H_a_), 3.43 (d, *J* = 17.6 Hz, 1H, 3-H_b_), 3.89 (d, 6H, 2 × OCH_3_), 4.99 (d, *J* = 12.4 Hz, 1H, 2-H), 6.87 (d, *J* = 7.6 Hz, 1H, 8-H), 6.94 (m, 4 H, 6-H, 2′-H, 5′-H, 6′-H), 7.35 (m, 3H, 7-H, 3″-H, 5″-H), 7.80 (d, *J* = 8.4 Hz, 1H, 5-H), 7.93 (d, 2H, 2″-H, 6″-H); ^13^C-NMR (100 MHz, CDCl_3_): δ = 21.7 (C-CH_3_), 31.7 (C-3), 56.0 (2 × C-OCH_3_), 76.7 (C-2), 109.4 (C-5′), 111.2 (C-2′), 115.5 (C-4a), 118.3 (C-8), 118.9 (C-6′), 121.8 (C-6), 125.0 (C-5), 129.0 (C-3″, C-5″), 129.7 (C-2″, C-6″), 131.1 (C-1″), 132.5 (C-1′), 133.4 (C-7), 145.3 (C-4″), 149.3 (C-4′), 149.5 (C-3′), 157.3 (C-4), 157.8 (C-8a); HRMS (ESI) calcd. for C_24_H_23_NaNO_6_S [M+Na]^+^ 476.114; found 476.113.

*N-{[(4-methylphenyl)sulfonyl]oxy}-2-(3,5-dimethoxyphenyl)-2,3-dihydro-4H-chromen-4-imine* (**5d**). Off-white crystals, yield: 89%, mp 142–144 °C; ^1^H-NMR (400 MHz, CDCl_3_) δ: 2.44 (s, 3H, CH_3_), 2.76 (d, *J* = 17.6, 12.8 Hz, 1H, 3-H_a_), 3.44 (dd, *J* = 17.6, 3.2 Hz, 1H, 3-H_b_), 3.8 (s, 6H, 2xOCH_3_), 4.96 (dd, *J* = 12.8, 3.2 Hz, 1H, 2-H), 6.44 (t, *J* = 2.4 Hz, 1H, 4′-H), 6.55 (d, 2H, 2′-H, 6′-H), 6.94 (m, 2H, 6-H, 8-H), 7.32 (7-H, 3″-H, 5″-H), 7.80 (dd, *J* = 8.4, 1.6 Hz, 1H, 5-H), 7.92 (d, 2H, 2″-H, 6″-H); ^13^C-NMR (100 MHz, CDCl_3_) δ: 21.8 (C-CH_3_), 32.0 (C-3), 55.5 (2xC-OCH_3_), 76.8 (C-2), 100.6 (C-4′), 104.2 (C-2′, C-6′), 115.6 (C-4a), 118.4 (C-8), 121.9 (C-6), 125.1 (C-5), 129.1 (C-3″, C-5″), 129.7 (C-2″, C-6″), 132.6 (C-1″), 133.5 (C-7), 141.1 (C-1′), 145.3 (C-4″), 157.2 (C-4), 157.7 (C-8a), 161.2 (C-3′, C-5′); HRMS (ESI) calcd. for C_24_H_23_NO_6_S [M + H]^+^ 454.132; found 454.131.

*N-{[(4-methylphenyl)sulfonyl]oxy}-2-(3,4,5-trimethoxyphenyl)-2,3-dihydro-4H-chromen-4-imine* (**5e**). Off-white crystals, yield: 84%, mp 157–159 °C; ^1^H-NMR (400 MHz, CDCl_3_) δ: 2.45 (s, 3H, CH_3_), 2.81 (dd, *J* = 17.6, 12.4 Hz, 1H, 3-H_a_), 3.45 (dd, *J* = 17.6, 3.2 Hz, 1H, 3-H_b_), 3.85 (m, 9H, 3 × OCH_3_), 4.98 (dd, *J* = 12.8, 2.8 Hz, 1H, 2-H), 6.65 (s, 2H, 2′-H, 6′-H), 6.95 (m, 2H, 6-H, 8-H), 7.34 (m, 3H, 7-H, 3″-H, 5″-H), 7.81 (dd, *J* = 8.4, 1.6 Hz, 1H, 5-H), 7.92 (d, 2H, 2″-H, 6″-H); ^13^C-NMR (100 MHz, CDCl_3_) δ: 21.8 (C-CH_3_), 32.0 (C-3), 56.3 (2xC-OCH_3_), 60.9 (C-OCH_3_), 77.0 (C-2), 103.3 (C-2′, C-6′), 115.6 (C-4a), 118.3 (C-8), 121.9 (C-6), 125.1 (C-5), 129.1 (C-3″, C-5″), 129.7 (C-2″, C-6″), 132.5 (C-1″), 133.5 (C-7), 134.3 (C-1′), 138.3 (C-4′), 145.3 (C-4″), 153.6 (C-3′, C-5′), 157.1 (C-4), 157.7 (C-8a); HRMS (ESI) calcd. for C_25_H_25_NO_7_S [M + H]^+^ 484.143; found 484.141.

*N-{[(4-methylphenyl)sulfonyl]oxy}-2-(naphthalene-1-yl)-2,3-dihydro-4H-chromen-4-imine* (**5f**). Off-white crystals, yield 80%, mp 136–138 °C; ^1^H-NMR (400 MHz, CDCl_3_) δ: 2.40 (s, 3H, CH_3_), 2.93 (dd, *J* = 17.6, 12.8 Hz, 1H, 3-H_a_), 3.61 (dd, *J* = 17.6, 3.2 Hz, 1H, 3-H_b_), 5.69 (dd, *J* = 12.8, 2.8 Hz, 1H, 2-H), 6.95 (m, 2H, 6-H, 8-H), 7.31 (m, 3H, 7-H, 3″-H, 5″-H), 7.44 (m, 3H, 2′-H, 3′-H, 7′-H), 7.62 (d, *J* = 6.8 Hz, 1H, 6′-H), 7.82 (m, 5H, 5-H, 4′-H, 5′-H, 8′-H, 2″-H, 6″-H); ^13^C-NMR (100 MHz, CDCl_3_) δ: 21.8 (C-CH_3_), 31.1 (C-3), 74.2 (C-2), 115.8 (C-4a), 118.4 (C-8), 122.0 (C-6), 122.9 (C-8′), 124.0 (C-5), 125.2 (C-2′), 125.4 (C-3′), 126.0 (C-7′), 126.8 (C-6′), 129.0 (C-3″, C-5″), 129.2 (C-5′), 129.5 (C-4′), 129.7 (C-2″, C-6″), 130.3 (C-8a’), 132.5 (C-1″), 133.5 (C-7), 133.9 (C-4a’), 134.0 (C-2′), 145.3 (C-4″), 157.5 (C-4), 158.1 (C-8a); HRMS (ESI) calcd. for C_26_H_21_NaNO_4_S [M+Na]^+^ 466.109; found 466.108.

*N-{[(4-methylphenyl)sulfonyl]oxy}-2-(naphthalene-2-yl)-2,3-dihydro-4H-chromen-4-imine* (**5g**). Off-white crystals, yield: 83%, mp 207–209 °C; ^1^H-NMR (400 MHz, CDCl_3_) δ: 2.45 (s, 3H, CH_3_), 2.90 (dd, *J* = 17.6, 12.4 Hz, 1H, 3-H_a_), 3.55 (dd, *J* = 17.6, 3.2 Hz, 1H, 3-H_b_), 5.22 (dd, *J* = 12.4, 3.2 Hz, 1H, 2-H), 6.96 (m, 2H, 6-H, 8-H), 7.35 (m, 3H, 7-H, 3″-H, 5″-H), 7.50 (m, 3H, 3′-H, 6′-H, 7′-H), 7.85 (m, 7H, 5-H, 1′-H, 4′-H, 5′-H, 8′-H, 2″-H, 6″-H); ^13^C-NMR (100 MHz, CDCl_3_) δ: 21.9 (C-CH_3_), 31.8 (C-3), 76.9 (C-2), 115.7 (C-4a), 118.4 (C-8), 121.9 (C-6), 123.7 (C-5), 125.2 (C-1′), 125.5 (C-8′), 126.7 (C-3′, C-7′), 127.9 (C-6′), 128.3 (C-5′), 128.9 (C-4′), 129.2 (C-3″, C-5″), 129.8 (C-2″, C-6″), 132.6 (C-8a’), 133.3 (C-1″), 133.5 (C-4a’), 133.6 (C-7), 136.1 (C-2′), 145.4 (C-4″), 157.3 (C-4), 157.9 (C-8a); HRMS (ESI) calcd. for C_26_H_21_NaNO_4_S [M+Na]^+^ 466.109; found 466.108.

### 2.3. General Procedure for Neber Rearrangement (*rac*-*cis*-***1a***-***g***, rac-trans-***1a***-***g***, ***17a***-***g***)

Tosyl oxime derivatives **5a-g** (8.895 mmol) were dissolved in anhydrous toluene (50 mL) under inert atmosphere and then 10.9 mL NaOEt (940 mg Na in 50 mL EtOH) was added dropwise to the solution. Stirring for 1 day at room temperature afforded an orange suspension. The suspension was filtered through cellite and washed with EtOH. Concentration of the filtrate under vacuum provided the crude product as an orange oil. Then it was dissolved in CH_2_Cl_2_ and 3 N HCl solution (3 mL) was added to it. After 2 h stirring at room temperature, an orange suspension was obtained. Filtration and washing with acetone provided the *cis* product as white powder. Then the filtrate was thoroughly concentrated under reduced pressure and trituration with acetone afforded the *trans* product as off-white powder. The residue filtrate was purified after concentration by column chromatography using toluene/ethyl acetate 4:1 as eluent. The benzoxazole derivates **17a-g** were obtained by this procedure.

*(±)-(2R*,3R*)-3-amino-2-phenyl-2,3-dihydro-4H-chroman-4-one hydrochloride* (*rac*-*trans*-**1a**) [5]: Off-white solid, yield 30%, mp 197–199 °C; ^1^H-NMR (400 MHz, DMSO-*d*_6_) δ: 5.01 (d, *J* = 12.4 Hz, 1H, 3-H), 5.77 (d, *J* = 12.4 Hz, 1H, 2-H), 7.13 (d, *J* = 8 Hz, 1H, 8-H), 7.20 (m, 1H, 6-H), 7.50 (m, 3H, 3′-H, 4′-H, 5′-H), 7.68 (m, 3H, 7-H, 2′-H, 6′-H), 7.87 (dd, *J* = 8.0, 1.6 Hz, 1H, 5-H), 8.72 (bs, 3H, NH_3_); ^13^C-NMR (100 MHz, DMSO-*d*_6_) δ: 55.7 (C-3), 80.2 (C-2), 118.0 (C-8), 118.5 (C-4a), 122.5 (C-6), 126.9 (C-5), 128.6 (C-2′, C-6′), 128.8 (C-3′, C-5′), 129.9 (C-4′), 134.3 (C-1′), 137.6 (C-7), 160.8 (C-8a), 187.6 (C-4); HRMS (ESI) calcd. for C_15_H_13_NO_2_ [M + H]^+^ 240.102; found 240.101.

*(±)-(2R*,3R*)-3-amino-2-(4-methoxyphenyl)-2,3-dihydro-4H-chroman-4-one hydrochloride* (*rac*-*trans*-**1b**): Off-white solid, yield 32%, mp 206–209 °C; ^1^H-NMR (400 MHz, DMSO-*d*_6_) δ: 3.80 (s, 3H, OCH_3_), 5.03 (d, *J* = 12.4 Hz, 1H, 3-H), 5.83 (d, *J* = 12.4 Hz, 1H, 2-H), 7.03 (d, 2H, 3′-H, 5′-H), 7.10 (d, *J* = 8.0 Hz, 1H, 8-H), 7.18 (t, *J* = 7.6 Hz, 1H, 6-H), 7.64 (m, 3H, 7-H, 2′-H, 6′-H), 7.85 (d, *J* = 7.6 Hz, 1H, 5-H), 8.83 (s, 3H, NH_3_); ^13^C-NMR (100 MHz, DMSO-*d*_6_) δ: 55.4 (C-OCH_3_), 55.9 (C-3), 80.0 (C-2), 114.2 (C-3′, C-5′), 118.0 (C-8), 118.6 (C-4a), 122.4 (C-6), 126.5 (C-1′), 126.9 (C-5), 130.2 (C-2′, C-6′), 137.5 (C-7), 160.4 (C-4′), 160.9 (C-8a), 187.8 (C-4); HRMS (ESI) calcd. for C_16_H_15_NO_3_ [M + H]^+^ 270.113; found 270.111.

*(±)-(2R*,3R*)-3-amino-2-(3,4-dimethoxyphenyl)-2,3-dihydro-4H-chroman-4-one hydrochloride* (*rac*-*trans*-**1c**): Off-white solid, yield 39%, mp 187–189 °C; ^1^H-NMR (400 MHz, DMSO-*d*_6_) δ: 3.80 (s, 6H, 2 × OCH_3_), 5.08 (d, *J* = 12.6 Hz, 1H, 3-H), 5.71 (d, *J* = 12.6 Hz, 1H, 2-H), 7.03 (d, *J* = 9.2 Hz, 1H, 8-H), 7.12 (m, 3H, 6-H, 5′-H, 6′-H), 7.38 (s, 1H, 2′-H), 7.67 (t, *J* = 8.0 Hz, 1H, 7-H), 7.86 (d, *J* = 8.0 Hz, 1H, 5-H), 8.71 (bs, 3H, NH_3_); ^13^C-NMR (100 MHz, DMSO-*d*_6_) δ: 55.6 (2 × C-OCH_3_), 55.9 (C-3), 80.3 (C-2), 111.7 (C-2′), 111.9 (C-5′), 118.0 (C-8), 118.5 (C-4a), 121.6 (C-6′), 122.3 (C-6), 126.5 (C-1′), 126.8 (C-5), 137.4 (C-7), 148.9 (C-4′), 150.0 (C-3′), 160.8 (C-8a), 187.7 (C-4); HRMS (ESI) calcd. for C_17_H_17_NO_4_ [M + H]^+^ 300.123; found 300.122.

*(±)-(2R*,3R*)-3-amino-2-(3,5-dimethoxyphenyl)-2,3-dihydro-4H-chroman-4-one hydrochloride* (*rac*-*trans*-**1d**): Off-white solid, yield: 46%, mp 187–189 °C; ^1^H-NMR (360 MHz, DMSO-*d*_6_) δ: 3.79 (s, 6H, 2 × OCH_3_), 5.04 (d, *J* = 12.6 Hz, 1H, 3-H), 5.68 (d, *J* = 12.6 Hz, 1H, 2-H), 6.60 (t, *J* = 1.8 Hz, 1H, 4′-H), 6.88 (d, 2 H, 2′-H, 6′-H), 7.17 (d, *J* = 8.3 Hz, 1H, 8-H), 7.20 (t, *J* = 7.9 Hz, 1H, 6-H), 7.69 (m, 1H, 7-H), 7.87 (dd, *J* = 7.9, 1.4 Hz, 1H, 5-H), 8.69 (bs, 3H, NH_3_); ^13^C-NMR (90 MHz, DMSO-*d*_6_) δ: 55.8 (2 × C-OCH_3_), 56.2 (C-3), 80.7 (C-2), 102.0 (C-4′), 107.0 (C-2′, C-6′), 118.5 (C-8), 118.9 (C-4a), 123.0 (C-6), 127.3 (C-5), 136.7 (C-1′), 138.1 (C-7), 161.2 (C-8a, C-3′, C-5′), 188.1 (C-4); HRMS (ESI) calcd. for C_17_H_17_NO_4_ [M + H]^+^ 300.123; found 300.124.

*(±)-(2R*,3R*)-3-amino-2-(3,4,5-trimethoxyphenyl)-2,3-dihydro-4H-chroman-4-one hydrochloride* (*rac*-*trans*-**1e**): Off-white solid, yield 48%, mp 188–190 °C; ^1^H-NMR (400 MHz, DMSO-*d*_6_) δ: 3.71 (s, 3H, OCH_3_), 3.82 (s, 6H, 2 × OCH_3_), 5.09 (d, *J* = 12.4 Hz, 1H, 3-H), 5.65 (d, *J* = 12.4 Hz, 1H, 2-H), 7.05 (s, 2H, 2′-H, 6′-H), 7.15 (d, *J* = 8.4 Hz, 1H, 8-H), 7.20 (m, 1H, 6-H), 7.69 (m, 1H, 7-H), 7.87 (d, *J* = 8.0, 1.6 Hz, 1H, 5-H), 8.63 (bs, 3H, NH_3_); ^13^C-NMR (100 MHz, DMSO-*d*_6_) δ: 55.9 (C-3), 56.0 (2 × C-OCH_3_), 60.0 (C-OCH_3_), 80.5 (C-2), 106.1 (C-2′, C-6′), 118.1 (C-8), 118.5 (C-4a), 122.5 (C-6), 126.9 (C-5), 129.6 (C-1′), 137.6 (C-7), 138.5 (C-4′), 153.1 (C-3′, C-5′), 160.8 (C-8a), 187.8 (C-4); HRMS (ESI) calcd. for C_18_H_19_NO_5_ [M + H]^+^ 330.134; found 330.133.

*(±)-(2R*,3R*)-3-amino-2-naphthalen-1-yl-2,3-dihydro-4H-chroman-4-one hydrochloride* (*rac*-*trans*-**1f**): Off-white solid, yield 64%, mp 212–215 °C; ^1^H-NMR (360 MHz, DMSO-*d*_6_) δ: 5.40 (d, *J* = 12.6 Hz, 1H, 3-H), 6.53 (d, *J* = 12.6 Hz, 1H, 2-H), 7.11 (d, *J* = 8.3 Hz, 1H, 8-H), 7.23 (t, *J* = 7.9 Hz, 1H, 6-H), 7.56 (m, 3H, 2′-H, 3′-H, 7′-H), 7.68 (t, *J* = 7.9 Hz, 1H, 7-H), 7.94 (m, 2H, 5-H, 6′-H), 8.04 (m, 2H, 4′-H, 5′-H), 8.49 (d, *J* = 7.9 Hz, 1H, 8′-H), 8.79 (bs, 3H, NH_3_); ^13^C-NMR (90 MHz, DMSO-*d*_6_) δ: 55.2 (C-3), 78.8 (C-2), 118.5 (C-8), 119.3 (C-4a), 123.0 (C-6), 124.6 (C-8′), 125.9 (C-2′), 126.5 (C-3′), 126.5 (C-5), 127.1 (C-7′), 127.5 (C-6′), 129.3 (C-5′), 130.1 (C-8a’), 131.1 (C-4′), 131.7 (C-1′), 134.4 (C-4a’), 137.9 (C-7), 161.3 (C-8a), 188.2 (C-4); HRMS (ESI) calcd. for C_19_H_15_NO_2_ [M + H]^+^ 290.118; found 290.118.

*(±)-(2R*,3R*)-3-amino-2-naphthalen-2-yl-2,3-dihydro-4H-chroman-4-one hydrochloride* (*rac*-*trans*-**1g**): Off-white solid, yield 48%, mp 201–203 °C; ^1^H-NMR (360 MHz, DMSO-*d*_6_) δ: 5.16 (d, *J* = 12.6 Hz, 1H, 3-H), 5.96 (d, *J* = 12.6 Hz, 1H, 2-H), 7.16 (d, *J* = 8.3 Hz, 1H, 8-H), 7.22 (m, 1H, 6-H), 7.60 (m, 2H, 3′-H, 7′-H), 7.70 (m, 1H, 7-H), 7.84 (d, *J* = 8.3 Hz, 1H, 5-H), 7.90 (d, *J* = 8.3 Hz, 1H, 6′-H), 7.97 (m, 2H, 4′-H, 5′-H), 8.06 (d, *J* = 8.6 Hz, 1H, 8′-H), 8.21 (s, 1H, 1′-H), 8.74 (bs, 3H, NH_3_); ^13^C-NMR (90 MHz, DMSO-*d*_6_) δ: 55.8 (C-3), 80.4 (C-2), 118.0 (C-6), 118.6 (C-4a), 122.5 (C-8), 125.1 (C-5), 126.5 (C-1′), 126.9 (C-8′), 127.7 (C-6′, C-7′), 128.2 (C-3′), 128.7 (C-5′), 128.8 (C-4′), 131.8 (C-2′), 132.6 (C-8a’), 133.7 (C-4a’), 137.5 (C-7), 160.8 (C-8a), 187.4 (C-4).

*(±)-(2R*,3S*)-3-amino-2-phenyl-2,3-dihydro-4H-chroman-4-one hydrochloride* (*rac*-*cis*-**1a**): White solid, yield 30%, mp 202–204 °C; ^1^H-NMR (400 MHz, DMSO-*d*_6_) δ: 5.09 (d, *J* = 5.6 Hz, 1H, 3-H), 6.23 (d, *J* = 5.6 Hz, 1H, 2-H), 7.15 (m, 2H, 6-H, 8-H), 7.38 (m, 5 H, 2′-H, 3′-H, 4′-H, 5′-H, 6′-H), 7.68 (m, 1H, 7-H), 7.82 (dd, *J* = 7.6, 1.6 Hz, 1H, 5-H), 9.09 (bs, 3H, NH_3_); ^13^C-NMR (100 MHz, DMSO-*d*_6_): δ = 54.8 (C-3), 78.6 (C-2), 118.4 (C-8), 119.6 (C-4a), 122.4 (C-6), 126.7 (C-5), 127.7 (C-2′, C-6′), 129.2 (C-3′, C-5′), 129.6 (C-4′), 133.7 (C-1′), 138.0 (C-7), 160.2 (C-8a), 187.2 (C-4); HRMS (ESI) calcd. for C_15_H_13_NO_2_ [M + H]^+^ 240.102; found 240.101.

*(±)-(2R*,3S*)-3-amino-2-(4-methoxyphenyl)-2,3-dihydro-4H-chroman-4-one hydrochloride* (*rac*-*cis*-**1b**): white solid, yield 14%. mp 186–188 °C; ^1^H-NMR (400 MHz, DMSO-*d*_6_) δ: 3.72 (s, 3H, OCH_3_), 5.11 (d, *J* = 6.0 Hz, 1H, 3-H), 6.18 (d, *J* = 6.0 Hz, 1H, 2-H), 6.90 (d, 2H, 3′-H, 5′-H), 7.11 (d, *J* = 8.4 Hz, 1H, 8-H), 7.14 (t, *J* = 7.6 Hz, 1H, 6-H), 7.28 (d, 2H, 2′-H, 6′-H), 7.65 (m, 1H, 7-H), 7.80 (d, *J* = 7.6 Hz, 1H, 5-H), 9.08 (bs, 3H, NH_3_); ^13^C-NMR (100 MHz, DMSO-*d*_6_) δ: 54.5 (C-3), 55.0 (C-OCH_3_), 78.0 (C-2), 114.1 (C-3′, C-5′), 118.0 (C-8), 119.2 (C-4a), 121.8 (C-6), 125.1 (C-1′), 126.1 (C-5), 128.9 (C-2′, C-6′), 137.5 (C-7), 159.6 (C-4′), 159.8 (C-8a), 187.1 (C-4); HRMS (ESI) calcd. for C_16_H_15_NO_3_ [M + H]^+^ 270.113; found 270.111.

*(±)-(2R*,3S*)-3-amino-2-(3,4-dimethoxyphenyl)-2,3-dihydro-4H-chroman-4-one hydrochloride* (*rac*-*cis*-**1c**): White solid, yield 23%, mp 191–195°C; ^1^H-NMR (400 MHz, DMSO-*d*_6_) δ: 3.70 (d, 6H, 2 × OCH_3_), 5.09 (d, *J* = 5.6 Hz, 1H, 3-H), 6.16 (d, *J* = 5.6 Hz, 1H, 2-H), 6.76 (dd, *J* = 8.4, 2.0 Hz, 1H, 8-H), 6.89 (d, *J* = 8.4 Hz, 1H, 5′-H), 7.14 (m, 3H, 6-H, 2′-H, 6′-H), 7.68 (m, 1H, 7-H), 7.80 (dd, *J* = 8.0, 1.6 Hz, 1H, 5-H), 9.07 (bs, 3H, NH_3_); ^13^C-NMR (100 MHz, DMSO-*d*_6_) δ: 54.8 (C-3), 55.4 (C-OCH_3_), 55.5 (C-OCH_3_), 78.2 (C-2), 111.8 (C-2′, C-5′), 118.1 (C-8), 119.2 (C-6′), 119.4 (C-4a), 122.1 (C-6), 125.5 (C-1′), 126.3 (C-5), 137.7 (C-7), 148.6 (C-4′), 149.4 (C-3′), 160.1 (C-8a), 187.2 (C-4); HRMS (ESI) calcd. for C_17_H_17_NO_4_ [M + H]^+^ 300.123; found 300.122.

*(±)-(2R*,3S*)-3-amino-2-(3,5-dimethoxyphenyl)-2,3-dihydro-4H-chroman-4-one hydrochloride* (*rac*-*cis*-**1d**): White solid, yield 20%, mp 195–197 °C; ^1^H-NMR (400 MHz, DMSO-*d*_6_) δ: 3.68 (s, 6H, 2 × OCH_3_), 5.05 (d, *J* = 5.6 Hz, 1H, 3-H), 6.15 (d, *J* = 5.6 Hz, 1H, 2-H), 6.49 (t, *J* = 2.4 Hz, 1H, 4′-H), 6.56 (d, 2H, 2′-H, 6′-H), 7.16 (m, 2H, 6-H, 8-H), 7.68 (m, 1H, 7-H), 7.79 (dd, *J* = 8.0, 1.6 Hz, 1H, 5-H), 9.08 (bs, 3H, NH_3_); ^13^C-NMR (100 MHz, DMSO-*d*_6_) δ: 54.6 (C-3), 55.3 (2 × C-OCH_3_), 78.2 (C-2), 100.5 (C-4′), 105.5 (C-2′, C-6′), 118.1 (C-8), 119.4 (C-4a), 122.2 (C-6), 126.4 (C-5), 135.5 (C-1′), 137.8 (C-7), 160.1 (C-8a), 160.6 (C-3′, C-5′), 186.8 (C-4); HRMS (ESI) calcd. for C_17_H_17_NO_4_ [M + H]^+^ 300.123; found 300.124.

*(±)-(2R*,3S*)-3-amino-(3,4,5-trimethoxyphenyl)-2,3-dihydro-4H-chroman-4-one hydrochloride* (*rac*-*cis*-**1e**): Off-white solid, yield 17%, mp 194–196 °C; ^1^H-NMR (400 MHz, DMSO-*d*_6_) δ: 3.64 (d, 9H, 3 × OCH_3_), 5.01 (d, *J* = 5.2 Hz, 1H, 3-H), 6.14 (d, *J* = 5.2 Hz, 1H, 2-H), 6.75 (s, 2H, 2′-H, 6′-H), 7.17 (m, 2H, 6-H, 8-H), 7.70 (t, *J* = 7.6 Hz, 1H, 7-H), 7.82 (d, *J* = 8.0 Hz, 1H, 5-H), 9.08 (bs, 3H, NH_3_); ^13^C-NMR (100 MHz, DMSO-*d*_6_) δ: 54.8 (C-3), 55.8 (2 × C-OCH_3_), 59.9 (C-OCH_3_), 78.4 (C-2), 104.8 (C-2′, C-6′), 118.2 (C-8), 119.4 (C-4a), 122.3 (C-6), 126.4 (C-5), 128.8 (C-1′), 137.8 (C-7), 137.9 (C-4′), 153.0 (C-3′, C-5′), 160.2 (C-8a), 187.0 (C-4); HRMS (ESI) calcd. for C_18_H_19_NO_5_ [M + H]^+^ 330.134; found 330.133.

*2. -[(E)-2-phenylethenyl]-1,3-benzoxazole (**17a**)* [18]: Pale yellow crystals, yield 15%, mp 62–64 °C; ^1^H-NMR (400 MHz, CDCl_3_) δ: 7.05 (d, *J* = 16.4 Hz, 1H, 2′-H), 7.30 (m, 2H, 4-H, 7-H), 7.36 (m, 3H, 5′-H, 6′-H, 7′-H), 7.50 (m, 1H, 5-H), 7.57 (d, 2H, 4′-H, 8′-H), 7.69 (m, 1H, 6-H), 7.76 (d, *J* = 16.4 Hz, 1H, 1′-H); ^13^C-NMR (100 MHz, CDCl_3_) δ: 110.4 (C-2′), 114.0 (C-5), 119.9 (C-6), 124.6 (C-4), 125.3 (C-7), 127.6 (C-5′, C-7′), 129.0 (C-4′, C-8′), 129.9 (C-6′), 135.2 (C-3′), 139.6 (C-1′), 142.2 (C-3a), 150.5 (C-7a), 162.9 (C-2); HRMS (ESI) calcd. for C_15_H_11_NO [M + H]^+^ 222.092; found 222.089.

*2. -[(E)-2-(4-methoxyphenyl)ethenyl]-1,3-benzoxazole (**17b**)* [19]: White chrystals, yield 20%, mp 128–130 °C; ^1^H-NMR (400 MHz, CDCl_3_) δ: 3.79 (s, 3H, OCH_3_), 6.88 (m, 3H, 2′-H, 5′-H, 7′-H), 7.27 (m, 2H, 4-H, 7-H), 7.46 (m, 3H, 5-H, 4′-H, 8′-H), 7.67 (m, 2H, 1’-H, 6-H); ^13^C-NMR (100 MHz, CDCl_3_) δ: 55.4 (C-OCH_3_), 110.2 (C-5), 111.5 (C-2′), 114.4 (C-5′, C-7′), 119.7 (C-6), 124.4 (C-4), 124.9 (C-7), 127.9 (C-3′), 129.1 (C-4′, C-8′), 139.1 (C-1′), 142.3 (C-3a), 150.4 (C-7a), 161.0 (C-6′), 163.2 (C-2); HRMS (ESI) calcd. for C_16_H_13_NO_2_ [M + H]^+^ 252.102; found 252.103.

*2. -[(E)-2-(3,4-dimethoxyphenyl)ethenyl]-1,3-benzoxazole (**17c**)*: Pale yellow crystals, yield: 16%, mp 119–120 °C; ^1^H-NMR (400 MHz, CDCl_3_) δ: 3.90 (d, 6H, 2 × OCH_3_), 6.86 (d, *J* = 8.0 Hz, 1H, 7′-H), 6.91 (d, *J* = 16.4 Hz, 1H, 2′-H), 7.11 (s, 1H, 4′-H), 7.13 (d, *J* = 8.0 Hz, 1H, 8′-H), 7.29 (m, 2H, 4-H, 7-H), 7.48 (m, 1H, 5-H), 7.68 (m, 2H, 6-H, 1′-H); ^13^C-NMR (100 MHz, CDCl_3_) δ: 55.9 (C-OCH_3_), 56.0 (C-OCH_3_), 109.3 (C-4′), 110.2 (C-5), 111.2 (C-7′), 111.8 (C-2′), 119.7 (C-6), 121.9 (C-8′), 124.5 (C-4), 125.0 (C-7), 128.3 (C-3′), 139.3 (C-1′), 142.3 (C-3a), 149.4 (C-6′), 150.4 (C-5′), 150.8 (C-7a), 163.2 (C-2); HRMS (ESI) calcd. for C_17_H_15_NO_3_ [M + H]^+^ 282.113; found 282.112.

*2. -[(E)-2-(3,4,5-trimethoxyphenyl)ethenyl]-1,3-benzoxazole (**17e**)*: Pale yellow crystals, yield 14%, mp. 147–149 °C; ^1^H-NMR (400 MHz, CDCl_3_) δ: 3.91 (s, 9 H, 3 × OCH_3_), 6.82 (s, 2H, 4′-H, 8′-H), 6.96 (d, *J* = 16.0 Hz, 1H, 2′-H), 7.33 (m, 2H, 4-H, 7-H), 7.50 (m, 1H, 5-H), 7.68 (m, 2H, 6-H, 1′-H); ^13^C-NMR (100 MHz, CDCl_3_) δ: 56.2 (2 × C-OCH_3_), 61.1 (C-OCH_3_), 104.7 (C-4′, C-8′), 110.3 (C-5), 113.3 (C-2′), 119.9 (C-6), 124.6 (C-4), 125.2 (C-7), 130.8 (C-3′), 139.4 (C-1′), 139.8 (C-6′), 142.3 (C-3a), 150.5 (C-7a), 153.6 (C-5′, C-7′), 162.8 (C-2); HRMS (ESI) calcd. for C_18_H_17_NO_4_ [M + H]^+^ 312.123; found 312.125.

*2. -[(E)-2-(naphthalen-1-yl)ethenyl]-1,3-benzoxazole (**17f**)*: Pale yellow crystals, yield: 10%, mp 124–126 °C; ^1^H-NMR (400 MHz, CDCl_3_) δ: 7.15 (d, *J* = 16.0 Hz, 1H, 2′-H), 7.34 (m, 2H, 4-H, 7-H), 7.50 (m, 4 H, 5-H, 4′-H, 5′-H, 9′-H), 7.74 (m, 1H, 6-H), 7.84 (m, 3H, 6′-H, 7′-H, 8′-H), 8.28 (d, *J* = 8.4 Hz, 1H, 10′-H), 8.59 (d, *J* = 16.0 Hz, 1H, 1′-H); ^1^H-NMR (400 MHz, CDCl_3_) δ: 110.5 (C-5), 116.5 (C-2′), 120.1 (C-6), 123.5 (C-10′), 124.6 (C-4), 124.7 (C-4′), 125.4 (C-7), 125.7 (C-5′), 126.3 (C-9′), 126.9 (C-8′), 128.9 (C-7′), 130.2 (C-6′), 131.4 (C-6a’), 132.6 (C-10a’), 133.9 (C-3′), 136.4 (C-1′), 142.4 (C-3a), 150.6 (C-7a), 162.9 (C-2); HRMS (ESI) calcd. for C_19_H_13_NO [M + H]^+^ 272.107; found 272.106.

*2. -[(E)-2-(naphthlaen-2-yl)ethenyl]-1,3-benzoxazole (**17g**)* [18]: White crystals, yield 11%, mp 129–131 °C; ^1^H-NMR (400 MHz, CDCl_3_) δ: 7.11 (d, *J* = 16.4 Hz, 1H, 2′-H), 7.28 (m, 2H, 4-H, 7-H), 7.45 (m, 3H, 5-H, 4′-H, 8′-H), 7.68 (m, 2H, 6-H, 7′-H), 7.77 (m, 3H, 5′-H, 6′-H, 9′-H), 7.90 (m, 2H, 10′-H, 1′-H); ^1^H-NMR (400 MHz, CDCl_3_) δ: 110.4 (C-5), 114.1 (C-2′), 120.0 (C-6), 123.2 (C-10′), 124.6 (C-4), 125.3 (C-7), 126.8 (C-9′), 127.1 (C-4′), 127.9 (C-8′), 128.5 (C-7′), 128.8 (C-6′), 129.2 (C-5′), 132.7 (C-5a’), 133.5 (C-9a’), 134.0 (C-3′), 139.5 (C-1′), 142.3 (C-3a), 150.5 (C-7a), 162.9 (C-2); HRMS (ESI) calcd. for C_19_H_13_NO [M + H]^+^ 272.107; found 272.107.

### 2.4. General Procedure for the Synthesis of 2-Chloroacetamide Derivatives (*rac*-*trans*-***18a***-***g***)

The hydrochloride salt of 3-aminoflavanone derivatives *rac-cis*-**1a-g** or *rac-trans*-**1a-g** (1.452 mmol) was suspended in anhydrous THF under inert atmosphere. After addition of Et_3_N (510 µL, 3.630 mmol), the reaction was stirred for 5 min at room temperature or at 0 °C. Then chloroacetyl chloride (139 µL, 1.742 mmol) was added dropwise to the suspension and it was stirred further for 15 min. The reaction was quenched with water and then it was extracted with CH_2_Cl_2_. The combined organic phases were dried over MgSO_4_ and concentrated under reduced pressure. Trituration with cold Et_2_O afforded the pure product.

*(±)-2-chloro-N-[(2R*,3R*)-2-phenyl-4-oxochroman-3-yl]acetamide* (*rac*-*trans*-**18a**): White crystals, yield 77%, mp 214–216 °C; ^1^H-NMR (400 MHz, CDCl_3_) δ: 3.82 (d, *J* = 15.6 Hz, 1H, CH_2_-H_a_), 3.95 (d, *J* = 15.6 Hz, 1H, CH_2_-H_b_), 5.08 (dd, *J* = 12.0, 8.4 Hz, 1H, 3-H), 5.39 (d, *J* = 12.0 Hz, 1H, 2-H), 6.79 (d, *J* = 8.0 Hz, 1H, NH), 7.06 (d, *J* = 8.4 Hz, 1H, 8-H), 7.09 (t, *J* = 7.6 Hz, 1H, 6-H), 7.41 (m, 3H, 3′-H, 4′-H, 5′-H), 7.50 (m, 3H, 7-H, 2′-H, 6′-H), 7.93 (dd, *J* = 7.6, 1.2 Hz, 1H, 5-H); ^13^C-NMR (100 MHz, CDCl_3_) δ: 42.4 (C-CH_2_), 58.4 (C-3), 83.2 (C-2), 118.2 (C-8), 120.0 (C-4a), 122.4 (C-6), 127.7 (C-2′, C-6′), 127.8 (C-5), 128.7 (C-3′, C-5′), 129.6 (C-4′), 135.7 (C-1′), 136.9 (C-7), 161.4 (C-8a), 166.2 (amide carbonyl), 189.9 (C-4); HRMS (ESI) calcd. for C_17_H_14_ClNaNO_3_ [M+Na]^+^ 338.056; found 338.056.

*(±)-2-chloro-N-[(2R*,3R*)-2-(4-methoxyphenyl)-4-oxochroman-3-yl]acetamide* (*rac*-*trans*-**18b**): White crystals, yield 71%, mp 179–180 °C; ^1^H-NMR (400 MHz, DMSO-*d*_6_) δ: 3.77 (s, 3H, OCH_3_), 3.99 (m, 2H, CH_2_), 5.00 (dd, *J* = 12.4, 8.4 Hz, 1H, 3-H), 5.57 (d, *J* = 12.4 Hz, 1H, 2-H), 6.95 (d, 2H, 3′-H, 5′-H), 7.07 (d, *J* = 8.0 Hz, 1H, 8-H), 7.13 (m, 1H, 6-H), 7.42 (d, 2H, 2′-H, 6′-H), 7.61 (m, 1H, 7-H), 7.81 (dd, *J* = 7.6, 1.6 Hz, 1H, 5-H), 8.51 (d, *J* = 8.4 Hz, 1H, NH); ^13^C-NMR (100 MHz, DMSO-*d*_6_) δ: 42.3 (C-CH_2_), 55.3 (C-OCH_3_), 57.8 (C-3), 81.2 (C-2), 113.8 (C-3′, C-5′), 118.2 (C-8), 120.1 (C-4a), 122.1 (C-6), 127.1 (C-5), 128.9 (C-1′), 129.4 (C-2′, C-6′), 136.7 (C-7), 159.8 (C-4′), 161.1 (C-8a), 166.2 (amide carbonyl), 190.0 (C-4); HRMS (ESI) calcd. for C_18_H_16_ClNaNO_4_ [M+Na]^+^ 368.066; found 368.067.

*(±)-2-chloro-N-[(2R*,3R*)-2-(3,4-dimethoxyphenyl)-4-oxochroman-3-yl]acetamide* (*rac*-*trans*-**18c**): White crystals, yield 80%, mp 183–185 °C; ^1^H-NMR (400 MHz, DMSO-*d*_6_) δ: 3.77 (s, 6H, 2xOCH_3_), 3.96 (m, 2H, CH_2_), 5.03 (dd, *J* = 12.4, 8.4 Hz, 1H, 3-H), 5.56 (d, *J* = 12.4 Hz, 1H, 2-H), 6.95 (d, *J* = 8.4 Hz, 1H, 5′-H), 7.00 (dd, *J* = 8.4, 1.6 Hz, 1H, 6′-H), 7.08 (d, *J* = 8.0 Hz, 1H, 8-H), 7.12 (m, 2H, 6-H, 2′-H), 7.60 (m, 1H, 7-H), 7.82 (dd, *J* = 7.6, 1.6 Hz, 1H, 5-H), 8.52 (d, *J* = 8.4 Hz, 1H, NH); ^13^C-NMR (100 MHz, DMSO-*d*_6_) δ: 42.2 (C-CH_2_), 55.5 (2xC-OCH_3_), 57.6 (C-3), 81.2 (C-2), 109.5 (C-2′), 111.2 (C-5′), 118.0 (C-8), 119.9 (C-4a), 120.7 (C-6′), 121.9 (C-6), 126.9 (C-5), 129.0 (C-1′), 136.4 (C-7), 148.4 (C-4′), 149.2 (C-3′), 160.8 (C-8a), 166.0 (amide carbonyl), 189.8 (C-4); HRMS (ESI) calcd. for C_19_H_18_ClNaNO_5_ [M+Na]^+^ 398.077; found 398.078.

*(±)-2-chloro-N-[(2R*,3R*)-2-(3,5-dimethoxyphenyl)-4-oxochroman-3-yl]acetamide* (*rac*-*trans*-**18d**): White crystals, yield 80%, mp 206–208 °C; ^1^H-NMR (400 MHz, DMSO-*d*_6_) δ: 3.74 (s, 6 H, 2 × OCH_3_), 4.01 (s, 2H, CH_2_), 4.96 (dd, *J* = 12.0, 8.4 Hz, 1H, 3-H), 5.57 (d, *J* = 12.0 Hz, 1H, 2-H), 6.49 (t, *J* = 2.4 Hz, 1H, 4′-H), 6.68 (d, 2H, 2′-H, 6′-H), 7.09 (d, *J* = 8.0 Hz, 1H, 8-H), 7.13 (t, *J* = 7.6 Hz, 1H, 6-H), 7.61 (m, 1H, 7-H), 7.80 (dd, *J* = 7.6, 1.6 Hz, 5-H), 8.67 (d, *J* = 8.4 Hz, 1H, NH); ^13^C-NMR (100 MHz, DMSO-*d*_6_) δ: 42.1 (C-CH_2_), 55.3 (2 × C-OCH_3_), 57.6 (C-3), 81.0 (C-2), 100.7 (C-4′), 105.8 (C-2′, C-6′), 118.0 (C-8), 119.8 (C-4a), 122.0 (C-6), 126.9 (C-5), 136.5 (C-7), 138.8 (C-1′), 160.2 (C-3′, C-5′), 160.7 (C-8a), 166.1 (amide carbonyl), 189.5 (C-4); HRMS (ESI) calcd. for C_19_H_18_ClNaNO_5_ [M+Na]^+^ 398.077; found 398.077.

*(±)-2-chloro-N-[(2R*,3R*)-2-(3,4,5-trimethoxyphenyl)-4-oxochroman-3-yl]acetamide* (*rac*-*trans*-**18e**): White crystals, yield 74%, mp 140–142 °C; ^1^H-NMR (400 MHz, CDCl_3_) δ: 3.83 (m, 9H, 3 × OCH_3_), 3.84 (m, 2H, CH_2_), 5.03 (dd, *J* = 12.4, 8.8 Hz, 1H, 3-H), 5.41 (d, *J* = 12.4 Hz, 1H, 2-H), 6.74 (s, 2H, 2′-H, 6′-H), 6.99 (m, 2H, 6-H, 8-H), 7.29 (d, *J* = 8.8 Hz, 1H, NH), 7.47 (m, 1H, 7-H), 7.80 (d, *J* = 7.6, 1.2 Hz, 1H, 5-H); ^13^C-NMR (100 MHz, CDCl_3_) δ: 41.9 (C-CH_2_), 55.8 (2 × C-OCH_3_), 57.8 (C-3), 60.4 (C-OCH_3_), 82.2 (C-2), 104.4 (C-2′, C-6′), 117.7 (C-8), 119.5 (C-4a), 121.8 (C-6), 127.1 (C-5), 131.0 (C-1′), 136.2 (C-7), 138.1 (C-4′), 152.8 (C-3′, C-5′), 160.7 (C-8a), 166.3 (amide carbonyl), 189.5 (C-4); HRMS (ESI) calcd. for C_20_H_20_ClNaNO_6_ [M+Na]^+^ 428.088; found 428.089.

*(±)-2-chloro-N-[(2R*,3R*)-2-(naphthalen-1-yl)-4-oxochroman-3-yl]acetamide* (*rac*-*trans*-**18f**): White crystals, yield 79%, mp 252–254 °C; ^1^H-NMR (400 MHz, DMSO-*d*_6_) δ: 3.80 (q, 2H, CH_2_), 5.33 (t, *J* = 10.8 Hz, 1H, 3-H), 6.48 (d, *J* = 12.0 Hz, 1H, 2-H), 7.09 (d, *J* = 8.0 Hz, 1H, 8-H), 7.18 (t, *J* = 7.6 Hz, 1H, 6-H), 7.55 (m, 4 H, 7-H, 2′-H, 3′-H, 7′-H), 7.82 (s, 1H, 6′-H), 7.90 (d, *J* = 7.6 Hz, 1H, 5-H), 7.98 (m, 2H, 4′-H, 5′-H), 8.29 (s, 1H, 8′-H), 8.57 (d, *J* = 8.0 Hz, 1H, NH); ^13^C-NMR (100 MHz, DMSO-*d*_6_) δ: 42.0 (C-CH_2_), 57.3 (C-3), 78.1 (C-2), 118.0 (C-8), 120.1 (C-4a), 122.1 (C-6), 125.2 (C-5), 125.8 (C-2′, C-3′), 126.5 (C-7′), 127.0 (C-6′), 128.7 (C-4′), 129.5 (C-5′), 131.2 (C-4a’), 132.4 (C-8a’), 133.3 (C-1′), 136.5 (C-7), 160.9 (C-8a), 166.1 (amide carbonyl), 189.6 (C-4); HRMS (ESI) calcd. for C_21_H_16_ClNaNO_3_ [M+Na]^+^ 388.072; found 388.073.

*(±)-2-chloro-N-[(2R*,3R*)-2-(naphthalen-2-yl)-4-oxochroman-3-yl]acetamide* (*rac*-*trans*-**18g**). White crystals, yield, 82%, mp 226–228 °C; ^1^H-NMR (360 MHz, DMSO-*d*_6_) δ: 3.91 (q, 2H, CH_2_), 5.11 (dd, *J* = 12.2, 8.3 Hz, 1H, 3-H), 5.82 (d, *J* = 12.2 Hz, 1H, 2-H), 7,12 (m, 2H, 6-H, 8-H), 7.54 (m, 2H, 3′-H, 7′-H), 7.62 (m, 1H, 7-H), 7.69 (dd, *J* = 8.6, 1.4 Hz, 1H, 5-H), 7.85 (dd, *J* = 7.6, 1.4 Hz, 1H, 6′-H), 7.94 (m, 4 H, 1′-H, 4′-H, 5′-H, 8′-H), 8.59 (d, *J* = 8.3 Hz, 1H, NH); ^13^C-NMR (90 MHz, DMSO-*d*_6_) δ: 42.0 (C-CH_2_), 57.7 (C-3), 81.3 (C-2), 118.0 (C-8), 120.0 (C-4a), 122.0 (C-6), 124.9 (C-5), 126.3 (C-1′), 126.5 (C-8′), 126.9 (C-3′), 127.2 (C-7′), 127.5 (C-6′), 127.8 (C-5′), 128.0 (C-4′), 132.4 (C-8a’), 133.1 (C-4a’), 134.2 (C-2′), 136.4 (C-7), 160.7 (C-8a), 166.0 (amide carbonyl), 189.4 (C-4); HRMS (ESI) calcd. for C_21_H_16_ClNaNO_3_ [M+Na]^+^ 388.072; found 388.073.

### 2.5. General Procedure for the Synthesis of Flavan-4-ol Derivatives rac-***19a***-***g*** and rac-***22a***-***g***

To the solution of the chloroacetamide derivatives (0.958 mmol) in MeOH (10 mL), NaBH_4_ (1.150 mmol) was added and reaction mixture was stirred at room temperature. The reaction was completed in 10 min. The pH was adjusted to about 5 with 10% HCl solution and the mixture was concentrated and the residue was extracted with ethyl acetate and water. The combined organic phase was dried over MgSO_4_ and the solvent was evaporated in vacuum. Column chromatography with CHCl_3_ as eluent provided the pure product.

*(±)-2-chloro-N-[(2R*,3S*,4R*)-4-Hydroxy-2-phenyl-3,4-dihydro-2H-chromen-3-yl]acetamide* (*rac*-**19a**): White crystals, yield 90%, mp 157–159 °C; ^1^H-NMR (400 MHz, acetone-*d*_6_) δ: 3.76 (d, *J* = 14.0 Hz, 1H, CH_2_-H_a_), 3.86 (d, *J* = 14.0 Hz, 1H, CH_2_-H_b_), 4.31 (q, *J* = 9.2 Hz, 1H, 3-H), 4.87 (d, *J* = 6.4 Hz, 1H, OH), 5.11 (dd, *J* = 9.2, 6.4 Hz, 1H, 4-H), 5.22 (d, *J* = 10.4 Hz, 1H, 2-H), 6.80 (dd, *J* = 8.0, 0.8 Hz, 1H, 8-H), 6.96 (m, 1H, 6-H), 7.17 (m, 1H, 7-H), 7.32 (m, 3H, 3′-H, 4′-H, 5′-H), 7.47 (m, 2H, 2′-H, 6′-H), 7.54 (d, *J* = 8.0 Hz, 1H, 5-H), 7.61 (d, *J* = 8.8 Hz, NH); ^13^C-NMR (100 MHz, acetone-*d*_6_) δ: 43.2 (C-CH_2_), 56.6 (C-3), 69.6 (C-4), 80.5 (C-2), 116.7 (C-8), 121.7 (C-6), 126.6 (C-4a), 128.7 (C-2′, C-6′), 128.8 (C-3′, C-5′), 128.9 (C-4′), 129.1 (C-5), 129.5 (C-7), 138.8 (C-1′), 154.9 (C-8a), 166.6 (amide carbonyl); HRMS (ESI) calcd. for C_17_H_16_ClNaNO_3_ [M+Na]^+^ 340.071; found 340.073.

(*±)-2-chloro-N-[(2R*,3S*,4R*)-4-Hydroxy-2-(4-methoxyphenyl)-3,4-dihydro-2H-chromen-3-yl]acetamide* (*rac*-**19b**): White crystals, yield 92%, mp 163–164 °C; ^1^H-NMR (400 MHz, DMSO-*d*_6_) δ: 3.74 (bs, 1H, OH), 3.81 (m, 4 H, CH_2_-H_a_, OCH_3_), 3.99 (d, *J* = 15.6 Hz, 1H, CH_2_-H_b_), 4.27 (m, 1H, 3-H), 4.99 (d, 2H, 2-H, 4-H), 6.51 (d, *J* = 6.4 Hz, 1H, NH), 6.88 (d, *J* = 7.2 Hz, 1H, 8-H), 6.92 (d, 2H, 3′-H, 5′-H), 7.02 (m, 1H, 6-H), 7.21 (m, 1H, 7-H), 7.35 (d, 2H, 2′-H, 6′-H), 7.54 (d, *J* = 7.6 Hz, 1H, 5-H); ^13^C-NMR (100 MHz, DMSO-*d*_6_) δ: 42.9 (C-CH_2_), 55.8 (C-OCH_3_), 57.5 (C-3), 70.8 (C-4), 79.4 (C-2), 114.8 (C-3′, C-5′), 116.9 (C-8), 122.3 (C-6), 124.4 (C-4a), 128.6 (C-1′), 128.7 (C-5), 129.3 (C-2′, C-6′), 129.9 (C-7), 154.0 (C-8a), 160.8 (C-4′), 167.5 (amide carbonyl); HRMS (ESI) calcd. for C_18_H_18_NaNO_4_ [M+Na]^+^ 370.109; found 370.110.

*(±)-2-chloro-N-[(2R*,3S*,4R*)-4-Hydroxy-2-(3,4-dimethoxyphenyl)-3,4-dihydro-2H-chromen-3-yl]acetamide* (*rac*-**19c**): White crystals, yield 98%, mp 156–158 °C; ^1^H-NMR (400 MHz, CDCl_3_) δ: 3.78 (d, *J* = 15.2 Hz, 1H, CH_2_-H_a_), 3.87 (d, 6H, 2 × OCH_3_), 3.92 (m, 2H, CH_2_-H_b_, OH), 4.26 (q, *J* = 9.2 Hz, 1H, 3-H), 4.96 (m, 2H, 2-H, 4-H), 6.56 (d, *J* = 6.8 Hz, 1H, NH), 6.85 (m, 2H, 2′-H, 5′-H), 6.95 (m, 3H, 8-H, 2′-H, 6′-H), 7.00 (t, *J* = 7.6 Hz, 1H, 6-H), 7.19 (t, *J* = 8.0 Hz, 1H, 7-H), 7.51 (d, *J* = 7.6 Hz, 1H, 5-H); ^13^C-NMR (100 MHz, CDCl_3_) δ: 42.6 (C-CH_2_), 56.0 (C-OCH_3_), 56.1 (C-OCH_3_), 56.9 (C-3), 70.3 (C-4), 79.3 (C-2), 110.1 (C-5′), 111.1 (C-2′), 116.6 (C-6′), 120.5 (C-8), 121.9 (C-6), 124.2 (C-4a), 128.3 (C-5), 128.7 (C-1′), 129.5 (C-7), 149.5 (C-4′), 149.9 (C-3′), 153.6 (C-8a), 167.2 (amide carbonyl); HRMS (ESI) calcd. for C_19_H_20_ClNaNO_5_ [M+Na]^+^ 400.092; found 400.094.

*(±)-2-chloro-N-[(2R*,3S*,4R*)-4-Hydroxy-2-(3,5-dimethoxyphenyl)-3,4-dihydro-2H-chromen-3-yl]acetamide* (*rac*-**19d**). White crystals, yield 94%, mp 160–162 °C; ^1^H-NMR (400 MHz, acetone-*d*_6_) δ: 3.78 (d, 6H, 2xOCH_3_), 3.83 (d, *J* = 14.0 Hz, 1H, CH_2_-H_a_), 3.92 (d, *J* = 14.0 Hz, 1H, CH_2_-H_b_), 4.31 (q, *J* = 9.2 Hz, 1H, 3-H), 4.90 (d, *J* = 6.0 Hz, 1H, OH), 5.11 (dd, *J* = 9.2, 6.0 Hz, 1H, 4-H), 5.17 (d, *J* = 10.4 Hz, 1H, 2-H), 6.44 (t, *J* = 2.4 Hz, 1H, 4′-H), 6.69 (d, 2H, 2′-H, 6′-H), 6.81 (d, *J* = 8.4 Hz, 1H, 8-H), 6.96 (m, 1H, 6-H), 7.17 (m, 1H, 7-H), 7.54 (d, *J* = 7.6 Hz, 1H, 5-H), 7.64 (d, *J* = 9.2 Hz, 1H, NH); ^13^C-NMR (101 MHz, acetone-*d*_6_) δ: 44.2 (C-CH_2_), 56.6 (2xC-OCH_3_), 57.4 (C-3), 70.6 (C-4), 81.4 (C-2), 102.2 (C-4′), 107.5 (C-2′, C-6′), 117.7 (C-8), 122.7 (C-6), 127.5 (C-4a), 129.8 (C-5), 130.4 (C-7), 141.9 (C-1′), 155.8 (C-8a), 162.4 (C-3′, C-5′), 167.7 (amide carbonyl); HRMS (ESI) calcd. for C_19_H_19_NaNO_5_ [M+Na]^+^ 364.116; found 364.118; HRMS (ESI) calcd. for C_19_H_20_ClNaNO_5_ [M+Na]^+^ 400.092; found 400.094.

*(±)-2-chloro-N-[(2R*,3S*,4R*)-4-Hydroxy-2-(3,4,5-trimethoxyphenyl)-3,4-dihydro-2H-chromen-3-yl]acetamide* (*rac*-**19e**): White crystals, yield 92%, mp 155–157 °C; ^1^H-NMR (400 MHz, CDCl_3_) δ: 3.79 (m, 10 H, CH_2_-H_a_, 3xOCH_3_), 3.92 (d, *J* = 15.2 Hz, 1H, CH_2_-H_b_), 4.06 (bs, 1H, OH), 4.23 (q, *J* = 8.8 Hz, 1H, 3-H), 4.96 (m, 2H, 2-H, 4-H), 6.64 (m, 3H, NH, 2′-H, 6′-H), 6.88 (d, *J* = 8.4 Hz, 1H, 8-H), 7.00 (t, *J* = 7.6 Hz, 1H, 6-H), 7.20 (t, *J* = 7.2 Hz, 1H, 7-H), 7.50 (d, *J* = 7.6 Hz, 1H, 5-H); ^13^C-NMR (100 MHz, CDCl_3_) δ: 42.2 (C-CH_2_), 55.9 (2xC-OCH_3_), 56.6 (C-3), 60.5 (C-OCH_3_), 69.6 (C-4), 79.2 (C-2), 104.2 (C-2′, C-6′), 116.1 (C-8), 121.6 (C-6), 123.8 (C-4a), 127.8 (C-5), 129.1 (C-7), 131.5 (C-1′), 138.2 (C-4′), 153.1 (C-8a), 153.2 (C-3′, C-5′), 166.8 (amide carbonyl); HRMS (ESI) calcd. for C_20_H_22_ClNaNO_6_ [M+Na]^+^ 430.103; found 430.104.

*(±)-2-chloro-N-[(2R*,3S*,4R*)-4-Hydroxy-2-(naphthalen-1-yl)-3,4-dihydro-2H-chromen-3-yl]acetamide* (*rac*-**19f**): White crystals, yield 88%, mp 247–249 °C; ^1^H-NMR (400 MHz, DMSO-*d*_6_) δ: 3.64 (m, 2H, CH_2_), 4.49 (q, *J* = 9.6 Hz, 1H, 3-H), 5.00 (t, *J* = 8.0 Hz, 1H, 4-H), 5.77 (d, *J* = 6.8 Hz, 1H, OH), 5.89 (d, *J* = 10.4 Hz, 1H, 2-H), 6.80 (d, *J* = 8.0 Hz, 1H, 8-H), 7.00 (t, *J* = 7.6 Hz, 1H, 6-H), 7.18 (t, *J* = 7.2 Hz, 1H, 7-H), 7.48 (m, 4 H, 2′-H, 3′-H, 6′-H, 7′-H), 7.65 (d, *J* = 7.2 Hz, 1H, 5-H), 7.90 (m, 2H, 4′-H, 5′-H), 8.21 (d, *J* = 9.2 Hz, 1H, NH), 8.28 (d, *J* = 9.2 Hz, 1H, 8′-H); ^13^C-NMR (100 MHz, DMSO-*d*_6_) δ: 42.4 (C-CH_2_), 53.8 (C-3), 67.9 (C-4), 76.2 (C-2), 115.7 (C-8), 120.9 (C-6), 124.0 (C-8′), 125.1 (C-2′, C-3′), 125.5 (C-5), 126.0 (C-7′), 126.2 (C-4a), 128.1 (C-6′), 128.5 (C-7, C-5′), 128.8 (C-4′), 131.4 (C-1′), 133.2 (C-4a’), 133.3 (C-8a’), 153.5 (C-8a), 165.3 (amide carbonyl); HRMS (ESI) calcd. for C_21_H_18_ClNaNO_3_ [M+Na]^+^ 390.087; found 390.088.

*(±)-2-chloro-N-[(2R*,3S*,4R*)-4-Hydroxy-2-(naphthalen-2-yl)-3,4-dihydro-2H-chromen-3-yl]acetamide* (*rac*-**19g**): White crystals, yield 89%, mp 210–212 °C; ^1^H-NMR (400 MHz, acetone-*d*_6_) δ: 3.69 (d, *J* = 14.0 Hz, 1H, CH_2_-H_a_), 3.83 (d, *J* = 14.0 Hz, 1H, CH_2_-H_b_), 4.41 (q, *J* = 9.2 Hz, 1H, 3-H), 4.89 (d, *J* = 6.4 Hz, 1H, OH), 5.20 (dd, *J* = 9.2, 6.8 Hz, 1H, 4-H), 5.43 (d, *J* = 10.4 Hz, 1H, 2-H), 6.85 (d, *J* = 8.4 Hz, 1H, 8-H), 6.99 (m, 1H, 6-H), 7.20 (m, 1H, 7-H), 7.50 (m, 2H, 3′-H, 7′-H), 7.58 (d, *J* = 7.6 Hz, 1H, 5-H), 7.64 (m, 2H, 6′-H, NH), 7.88 (m, 3H, 4′-H, 5′-H, 8′-H), 7.98 (s, 1H, 1′-H); ^13^C-NMR (100 MHz, acetone-*d*_6_) δ: 43.2 (C-CH_2_), 56.7 (C-3), 69.6 (C-4), 80.6 (C-2), 116.8 (C-8), 121.8 (C-6), 126.1 (C-8′), 126.7 (C-4a), 126.9 (C-1′), 127.0 (C-3′), 128.2 (C-5), 128.5 (C-7′), 128.6 (C-6′), 128.9 (C-4′, C-5′), 129.5 (C-7), 133.9 (C-4a’), 134.4 (C-8a’), 136.4 (C-2′), 154.9 (C-8a), 166.6 (amide carbonyl); HRMS (ESI) calcd. for C_21_H_18_ClNaNO_3_ [M+Na]^+^ 390.087; found 390.087.

*(±)-2-chloro-N-[(2R*,3R*,4R*)-4-Hydroxy-2-phenyl-3,4-dihydro-2H-chromen-3-yl]acetamide* (*rac*-**22a**): White crystals, overall yield of acylation and reduction 62%, mp 142–144 °C; ^1^H-NMR (400 MHz, CDCl_3_) δ: 3.70 (d, *J* = 15.2 Hz, 1H, CH_2_-H_a_), 3.84 (d, *J* = 15.2 Hz, 1H, CH_2_-H_b_), 3.90 (d, *J* = 5.2 Hz, 1H, OH), 4.72 (dd, *J* = 9.2, 5.2 Hz, 1H, 3-H), 5.27 (t, *J* = 5.2 Hz, 1H, 4-H), 5.34 (s, 1H, 2-H), 6.80 (d, *J* = 9.2 Hz, 1H, NH), 6.98 (d, *J* = 8.0 Hz, 1H, 8-H), 7.02 (t, *J* = 7.2 Hz, 1H, 6-H), 7.22 (t, *J* = 7.2 Hz, 1H, 7-H), 7.32 (m, 5H, 2′-H, 3′-H, 4′-H, 5′-H, 6′-H), 7.54 (d, *J* = 8.0 Hz, 1H, 5-H); ^13^C-NMR (100 MHz, CDCl_3_) δ: 42.6 (C-CH_2_), 51.8 (C-3), 67.4 (C-4), 77.3 (C-2), 116.6 (C-8), 122.4 (C-6), 122.9 (C-4a), 125.8 (C-2′, C-6′), 128.3 (C-5), 128.4 (C-4′), 128.6 (C-3′, C-5′), 129.5 (C-7), 136.9 (C-1′), 153.2 (C-8a), 168.1 (amide carbonyl); HRMS (ESI) calcd. for C_17_H_16_ClNaNO_3_ [M+Na]^+^ 340.071; found 340.073.

*(±)-2-chloro-N-[(2R*,3R*,4R*)-4-Hydroxy-2-(4-methoxyphenyl)-3,4-dihydro-2H-chromen-3-yl]acetamide* (*rac*-**22b**): White crystals, overall yield of acylation and reduction 54%, mp 179–181 °C; ^1^H-NMR (400 MHz, CDCl_3_) δ: 3.18 (bs, 1H, OH), 3.82 (s, 3H, OCH_3_), 3.83 (d, *J* = 15.2 Hz, 1H, CH_2_-H_a_), 3.95 (d, *J* = 15.2 Hz, 1H, CH_2_-H_b_), 4.72 (m, 1H, 3-H), 5.31 (bs, 1H, 4-H), 5.34 (s, 1H, 2-H), 6.83 (d, *J* = 8.8 Hz, 1H, NH), 6.91 (m, 2H, 3′-H, 5′-H), 6.98 (dd, *J* = 8.4, 1.2 Hz, 1H, 8-H), 7.05 (m, 1H, 6-H), 7.26 (m, 1H, 7-H), 7.36 (d, 2H, 2′-H, 6′-H), 7.56 (d, *J* = 7.6 Hz, 1H, 5-H); ^13^C-NMR (100 MHz, CDCl_3_) δ: 42.6 (C-CH_2_), 52.1 (C-3), 55.4 (C-OCH_3_), 68.0 (C-4), 76.9 (C-2), 114.2 (C-3′, C-5′), 116.7 (C-8), 122.5 (C-6), 123.0 (C-4a), 127.1 (C-2′, C-6′), 128.4 (C-5), 128.8 (C-1′), 129.6 (C-7), 153.3 (C-8a), 159.7 (C-4′), 168.4 (amide carbonyl); HRMS (ESI) calcd. for C_18_H_18_NaNO_4_ [M+Na]^+^ 370.109; found 370.110.

*(±)-2-chloro-N-[(2R*,3R*,4R*)-4-Hydroxy-2-(3,4-dimethoxyphenyl)-3,4-dihydro-2H-chromen-3-yl]acetamide* (*rac*-**22c**): White crystals, overall yield of acylation and reduction 70%, mp 124–126 °C; ^1^H-NMR (400 MHz, CDCl_3_) δ: 3.57 (d, *J* = 5.2 Hz, 1H, OH), 3.81 (d, *J* = 15.2 Hz, 1H, CH_2_-H_a_), 3.88 (d, 6H, 2xOCH_3_), 3.94 (d, *J* = 15.2 Hz, 1H, CH_2_-H_b_), 4.71 (m, 1H, 3-H), 5.29 (m, 2H, 2-H, 4-H), 6.86 (m, 2H, NH, 5′-H), 6.98 (m, 3H, 8-H, 2′-H, 6′-H), 7.05 (m, 1H, 6-H), 7.24 (m, 1H, 7-H), 7.55 (d, *J* = 7.6 Hz, 1H, 5-H); ^13^C-NMR (100 MHz, CDCl_3_) δ: 42.7 (C-CH_2_), 52.0 (C-3), 56.0 (C-OCH_3_), 56.1 (C-OCH_3_), 67.7 (C-4), 77.0 (C-2), 108.9 (C-5′), 111.2 (C-2′), 116.7 (C-6′), 118.2 (C-8), 122.5 (C-6), 123.0 (C-4a), 128.3 (C-5), 129.3 (C-1′), 129.5 (C-7), 149.0 (C-4′), 149.1 (C-3′), 153.2 (C-8a), 168.1 (amide carbonyl); HRMS (ESI) calcd. for C_19_H_20_ClNaNO_5_ [M+Na]^+^ 400.092; found 400.094.

*(±)-2-chloro-N-[(2R*,3R*,4R*)-4-Hydroxy-2-(3,5-dimethoxyphenyl)-3,4-dihydro-2H-chromen-3-yl]acetamide* (*rac*-**22d**): White crystals, overall yield of acylation and reduction 59%, mp 156–158 °C; ^1^H-NMR (360 MHz, CDCl_3_) δ: 3.61 (d, *J* = 5.0 Hz, 1H, OH), 3.78 (m, 8 H, CH_2_, 2xOCH_3_), 4.72 (dd, *J* = 7.9, 5.0 Hz, 1H, 3-H), 5.30 (s, 2H, 2-H, 4-H), 6.42 (t, *J* = 2.4 Hz, 1H, 4′-H), 6.60 (d, 2H, 2′-H, 6′-H), 6.87 (d, *J* = 9.0 Hz, 1H, NH), 6.99 (d, *J* = 7.9 Hz, 1H, 8-H), 7.04 (t, *J* = 7.6 Hz, 1H, 6-H), 7.24 (m, 1H, 7-H), 7.55 (d, *J* = 7.6 Hz, 1H, 5-H); ^13^C-NMR (90 MHz, CDCl_3_) δ: 42.7 (C-CH_2_), 52.2 (C-3), 55.5 (2xC-OCH_3_), 67.9 (C-4), 77.1 (C-2), 100.4 (C-4′), 103.9 (C-2′, C-6′), 116.6 (C-8), 122.5 (C-6), 123.1 (C-4a), 128.4 (C-5), 129.5 (C-7), 139.2 (C-1′), 153.1 (C-8a), 161.2 (C-3′, C-5′), 168.3 (amide carbonyl); HRMS (ESI) calcd. for C_19_H_20_ClNaNO_5_ [M+Na]^+^ 400.092; found 400.094.

*(±)-2-chloro-N-[(2R*,3R*,4R*)-4-Hydroxy-2-(3,4,5-trimethoxyphenyl)-3,4-dihydro-2H-chromen-3-yl]acetamide* (*rac*-**22e**): White crystals, overall yield of acylation and reduction 66%, mp 76–78 °C; ^1^H-NMR (400 MHz, CDCl_3_) δ: 3.77 (m, 12H, CH_2_, OH, 3 × OCH_3_), 4.70 (dd, *J* = 8.8, 4.8 Hz, 1H, 3-H), 5.27 (s, 2H, 2-H, 4-H), 6.66 (s, 2H, 2′-H, 6′-H), 6.84 (d, *J* = 9.2 Hz, 1H, NH), 6.98 (d, *J* = 8.4 Hz, 1H, 8-H), 7.02 (t, *J* = 7.6 Hz, 1H, 6-H), 7.22 (t, *J* = 7.6 Hz, 1H, 7-H), 7.52 (d, *J* = 7.6 Hz, 1H, 5-H); ^13^C-NMR (100 MHz, CDCl_3_) δ: 42.5 (C-CH_2_), 51.6 (C-3), 56.1 (2 × C-OCH_3_), 60.7 (C-OCH_3_), 67.3 (C-4), 77.1 (C-2), 102.8 (C-2′, C-6′), 116.4 (C-8), 122.3 (C-6), 122.8 (C-4a), 128.1 (C-5), 129.3 (C-7), 132.2 (C-1′), 137.7 (C-4′), 152.9 (C-8a), 153.2 (C-3′, C-5′), 167.9 (amide carbonyl); HRMS (ESI) calcd. for C_20_H_22_ClNaNO_6_ [M+Na]^+^ 430.103; found 430.104.

*(±)-2-chloro-N-[(2R*,3R*,4R*)-4-Hydroxy-2-(naphthalen-2-yl)-3,4-dihydro-2H-chromen-3-yl]acetamide* (*rac*-**22g**): White crystals, overall yield of acylation and reduction 68%, mp 69–71 °C; ^1^H-NMR (360 MHz, acetone-*d*_6_) δ: 3.82 (q, 2H, CH_2_), 4.59 (d, *J* = 7.9 Hz, 1H, OH), 4.94 (m, 1H, 3-H), 5.40 (t, *J* = 5.4 Hz, 1H, 4-H), 5.71 (s, 1H, 2-H), 6.95 (d, *J* = 8.3 Hz, 1H, 8-H), 7.01 (m, 1H, 6-H), 7.21 (m, 2H, 7-H, NH), 7.48 (m, 2H, 6′-H, 7′-H), 7.58 (d, *J* = 7.6 Hz, 1H, 5-H), 7.69 (dd, *J* = 8.6, 1.4 Hz, 1H, 3′-H), 7.9 (m, 3H, 4′-H, 5′-H, 8′-H), 8.09 (s, 1H, 1′-H); ^13^C-NMR (90 MHz, acetone-*d*_6_) δ: 43.3 (C-CH_2_), 52.0 (C-3), 67.3 (C-4), 78.8 (C-2), 117.0 (C-8), 122.3 (C-6), 125.2 (C-8′), 125.3 (C-4a), 126.0 (C-1′), 126.8 (C-3′), 126.9 (C-5), 128.5 (C-6′, C-7′), 128.9 (C-4′), 129.1 (C-5′), 129.6 (C-7), 134.0 (C-4a’, C-8a’), 136.6 (C-2′), 154.6 (C-8a), 167.4 (amide carbonyl); HRMS (ESI) calcd. for C_21_H_18_ClNaNO_3_ [M+Na]^+^ 390.087; found 390.087.

### 2.6. General Procedure for the Synthesis of 1,4-oxazin-3-one Derivatives rac-***20a***-***g*** and rac-***23a***-***e***, ***g***

The flavan-4-ol derivative *rac*-**19g** or *rac*-**22g** (0.629 mmol) was dissolved in anhydrous THF (10 mL) under inert atmosphere. To the stirred solution, 60% dispersion of NaH (0.755 mmol) was added at room temperature. The reaction was quenched after 15 min with the addition of water. The pH was adjusted to about 5 with 10% HCl solution and then the mixture was extracted with CH_2_Cl_2_. The organic phases dried over MgSO_4_ and concentrated under reduced pressure. Column chromatography using CHCl_3_ as eluent provided the pure product.

*(±)-(4aS*,5R*,10bR*)-5-phenyl-4,4a,5,10b-tetrahydrochromeno [4,3-b][1,4]oxazin-3(2H)-one* (*rac*-**20a**): White crystals, yield 88%, mp 255–257 °C; ^1^H-NMR (400 MHz, CDCl_3_) δ: 3.84 (t, *J* = 10.0 Hz, 1H, 4a-H), 4.40 (q, 2H, 2-H), 4.83 (d, *J* = 9.2 Hz, 1H, 10b-H), 4.98 (d, *J* = 10.4 Hz, 1H, 5-H), 5.45 (bs, 1H, NH), 6.90 (d, *J* = 8.0 Hz, 1H, 7-H), 7.02 (t, *J* = 7.6 Hz, 1H, 9-H), 7.24 (t, *J* = 7.6 Hz, 1H, 8-H), 7.44 (m, 6 H, 10-H, 2′-H, 3′-H, 4′-H, 5′-H, 6′-H); ^13^C-NMR (100 MHz, CDCl_3_) δ: 55.6 (C-4a), 68.4 (C-2), 73.8 (C-10b), 79.4 (C-5), 116.5 (C-7), 120.1 (C-10a), 121.6 (C-9), 125.6 (C-10), 127.7 (C-2′, C-6′), 129.6 (C-3′, C-5′), 130.0 (C-8), 130.2 (C-4′), 134.7 (C-1′), 153.3 (C-6a), 168.9 (C-3); HRMS (ESI) calcd. for C_17_H_15_NO_3_ [M + H]^+^ 282.113; found 282.115. 

(4a*S*,5*R*,10b*R*)-**20a**: *t*_R_ = 4.52 min on Chiralpak IA column (hexane/2-propanol 80:20), HPLC-ECD {λ [nm] (ϕ)}: 282sh (−3.79), 274 (−3.54), 228 (−11.27), 217 (4.28).

(4a*R*,5*S*,10b*S*)-**20a**: *t*_R_ = 5.30 min on Chiralpak IA column (hexane/2-propanol 80:20), HPLC-ECD {λ [nm] (ϕ)}: 282sh (4.70), 274 (5.26), 228 (14.69), 217 (−5.67).

*(±)-(4aS*,5SR*,10bR*)-5-(4-methoxyphenyl)-4,4a,5,10b-tetrahydrochromeno[4,3-b][1,4]oxazin-3(2H)-one* (*rac*-**20b**): White crystals, yield 89%, mp 224–226 °C; ^1^H-NMR (400 MHz, DMSO-*d*_6_) δ: 3.79 (s, 3H, OCH_3_), 3.90 (t, *J* = 10.0 Hz, 1H, 4-H), 4.33 (s, 2H, 2-H), 4.95 (d, *J* = 9.2 Hz, 1H, 10b-H), 5.11 (d, *J* = 10.0 Hz, 1H, 5-H), 6.80 (d, *J* = 8.0 Hz, 1H, 7-H), 6.98 (m, 4 H, 9-H, 3′-H, 5′-H, NH), 7.20 (t, *J* = 8.0 Hz, 1H, 8-H), 7.37 (d, *J* = 7.6 Hz, 1H, 10-H), 7.41 (d, 2H, 2′-H, 6′-H); ^13^C-NMR (100 MHz, DMSO-*d*_6_) δ: 53.9 (C-OCH_3_), 55.2 (C-4a), 67.8 (C-2), 72.7 (C-10b), 78.6 (C-5), 114.2 (C-3′, C-5′), 115.7 (C-7), 120.7 (C-9), 120.9 (C-10a), 125.2 (C-10), 127.3 (C-1′), 129.3 (C-8), 130.0 (C-2′, C-6′), 153.2 (C-6a), 160.1 (C-4′), 168.3 (C-3); HRMS (ESI) calcd. for C_18_H_17_NO_4_ [M + H]^+^ 312.123; found 312.124.

(4a*S*,5*R*,10b*R*)-**20b**: *t*_R_ = 5.82 min on Chiralpak IA column (hexane/2-propanol 80:20), HPLC-ECD {λ [nm] (ϕ)}: 273 (−3.94), 233 (7.54), 215 (−1.82).

(4a*R*,5*S*,10b*S*)-**20b**: *t*_R_ = 6.85 min on Chiralpak IA column (hexane/2-propanol 80:20), HPLC-ECD {λ [nm] (ϕ)}: 273 (3.70), 233 (−9.20), 215 (2.98).

*(±)-(4aS*,5R*,10bR*)-5-(3,4-dimethoxyphenyl)-4,4a,5,10b-tetrahydrochromeno[4,3-b][1,4]oxazin-3(2H)-one* (*rac*-**20c**): White crystals, yield 90%, mp 255–257 °C; ^1^H-NMR (400 MHz, DMSO-*d*_6_) δ: 3.78 (d, 6H, 2 × OCH_3_), 3.97 (t, *J* = 9.6 Hz, 1H, 4a-H), 4.33 (s, 2H, 2-H_a_, 2-H_b_), 4.95 (d, *J* = 9.2 Hz, 1H, 10b-H), 5.10 (d, *J* = 10.4 Hz, 1H, 5-H), 6.81 (d, *J* = 8.0 Hz, 1H, 7-H), 6.97 (m, 4 H, 9-H, 2′-H, 5′-H, 6′-H), 7.21 (t, *J* = 8.0 Hz, 1H, 8-H), 7.37 (d, *J* = 7.6 Hz, 1H, 10-H); ^13^C-NMR (100 MHz, DMSO-*d*_6_) δ: 53.8 (C-4a), 55.5 (2 × C-OCH_3_), 67.7 (C-2), 72.6 (C-10b), 78.9 (C-5), 111.8 (C-5′), 111.8 (C-2′), 115.7 (C-6′), 120.5 (C-7), 120.9 (C-10a), 121.3 (C-9), 125.1 (C-10), 127.4 (C-1′), 129.2 (C-8), 149.1 (C-4′), 150.2 (C-3′), 153.2 (C-6a), 168.1 (C-3); HRMS (ESI) calcd. for C_19_H_19_NaNO_5_ [M+Na]^+^ 364.116; found 364.113.

(4a*R*,5*S*,10b*S*)-**20c**: *t*_R_ = 7.86 min on Chiralpak IA column (hexane/2-propanol 80:20), HPLC-ECD {λ [nm] (ϕ)}: 287 (−0.89), 282sh (1.66), 274 (1.92), 235 (6.17), 224sh (4.71).

(4a*S*,5*R*,10b*R*)-**20c**: *t*_R_ = 9.50 min on Chiralpak IA column (hexane/2-propanol 80:20), HPLC-ECD {λ [nm] (ϕ)}:287 (0.35), 282sh (−1.61), 274 (−1.93), 235 (−6.62), 224sh (−5.44).

*(±)-(4aS*,5R*,10bR*)-5-(3,5-dimethoxyphenyl)-4,4a,5,10b-tetrahydrochromeno[4,3-b][1,4]oxazin-3(2H)-one* (*rac*-**20d**): White crystals, yield 80%, mp 189–190 °C; ^1^H-NMR (400 MHz, CDCl_3_) δ: 3.77 (m, 7H, 4a-H, 2 × OCH_3_), 4.38 (d, *J* = 17.2 Hz, 1H, 2-H_a_), 4.45 (d, *J* = 17.2 Hz, 1H, 2-H_b_), 4.78 (d, *J* = 9.2 Hz, 1H, 10b-H), 4.88 (d, *J* = 10.4 Hz, 1H, 5-H), 5.59 (s, 1H, NH), 6.50 (t, *J* = 2.4 Hz, 1H, 4′-H), 6.56 (d, 2H, 2′-H, 6′-H), 6.91 (d, *J* = 8.0 Hz, 1H, 7-H), 7.01 (t, *J* = 7.6 Hz, 1H, 9-H), 7.23 (m, 1H, 8-H), 7.44 (d, *J* = 7.6 Hz, 1H, 10-H); ^13^C-NMR (100 MHz, CDCl_3_) δ: 55.1 (C-4a), 55.2 (2 × C-OCH_3_), 67.9 (C-2), 73.3 (C-10b), 79.0 (C-5), 101.2 (C-4′), 105.1 (C-2′, C-6′), 116.1 (C-7), 119.8 (C-10a), 121.2 (C-9), 125.3 (C-10), 129.6 (C-8), 136.5 (C-1′), 152.9 (C-6a), 161.3 (C-3′, C-5′), 168.5 (C-3); HRMS (ESI) calcd. for C_19_H_19_NaNO_5_ [M+Na]^+^ 364.116; found 364.113.

(4a*R*,5*S*,10b*S*)-**20d**: *t*_R_ = 6.72 min on Chiralpak IA column (hexane/2-propanol 80:20), HPLC-ECD {λ [nm] (ϕ)}: 283 (4.08), 240sh (−1.16), 233 (−3.03), 222 (1.82), 213 (−10.39).

(4a*S*,5*R*,10b*R*)-**20d**: *t*_R_ = 7.07 min on Chiralpak IA column (hexane/2-propanol 80:20), HPLC-ECD {λ [nm] (ϕ)}:283 (−4.52), 240sh (1.28), 233 (2.95), 222 (−3.55), 213 (6.44).

*(±)-(4aS*,5R*,10bR*)-5-(3,4,5-trimethoxyphenyl)-4,4a,5,10b-tetrahydrochromeno[4,3-b][1,4]oxazin-3(2H)-one* (*rac*-**20e**): White crystals, yield 83%, mp 146–148 °C; ^1^H-NMR (400 MHz, CDCl_3_) δ: 3.84 (m, 10 H, 4a-H, 3 × OCH_3_), 4.41 (d, *J* = 16.8 Hz, 1H, 2-H_a_), 4.48 (d, *J* = 16.8 Hz, 1H, 2-H_b_), 4.82 (d, *J* = 9.2 Hz, 1H, 10b-H), 4.91 (d, *J* = 10.4 Hz, 1H, 5-H), 5.63 (s, 1H, NH), 6.65 (s, 2H, 2′-H, 6′-H), 6.92 (d, *J* = 8.4 Hz, 1H, 7-H), 7.03 (t, *J* = 7.2 Hz, 1H, 9-H), 7.26 (t, *J* = 7.2 Hz, 1H, 8-H), 7.46 (d, *J* = 7.6 Hz, 1H, 10-H); ^13^C-NMR (101 MHz, CDCl_3_) δ: 55.2 (C-4a), 56.0 (2 × C-OCH_3_), 60.6 (C-OCH_3_), 68.0 (C-2), 73.4 (C-10b), 79.3 (C-5), 104.2 (C-2′, C-6′), 116.2 (C-7), 119.8 (C-10a), 121.3 (C-9), 125.3 (C-10), 129.6 (C-1′), 129.6 (C-8), 138.9 (C-4′), 152.9 (C-6a), 153.8 (C-3′, C-5′), 168.7 (C-3); HRMS (ESI) calcd. for C_20_H_21_NaNO_6_ [M+Na]^+^ 394.126; found 394.124.

(4a*R*,5*S*,10b*S*)-**20e**: *t*_R_ = 10.18 min on Chiralpak IA column (hexane/2-propanol 80:20), HPLC-ECD {λ [nm] (ϕ)}: 282 (1.51), 275sh (1.37), 245 (−1.52), 233sh (1.49), 225 (2.54), 216 (−0.36).

(4a*S*,5*R*,10b*R*)-**20e**: *t*_R_ = 15.71 min on Chiralpak IA column (hexane/2-propanol 80:20), HPLC-ECD {λ [nm] (ϕ)}:282 (−1.44), 275sh (−1.31), 245 (0.65), 233sh (−1.82), 225 (−2.09), 216 (0.52).

*(±)-(4aS*,5R*,10bR*)-5-(naphthalen-1-yl)-4,4a,5,10b-tetrahydrochromeno[4,3-b][1,4]oxazin-3(2H)-one* (*rac*-**20f**): White crystals, yield 83%, mp 254–255 °C; ^1^H-NMR (360 MHz, DMSO-*d*_6_) δ: 4.20 (t, *J* = 9.0 Hz, 1H, 4a-H), 4.37 (s, 2H, 2-H), 5.19 (d, *J* = 8.3 Hz, 1H, 10b-H), 6.12 (bs, 1H, 5-H), 6.83 (d, *J* = 7.9 Hz, 1H, 7-H), 7.01 (t, *J* = 7.2 Hz, 1H, 9-H), 7.24 (m, 2H, 8-H, NH), 7.44 (d, *J* = 7.2 Hz, 1H, 10-H), 7.59 (m, 3H, 2′-H, 3′-H, 7′-H), 7.77 (m, 1H, 6′-H), 8.01 (m, 2H, 4′-H, 5′-H), 8.28 (m, 1H, 8′-H); ^13^C-NMR (91 MHz, DMSO-*d*_6_) δ: 53.8 (C-4a), 67.8 (C-2), 72.7 (C-10b), 82.4 (C-5), 115.7 (C-7), 120.7 (C-9), 121.2 (C-10a), 123.3 (C-8′), 125.2 (C-10), 125.7 (C-2′, C-3′), 126.3 (C-6′, C-7′), 128.8 (C-5′), 129.3 (C-4′), 129.6 (C-8), 131.0 (C-4a’), 131.9 (C-8a’), 133.7 (C-1′), 153.2 (C-6a), 168.3 (C-3); HRMS (ESI) calcd. for C_21_H_17_NaNO_3_ [M+Na]^+^ 354.110; found 354.112.

(4a*S*,5*R*,10b*R*)-**20f**: *t*_R_ = 5.71 min on Chiralpak IA column (hexane/2-propanol 80:20), HPLC-ECD {λ [nm] (ϕ)}: 293 (−1.64), 280 (−1.36), 270 (−1.79), 225 (34.72), 211 (−8.76).

(4a*R*,5*S*,10b*S*)-**20f**: *t*_R_ = 7.06 min on Chiralpak IA column (hexane/2-propanol 80:20), HPLC-ECD {λ [nm] (ϕ)}: 293 (1.13), 280 (1.01), 270 (0.98), 225 (−34.55), 211 (13.28).

*(±)-(4aS*,5R*,10bR*)-5-(naphthalen-2-yl)-4,4a,5,10b-tetrahydrochromeno[4,3-b][1,4]oxazin-3(2H)-one* (*rac*-**20g**): White crystals, yield 91%, mp 273–275 °C; ^1^H-NMR (400 MHz, DMSO-*d*_6_) δ: 4.04 (t, *J* = 10.0 Hz, 1H, 4a-H), 4.31 (m, 2H, 2-H), 5.03 (d, *J* = 9.6 Hz, 1H, 10b-H), 5.35 (d, *J* = 10.4 Hz, 1H, 5-H), 6.85 (d, *J* = 8.0 Hz, 1H, 7-H), 6.99 (m, 1H, 9-H), 7.23 (m, 1H, 8-H), 7.37 (s, 1H, NH), 7.41 (d, *J* = 7.6 Hz, 1H, 10-H), 7.56 (m, 2H, 6′-H, 7′-H), 7.62 (dd, *J* = 8.4, 1.2 Hz, 1H, 3′-H), 7.96 (m, 3H, 4′-H, 5′-H, 8′-H), 8.04 (s, 1H, 1′-H); ^13^C-NMR (100 MHz, DMSO-*d*_6_) δ: 54.0 (C-4a), 67.8 (C-2), 72.6 (C-10b), 79.3 (C-5), 115.7 (C-7), 120.8 (C-8), 121.0 (C-10a), 125.2 (C-1′, C-8′), 126.2 (C-3′), 126.5 (C-7′), 127.6 (C-9), 128.2 (C-6′), 128.6 (C-5′), 128.8 (C-4′), 129.3 (C-10), 132.9 (C-4a’, C-8a’), 133.5 (C-2′), 153.2 (C-6a), 168.3 (C-3); HRMS (ESI) calcd. for C_21_H_17_NaNO_3_ [M+Na]^+^ 354.110; found 354.111.

(4a*S*,5*R*,10b*R*)-**20g**: *t*_R_ = 21.46 min on Chiralpak IA column (hexane/2-propanol 90:10), HPLC-ECD {λ [nm] (ϕ)}: 283 (−1.06), 240sh (1.27), 228 (9.56), 216 (−3.76).

(4a*R*,5*S*,10b*S*)-**20g**: *t*_R_ = 22.63 min on Chiralpak IA column (hexane/2-propanol 90:10), HPLC-ECD {λ [nm] (ϕ)}: 283 (1.35), 240sh (−0.81), 228 (−10.02), 216 (3.61).

*(±)-(4aR*,5R*,10bR*)-5-phenyl-4,4a,5,10b-tetrahydrochromeno[4,3-b][1,4]oxazin-3(2H)-one* (rac-**23a**): White crystals, yield 93%, mp 151–153 °C; ^1^H-NMR (400 MHz, CDCl_3_) δ: 3.83 (d, *J* = 16.8 Hz, 1H, 2-H_a_), 4.02 (d, *J* = 16.8 Hz, 1H, 2-H_b_) 4.20 (d, *J* = 5.6 Hz, 1H, 4a-H), 5.29 (s, 1H, 5-H), 5.46 (d, *J* = 5.6 Hz, 1H, 10b-H), 5.70 (bs, 1H, NH), 6.98 (d, *J* = 8.0 Hz, 1H, 7-H), 7.06 (m, 1H, 9-H), 7.28 (m, 1H, 8-H), 7.40 (m, 1H, 4′-H), 7.45 (m, 5H, 5-H, 2′-H, 3′-H, 5′-H, 6′-H); ^13^C-NMR (100 MHz, CDCl_3_) δ: 52.4 (C-4a), 61.9 (C-2), 67.7 (C-10b), 76.4 (C-5), 117.5 (C-10a), 117.8 (C-7), 122.8 (C-9), 126.0 (C-2′, C-6′), 128.2 (C-5), 129.3 (C-4′), 129.7 (C-3′, C-5′), 130.5 (C-8), 135.8 (C-1′), 155.4 (C-6a), 169.1 (C-3); HRMS (ESI) calcd. for C_17_H_15_NO_3_ [M + H]^+^ 282.113; found 282.115.

(4a*S*,5*S*,10b*S*)-**23a**: *t*_R_ = 6.38 min on Chiralpak IA column (hexane/2-propanol 80:20), HPLC-ECD {λ [nm] (ϕ)}: 283sh (10.79), 276 (11.06), 237 (3.05), 229 (−13.14), 214 (32.68).

(4a*R*,5*R*,10b*R*)-**23a**: *t*_R_ = 7.92 min on Chiralpak IA column (hexane/2-propanol 80:20), HPLC-ECD {λ [nm] (ϕ)}: 283sh (−7.27), 276 (−7.62), 237 (−1.40), 229 (11.40), 214 (−25.80).

*(±)-(4aR*,5R*,10bR*)-5-(4-methoxyphenyl)-4,4a,5,10b-tetrahydrochromeno[4,3-b][1,4]oxazin-3(2H)-one* (*rac*-**23b**): White crystals, yield 74%, mp 167–169 °C; ^1^H-NMR (400 MHz, CDCl_3_) δ: 3.79 (m, 4 H, OCH_3_, 2-H_a_), 3.98 (d, *J* = 16.8 Hz, 1H, 2-H_b_), 4.12 (d, *J* = 5.6 Hz, 1H, 4a-H), 5.20 (s, 1H, 10b-H), 5.40 (d, *J* = 5.6 Hz, 1H, 5-H), 5.81 (s, 1H, NH), 6.92 (d, 2H, 3′-H, 5′-H), 6.96 (d, *J* = 8.4 Hz, 1H, 7-H), 7.04 (t, *J* = 7.6 Hz, 1H, 9-H), 7.24 (t, *J* = 7.2 Hz, 1H, 8-H), 7.38 (d, 2H, 2′-H, 6′-H), 7.43 (d, *J* = 7.6 Hz, 1H, 10-H); ^13^C-NMR (100 MHz, CDCl_3_) δ: 51.9 (C-4a), 55.4 (C-OCH_3_), 61.5 (C-2), 67.2 (C-10b), 75.7 (C-5), 114.6 (C-3′, C-5′), 117.1 (C-10a), 117.3 (C-7), 122.3 (C-9), 126.8 (C-2′, C-6′), 127.2 (C-1′), 127.9 (C-10), 130.0 (C-8), 155.1 (C-6a), 159.8 (C-4′), 168.6 (C-3); HRMS (ESI) calcd. for C_18_H_17_NO_4_ [M + H]^+^ 312.123; found 312.124.

(4a*R*,5*R*,10b*R*)-**23b**: *t*_R_ = 8.78 min on Chiralpak IA column (hexane/2-propanol 80:20), HPLC-ECD {λ [nm] (ϕ)}: 283 (−7.85), 275 (−7.80), 231sh (−18.86), 225 (−19.61).

(4a*S*,5*S*,10b*S*)-**23b**: *t*_R_ = 9.75 min on Chiralpak IA column (hexane/2-propanol 80:20), HPLC-ECD {λ [nm] (ϕ)}: 283 (8.34), 275 (7.98), 231sh (18.54), 225 (19.96).

*(±)-(4aR*,5R*,10bR*)-5-(3,4-dimethoxyphenyl)-4,4a,5,10b-tetrahydrochromeno[4,3-b][1,4]oxazin-3(2H)-one* (*rac*-**23c**): White crystals, yield 69%, mp 139–140 °C; ^1^H-NMR (400 MHz, CDCl_3_) δ: 3.83 (d, *J* = 17.2 Hz, 1H, 2-H_a_), 3.92 (d, 6H, 2 × OCH_3_), 4.04 (d, *J* = 17.2 Hz, 1H, 2-H_b_), 4.18 (d, *J* = 5.2 Hz, 1H, 4a-H), 5.24 (s, 1H, 10b-H), 5.45 (d, *J* = 5.2 Hz, 1H, 5-H), 5.73 (s, 1H, NH), 6.94 (m, 3H, 7-H, 2′-H, 5′-H), 7.05 (m, 2H, 9-H, 6′-H), 7.27 (m, 1H, 8-H), 7.46 (d, *J* = 7.2 Hz, 1H, 10-H); ^13^C-NMR (100 MHz, CDCl_3_) δ: 52.2 (C-4a), 56.2 (2 × C-OCH_3_), 61.6 (C-2), 67.3 (C-10b), 75.9 (C-5), 108.6 (C-5′), 111.8 (C-2′), 117.2 (C-10a), 117.5 (C-6′), 117.9 (C-7), 122.5 (C-9), 127.8 (C-1′), 127.9 (C-10), 130.2 (C-8), 149.4 (C-4′), 149.8 (C-3′), 155.1 (C-6a), 168.7 (C-3); HRMS (ESI) calcd. for C_19_H_19_NaNO_5_ [M+Na]^+^ 364.116; found 364.113.

(4a*R*,5*R*,10b*R)*-**23c**: *t*_R_ = 12.43 min on Chiralpak IA column (hexane/2-propanol 80:20), HPLC-ECD {λ [nm] (ϕ)}: 283 (−10.06), 276sh (−9.22), 235 (−10.58), 224sh (−2.55), 211 (−18.11).

(4a*S*,5*S*,10b*S*)-**23c**: *t*_R_ = 15.97 min on Chiralpak IA column (hexane/2-propanol 80:20), HPLC-ECD {λ [nm] (ϕ)}: 283 (10.17), 276sh (9.61), 235 (11.92), 224sh (4.81), 211 (11.93).

*(±)-(4aR*,5R*,10bR*)-5-(3,5-dimethoxyphenyl)-4,4a,5,10b-tetrahydrochromeno[4,3-b][1,4]oxazin-3(2H)-one* (*rac*-**23d**): White crystals, yield 96%, mp 136–138 °C; ^1^H-NMR (400 MHz, CDCl_3_) δ: 3.83 (m, 7H, 2-H_a_, 2 × OCH_3_), 4.02 (d, *J* = 16.8 Hz, 1H, 2-H_b_), 4.18 (d, *J* = 5.6 Hz, 1H, 4a-H), 5.20 (s, 1H, 10b-H), 5.44 (d, *J* = 5.6 Hz, 1H, 5-H), 5.73 (s, 1H, NH), 6.47 (s, 1H, 4′-H), 6.64 (s, 2H, 2′-H, 6′-H), 6.98 (d, *J* = 8.0 Hz, 1H, 7-H), 7.05 (t, *J* = 7.6 Hz, 1H, 9-H), 7.26 (t, *J* = 8.0 Hz, 1H, 8-H), 7.44 (d, *J* = 7.6 Hz, 1H, 10-H); ^13^C-NMR (100 MHz, CDCl_3_) δ: 52.2 (C-4a), 55.6 (2 × C-OCH_3_), 61.6 (C-2), 67.3 (C-10b), 76.0 (C-5), 100.4 (C-4′), 103.7 (C-2′, C-6′), 117.2 (C-10a), 117.5 (C-7), 122.5 (C-8), 127.9 (C-9), 130.2 (C-10), 137.8 (C-1′), 154.9 (C-6a), 161.7 (C-3′, C-5′), 168.6 (C-3); HRMS (ESI) calcd. for C_19_H_19_NaNO_5_ [M+Na]^+^ 364.116; found 364.113.

(4a*S*,5*S*,10b*S*)-**23d**: *t*_R_ = 10.35 min on Chiralpak IA column (hexane/2-propanol 80:20), HPLC-ECD {λ [nm] (ϕ)}: 283 (2.62), 276sh (2.71), 241 (0.83), 231 (−3.03), 222sh (4.32), 210 (15.46).

(4a*R*,5*R*,10b*R*)-**23d***: t*_R_ = 11.98 min on Chiralpak IA column (hexane/2-propanol 80:20), HPLC-ECD {λ [nm] (ϕ)}: 283 (−3.52), 276sh (−3.47), 241 (−1.00), 231 (2.81), 222sh (−7.12), 210 (−27.95).

*(±)-(4aR*,5R*,10bR*)-5-(3,4,5-trimethoxyphenyl)-4,4a,5,10b-tetrahydrochromeno[4,3-b][1,4]oxazin-3(2H)-one* (*rac*-**23e**): White crystals, yield 89%, mp 148–150 °C; ^1^H-NMR (400 MHz, CDCl_3_) δ: 3.83 (m, 10 H, 2-H_a_, 3 × OCH_3_), 4.04 (d, *J* = 16.8 Hz, 1H, 2-H_b_), 4.19 (d, *J* = 5.6 Hz, 1H, 4a-H), 5.22 (s, 1H, 10b-H), 5.45 (d, *J* = 5.6 Hz, 1H, 5-H), 5.73 (s, 1H, NH), 6.71 (s, 2H, 2′-H, 6′-H), 7.00 (d, *J* = 8.0 Hz, 1H, 7-H), 7.07 (t, *J* = 7.6 Hz, 1H, 9-H), 7.27 (t, *J* = 8.0 Hz, 1H, 8-H), 7.45 (d, *J* = 7.6 Hz, 1H, 10-H); ^13^C-NMR (100 MHz, CDCl_3_) δ: 52.1 (C-4a), 56.2 (2 × C-OCH_3_), 60.8 (C-OCH_3_), 61.4 (C-2), 67.1 (C-10b), 75.8 (C-5), 102.3 (C-2′, C-6′), 117.0 (C-10a), 117.3 (C-7), 122.4 (C-9), 127.7 (C-10), 130.0 (C-8), 130.8 (C-1′), 138.0 (C-4′), 153.8 (C-3′, C5′), 154.7 (C-6a), 168.5 (C-3); HRMS (ESI) calcd. for C_20_H_21_NaNO_6_ [M+Na]^+^ 394.126; found 394.124.

(4a*R*,5*R*,10b*R*)-**23e***: t*_R_ = 12.23 min on Chiralpak IA column (hexane/2-propanol 80:20), HPLC-ECD {λ [nm] (ϕ)}: 283 (−4.07), 276sh (−4.33), 240 (−6.43), 228 (1.76), 212 (−21.15).

(4a*S*,5*S*,10b*S*)-**23e**: *t*_R_ = 16.85 min on Chiralpak IA column (hexane/2-propanol 80:20), HPLC-ECD {λ [nm] (ϕ)}: 283 (3.63), 276sh (3.23), 240 (5.25), 228 (−0.84), 212 (11.85).

*(±)-(4aR*,5R*,10bR*)-5-naphthalen-2-yl-4,4a,5,10b-tetrahydrochromeno[4,3-b][1,4]oxazin-3(2H)-one* (*rac*-**23g**): White crystals, yield 79%, mp 193–195 °C; ^1^H-NMR (360 MHz, CDCl_3_) δ: 3.83 (d, *J* = 16.9 Hz, 1H, 2-H_a_), 4.01 (d, *J* = 16.9 Hz, 1H, 2-H_b_), 4.27 (d, *J* = 5.4 Hz, 1H, 4a-H), 5.41 (s, 1H, 5-H), 5.49 (d, *J* = 5.4 Hz, 1H, 10b-H), 5.69 (s, 1H, NH), 7.05 (m, 2H, 7-H, 9-H), 7.28 (t, *J* = 7.2 Hz,1H, 8-H), 7.44 (m, 4 H, 3′-H, 6′-H, 7′-H, 10-H), 7.86 (m, 3H, 4′-H, 5′-H, 8′-H), 8.08 (s, 1H, 1′-H); ^13^C-NMR (90 MHz, CDCl_3_) δ: 51.8 (C-4a), 61.6 (C-2), 67.4 (C-10b), 76.2 (C-5), 117.2 (C-10a), 117.5 (C-7), 122.5 (C-8), 122.8 (C-1′), 125.2 (C-8′), 126.9 (C-3′), 127.0 (C-7′), 127.9 (C-9), 128.0 (C-6′), 128.3 (C-5′), 129.3 (C-4′), 130.2 (C-10), 132.6 (C-4a’), 133.2 (C-8a’), 133.3 (C-2′), 155.1 (C-6a), 168.6 (C-3); HRMS (ESI) calcd. for C_21_H_17_NaNO_3_ [M+Na]^+^ 354.110; found 354.111.

(4a*R*,5*R*,10b*R*)-**23g***: t*_R_ = 18.43 min on Chiralpak IA column (hexane/2-propanol 80:20), HPLC-ECD {λ [nm] (ϕ)}: 283 (−4.33), 234sh (−2.97), 219 (−30.92), 203 (21.46).

(4a*S*,5*S*,10b*S*)-**23e**: *t*_R_ = 19.30 min on Chiralpak IA column (hexane/2-propanol 80:20), HPLC-ECD {λ [nm] (ϕ)}: 283 (3.93), 234sh (2.35), 219 (28.64), 203 (−21.27).

### 2.7. General Procedure for the Synthesis of Condensed Morpholinee Derivatives [rac-(4aR*,5S*,10aS*)-***2a***-***g***, rac-(4aR*,5S*,10aR*)-***2a***-***e***, ***g***, rac-(4aR*,5R*,10aR*)-***2a***-***e***, ***g***]

Under inert atmosphere, the condensed 1,4-oxazinone derivatives *rac*-**20a-g** or *rac*-**23a-e**, **g** (0.359 mmol) were dissolved in anhydrous dioxane (5 mL) and after heating the reaction mixture to 90 °C, 2 M LiAlH_4_ solution in THF (216 µL) was added. The reaction was quenched after 15 min with the addition of ethyl acetate and water. The organic phase was collected and dried over MgSO_4_, then it was concentrated under reduced pressure. Procedure A: The product was obtained as the hydrochloride salt after stirring for 2 h at room temperature in a mixture of ethyl acetate (5 mL) and 3 N HCl solution (124 µL). Procedure B: The product was isolated as the amine base after column chromatography using CHCl_3_ as eluent.

*(±)-(4aR*,5S*,10bS*)-5-phenyl-2,3,4,4a,5,10b-hexahydrochromeno[4,3-b][1,4]oxazine hydrochloride* [*rac-(4a*R**,5*S**,10a*S**)-***2a**]: White solid, yield 72%, mp > 300 °C; ^1^H-NMR (400 MHz, *DMSO-*d_6_) δ: 3.15 (bs, 1H, 3-Ha), 3.23 (d, *J* = 12.4 Hz, 1H, 3-Hb), 3.84 (m, 1H, 4a-H), 4.17 (d, 2H, 2-Ha, 2-Hb), 5.30 (d, *J* = 9.6 Hz, 1H, 10b-H), 5.51 (d, *J* = 10.0 Hz, 1H, 5-H), 6.87 (d, *J* = 8.4 Hz, 1H, 7-H), 7.02 (t, *J* = 7.2 Hz, 1H, 9-H), 7.26 (t, *J* = 7.2 Hz, 1H, 8-H), 7.39 (d, *J* = 7.6 Hz, 1H, 10-H), 7.49 (m, 3H, 3′-H, 4′-H, 5′-H), 7.63 (m, 2H, 2′-H, 6′-H), 8.38 (bs, 1H, NH_2_-Ha), 11.29 (bs, 1H, NH_2_-Hb); ^13^C-NMR (100 MHz, *DMSO-*d_6_) δ: 43.9 (C-3), 55.5 (C-4a), 63.4 (C-2), 71.9 (C-10b), 76.4 (C-5), 116.0 (C-7), 120.4 (C-10a), 121.2 (C-9), 125.6 (C-10), 128.5 (C-2′, C-6′), 128.9 (C-3′, C-5′), 129.6 (C-8), 129.7 (C-4′), 134.2 (C-1′), 152.7 (C-6a); HRMS (ESI) calcd. for C_17_H_17_NO_2_ [M + H]^+^ 268.1332; found 268.1137.

*(±)-(4aR*,5S*,10bR*)-5-phenyl-2,3,4,4a,5,10b-hexahydrochromeno[4,3-b][1,4]oxazine* [*rac-(4a*R**,5*S**,10a*R**)-***2a**]. Colorless oil, yield 60%, ^1^H-NMR (400 MHz, CDCl_3_) δ: 2.26 (s, 1H, NH), 2.90 (m, 3H, 2-Ha, 3-Ha, 3-Hb), 3.56 (d, *J* = 11.6 Hz, 1H, 2-Hb), 4.36 (s, 1H, 5-H), 4.89 (d, *J* = 7.6 Hz, 1H, 4a-H), 5.41 (d, *J* = 7.6 Hz, 1H, 10b-H), 6.92 (m, 2H, 7-H, 9-H), 7.26 (m, 5H, 3′-H, 4′-H, 5′-H, 8-H, 10-H), 7.48 (d, 2′-H, 6′-H); ^13^C-NMR (100 MHz, CDCl_3_) δ: 50.5 (C-3), 61.2 (C-5), 68.3 (C-2), 78.9 (C-10b), 90.1 (C-4a), 110.4 (C-7), 121.4 (C-9), 124.8 (C-10a), 126.4 (C-4′), 126.9 (C-2′, C-6′), 127.1 (C-10), 128.5 (C-3′, C-5′), 130.9 (C-8), 143.4 (C-1′), 160.1 (C-10a); HRMS (ESI) calcd. for C_17_H_17_NO_2_ [M + H]^+^ 268.1332; found 268.1132.

*(±)-(4aR*,5R*,10bR*)-5-phenyl-2,3,4,4a,5,10b-hexahydrochromeno[4,3-b][1,4]oxazine* [*rac-(4a*R**,5*R**,10a*R**) -***2a**]: White crystals, yield 20%, mp 134–136 °C; ^1^H-NMR (400 MHz, acetone-*d*_6_) δ: 2.64 (d, *J* = 12.0 Hz, 1H, 3-Ha), 2.75 (m, 1H, 3-Hb), 3.36 (4 H, 2-Ha, 2-Hb, 4a-H, NH), 5.13 (d, *J* = 4.0 Hz, 1H, 10b-H), 5.29 (s, 1H, 5-H), 6.89 (dd, *J* = 8.4, 1.2 Hz, 1H, 7-H), 6.97 (m, 1H, 9-H), 7.19 (m, 1H, 8-H), 7.23 (m, 1H, 4′-H), 7.39 (m, 3H, 3′-H, 5′-H, 10-H), 7.55 (d, 2H, 2′-H, 6′-H); ^13^C-NMR (100 MHz, acetone-*d*_6_) δ: 46.3 (C-3), 54.7 (C-4a), 61.2 (C-2), 70.5 (C-10b), 79.2 (C-5), 117.0 (C-7), 121.8 (C-9), 122.4 (C-10a), 127.1 (C-2′, C-6′), 128.3 (C-10), 128.5 (C-4′), 129.0 (C-3′, C-5′), 129.2 (C-8), 139.4 (C-1′), 156.2 (C-6a); HRMS (ESI) calcd. for C_17_H_17_NO_2_ [M + H]^+^ 268.1332; found 268.1137.

(±)-(4aR*,5S*,10bS*)-5-(4-methoxyphenyl)-2,3,4,4a,5,10b-hexahydrochromeno[4,3-b][1,4]oxazine hydrochloride [rac-(4aR*,5S*,10aS*)-**2b**]: White crystals, yield 70%, mp 245–247 °C; ^1^H-NMR (400 MHz, DMSO-*d*_6_) δ: 3.15 (bs, 1H, 3-H_a_), 3.22 (d, *J* = 12.8 Hz, 1H, 3-H_b_), 3.81 (m, 4 H, 4a-H, OCH_3_), 4.16 (d, 2H), 5.28 (d, *J* = 10.0 Hz, 1H, 10b-H), 5.45 (d, *J* = 10.4 Hz, 1H, 5-H), 6.85 (d, *J* = 8.4 Hz, 1H, 7-H), 7.00 (m, 3H, 3′-H, 5′-H, 9-H), 7.24 (m, 1H, 8-H), 7.38 (d, *J* = 7.6 Hz, 1H, 10-H), 7.54 (d, 2H, 2′-H, 6′-H), 8.33 (bs, 1H, NH_2_-H_a_), 11.25 (d, *J* = 5.6 Hz, 1H, NH_2_-H_b_); ^13^C-NMR (100 MHz, DMSO-*d*_6_) δ: 43.9 (C-3), 55.3 (C-OCH_3_), 55.6 (C-4a), 63.4 (C-2), 72.0 (C-10b), 76.0 (C-5), 114.3 (C-3′, C-5′), 116.0 (C-7), 120.4 (C-10a), 121.1 (C-9), 125.6 (C-10), 126.1 (C-1′), 129.5 (C-8), 130.0 (C-2′, C-6′), 152.8 (C-6a), 160.3 (C-4′); HRMS (ESI) calcd. for C_18_H_19_NO_3_ [M + H]^+^ 298.1438; found 298.1439.

*(±)-(4aR*,5S*,10bR*)-5-(4-methoxyphenyl)-2,3,4,4a,5,10b-hexahydrochromeno[4,3-b][1,4]oxazine* [*rac-(4a*R**,5*S**,10a*R**)-***2b**]: Colorless oil, yield: 41%, ^1^H-NMR (400 MHz, CDCl_3_) δ: 2.90 (m, 4 H, 2-H_a_, 3-H_a_, 3-H_b_, NH), 3.56 (d, *J* = 11.6 Hz, 1H, 2-H_b_), 3.80 (s, 3H, OCH_3_), 4.34 (s, 1H, 5-H), 4.88 (d, *J* = 7.6 Hz, 1H, 4a-H), 5.41 (d, *J* = 7.6 Hz, 1H, 10b-H), 6.90 (d, 2H, 3′-H, 5′-H), 6.94 (d, *J* = 8.4 Hz, 1H, 7-H), 6.97 (m, 1H, 9-H), 7.28 (m, 1H, 8-H), 7.39 (d, *J* = 7.2 Hz, 1H, 10-H), 7.43 (d, 2H, 2′-H, 6′-H); ^13^C-NMR (100 MHz, CDCl_3_) δ: 50.6 (C-3), 55.4 (C-OCH_3_), 60.9 (C-5), 68.1 (C-2), 79.0 (C-10b), 90.2 (C-4a), 110.6 (C-7), 114.1 (C-3′, C-5′), 121.5 (C-9), 124.9 (C-10a), 126.5 (C-10), 128.2 (C-2′, C-6′), 131.1 (C-8), 135.5 (C-1′), 158.8 (C-4′), 160.3 (C-6a); HRMS (ESI) calcd. for C_18_H_19_NO_3_ [M + H]^+^ 298.1438; found 298.1440.

*(±)-(4aR*,5R*,10bR*)-5-(4-methoxyphenyl)-2,3,4,4a,5,10b-hexahydrochromeno[4,3-b][1,4]oxazine* [*rac-(4a*R**,5*R**,10a*R**)-***2b**]: White crystals, yield 13%, mp 153–155 °C; ^1^H-NMR (400 MHz, aceton-*d*_6_) δ: 2.66 (d, *J* = 12.0 Hz, 1H, 3-Ha), 2.76 (m, 1H, 3-Hb), 3.33 (m, 4 H, 2-Ha, 2-Hb, 4a-H, NH), 3.81 (s, 3H, OCH_3_), 5.10 (d, *J* = 4.8 Hz, 1H, 10b-H), 5.23 (s, 1H, 5-H), 6.86 (dd, *J* = 8.4, 1.2 Hz, 1H, 7-H), 6.96 (m, 3H, 3′-H, 5′-H, 9-H), 7.18 (m, 1H, 8-H), 7.42 (d, *J* = 7.6 Hz, 1H, 10-H), 7.46 (d, 2H, 2′-H, 6′-H); ^13^C-NMR (100 MHz, acetone-*d*_6_) δ: 46.3 (C-3), 54.8 (C-4a), 55.5 (C-OCH_3_), 61.2 (C-2), 70.5 (C-10b), 78.9 (C-5), 114.4 (C-3′, C-5′), 116.9 (C-7), 121.6 (C-9), 122.4 (C-10a), 128.3 (C-2′, C-6′), 129.2 (C-8), 131.3 (C-1′), 156.4 (C-6a), 160.2 (C-4′); HRMS (ESI) calcd. for C_18_H_19_NO_3_ [M + H]^+^ 298.1438; found 298.1439.

*(±)-(4aR*,5S*,10bS*)-5-(3,4-dimethoxyphenyl)-2,3,4,4a,5,10b-hexahydrochromeno[4,3-b][1,4]oxazine* [*rac-(4a*R**,5*S**,10a*S**)-***2c**]: White crystals, yield 55%, mp 178–180 °C; ^1^H-NMR (360 MHz, CDCl_3_) δ: 1.71 (bs, 1H, NH), 2.89 (m, 2H, 3-Ha, 3-Hb, 4a-H), 3.86 (m, 7H, 2-Ha, 2xOCH_3_), 4.04 (dd, *J* = 11.2, 2.2 Hz, 1H, 2-Hb), 4.56 (d, *J* = 9.0 Hz, 1H, 10b-H), 4.90 (d, *J* = 9.7 Hz, 1H, 5-H), 6.87 (m, 2H, 2′-H, 5′-H), 6.97 (m, 3H, 6′-H, 7-H, 9-H), 7.21 (m, 1H, 8-H), 7.43 (d, *J* = 7.9 Hz, 1H, 10-H); ^13^C-NMR (90 MHz, CDCl_3_) δ: 46.4 (C-3), 56.1 (2xC-OCH_3_), 59.1 (C-4a), 67.5 (C-2), 76.1 (C-10b), 80.3 (C-5), 110.0 (C-5′), 111.3 (C-2′), 116.2 (C-6′), 120.2 (C-7), 121.0 (C-9), 122.3 (C-10a), 125.5 (C-10), 129.0 (C-8), 129.2 (C-1′), 149.5 (C-3′), 149.7 (C-4′), 153.6 (C-6a); HRMS (ESI) calcd. for C_19_H_21_NO_4_ [M + H]^+^ 328.1543; found 328.1543.

*(±)-(4aR*,5S*,10bR*)-5-(3,4-dimethoxyphenyl)-2,3,4,4a,5,10b-hexahydrochromeno[4,3-b][1,4]oxazine* [*rac-(4aR*,5S*,10aR*)-***2c**]): White crystals, yield 34%, mp 124–126 °C; ^1^H-NMR (400 MHz, CDCl_3_) δ: 2.31 (bs, 1H, NH), 2.90 (m, 3H, 2-Ha, 3-Ha, 3-Hb), 3.57 (d, *J* = 12.4 Hz, 1H, 2-Hb), 3.87 (s, 3H, OCH_3_), 3.93 (s, 3H, OCH_3_), 4.32 (s, 1H, 5-H), 4.89 (d, *J* = 7.6 Hz, 1H, 4a-H), 5.42 (d, *J* = 7.6 Hz, 1H, 10b-H), 6.85 (d, *J* = 8.4 Hz, 1H, 5′-H), 6.94 (d, *J* = 8.0 Hz, 1H, 7-H), 6.98 (m, 1H, 9-H), 7.03 (dd, *J* = 8.0, 2.0 Hz, 1H, 6′-H), 7.09 (d, *J* = 1.6 Hz, 1H, 2′-H), 7.28 (m, 1H, 8-H), 7.40 (d, *J* = 7.6 Hz, 1H, 10-H); ^13^C-NMR (100 MHz, CDCl_3_) δ: 50.6 (C-3), 56.0 (2xC-OCH_3_), 61.2 (C-5), 68.1 (C-2), 79.0 (C-10b), 90.3 (C-4a), 110.5 (C-7), 110.6 (C-5′), 111.3 (C-2′), 119.1 (C-6′), 121.5 (C-9), 124.9 (C-10a), 126.5 (C-10), 131.0 (C-8), 136.2 (C-1′), 148.2 (C-4′), 149.1 (C-3′), 160.2 (C-6a); HRMS (ESI) calcd. for C_19_H_21_NO_4_ [M + H]^+^ 328.1543; found 328.1544.

*(±)-(4aR*,5R*,10bR*)-5-(3,4-dimethoxyphenyl)-2,3,4,4a,5,10b-hexahydrochromeno[4,3-b][1,4]oxazine* [*rac-(4aR*,5R*,10aR*)-***2c**]: White crystals, yield 16%, mp 139–140 °C; ^1^H-NMR (400 MHz, acetone-*d*_6_) δ: 2.66 (d, *J* = 12.0 Hz, 1H, 3-Ha), 2.77 (m, 1H, 3-Hb), 3.35 (m, 4 H, 2-Ha, 2-Hb, 4a-H, NH), 3.82 (d, 6H, 2 × OCH_3_), 5.10 (d, *J* = 4.0 Hz, 1H, 10b-H), 5.23 (s, 1H, 5-H), 6.86 (dd, *J* = 8.4, 1.2 Hz, 1H, 7-H), 6.96 (m, 2H, 6′-H, 9-H), 7.06 (m, 1H, 5′-H), 7.16 (d, *J* = 2.0 Hz, 1H, 2′-H), 7.18 (m, 1H, 8-H), 7.42 (d, *J* = 7.6 Hz, 1H, 10-H); ^13^C-NMR (100 MHz, acetone-*d*_6_) δ: 45.5 (C-3), 54.1 (C-OCH_3_), 55.2 (C-4a), 55.3 (C-OCH_3_), 60.3 (C-2), 69.7 (C-10b), 78.1 (C-5), 110.3 (C-5′), 111.7 (C-2′), 116.1 (C-6′), 118.4 (C-7), 120.7 (C-9), 121.5 (C-10a), 127.3 (C-10), 128.3 (C-8), 130.9 (C-1′), 149.0 (C-4′), 149.3 (C-3′) 155.4 (C-6a); HRMS (ESI) calcd. for C_19_H_21_NO_4_ [M + H]^+^ 328.1543; found 328.1543.

(±)-(4aR*,5S*,10bS*)-5-(3,5-dimethoxyphenyl)-2,3,4,4a,5,10b-hexahydrochromeno[4,3-b][1,4]oxazine hydrochloride [rac-(4aR*,5S*,10aS*)-**2d**]: White crystals, yield 84%, mp 238–241 °C; ^1^H-NMR (400 MHz, DMSO-*d*_6_) δ: 3.12 (m, 1H, 3-Ha), 3.24 (d, *J* = 12.8 Hz, 1H, 3-Hb), 3.75 (m, 7H, 4a-H, 2xOCH_3_), 4.08 (m, 1H, 2-Ha), 4.17 (dd, *J* = 12.4, 4.0 Hz, 1H, 2-Hb), 5.20 (d, *J* = 9.6 Hz, 1H, 10b-H), 5.38 (d, *J* = 10.4 Hz, 1H, 5-H), 6.58 (t, *J* = 2.0 Hz, 1H, 4′-H), 6.60 (d, 2H, 2′-H, 6′-H), 6.89 (d, *J* = 8.4 Hz, 1H, 7-H), 7.01 (m, 1H, 9-H), 7.26 (m, 1H, 8-H), 7.38 (d, *J* = 7.6 Hz, 1H, 10-H), 8.33 (bs, 1H, NH_2_-Ha), 10.93 (bs, 1H, NH_2_-Hb); ^13^C-NMR (100 MHz, DMSO-*d*_6_) δ: 43.9 (C-3), 55.3 (2 × C-OCH_3_), 55.6 (C-4a), 63.4 (C-2), 71.9 (C-10b), 76.2 (C-5), 101.3 (C-4′), 106.2 (C-2′, C-6′), 116.1 (C-7), 120.5 (C-10a), 121.3 (C-9), 125.6 (C-10), 129.6 (C-8), 136.3 (C-1′), 152.6 (C-6a), 160.7 (C-3′, C-5′); HRMS (ESI) calcd. for C_19_H_21_NO_4_ [M + H]^+^ 328.1543; found 328.1544.

*(±)-(4aR*,5S*,10bR*)-5-(3,5-dimethoxyphenyl)-2,3,4,4a,5,10b-hexahydrochromeno[4,3-b][1,4]oxazine [rac-(4aR*,5S*,10aR*)-***2d***]*. Colorless oil, yield 48%, ^1^H-NMR (400 MHz, CDCl_3_) δ: 2.80 (m, 4 H, 2-Ha, 3-Ha, 3-Hb, NH), 3.46 (d, *J* = 12.0 Hz, 1H, 2-Hb), 3.71 (s, 6H, 2 × OCH_3_), 4.19 (s, 1H, 5-H), 4.78 (d, *J* = 7.6 Hz, 1H, 4a-H), 5.31 (d, *J* = 7.6 Hz, 1H, 10b-H), 6.27 (t, *J* = 2.0 Hz, 1H, 4′-H), 6.57 (d, 2H, 2′-H, 6′-H), 6.82 (d, *J* = 8.4 Hz, 1H, 7-H), 6.87 (t, *J* = 7.2 Hz, 1H, 9-H), 7.18 (m, 1H, 8-H), 7.29 (d, *J* = 7.2 Hz, 1H, 10-H); ^13^C-NMR (100 MHz, CDCl_3_) δ: 50.5 (C-3), 55.5 (2 × C-OCH_3_), 61.5 (C-5), 68.2 (C-2), 79.0 (C-10b), 90.2 (C-4a), 99.2 (C-4′), 105.2 (C-2′, C-6′), 110.6 (C-7), 121.5 (C-9), 124.8 (C-10a), 126.5 (C-10), 131.1 (C-8), 145.7 (C-1′), 160.2 (C-6a), 161.0 (C-3′, C-5′); HRMS (ESI) calcd. for C_19_H_21_NO_4_ [M + H]^+^ 328.1543; found 328.1543.

*(±)-(4aR*,5R*,10bR*)-5-(3,5-dimethoxyphenyl)-2,3,4,4a,5,10b-hexahydrochromeno[4,3-b][1,4]oxazine [rac-(4aR*,5R*,10aR*)-**2d**]:* White crystals, yield 5%, mp 145–147 °C; ^1^H-NMR (400 MHz, acetone-*d*_6_) δ: 2.68 (d, *J* = 12.0 Hz, 1H, 3-Ha), 2.79 (m, 2H, 3-Hb, NH), 3.38 (m, 3H, 2-Ha, 2-Hb, 4a-H), 3.81 (s, 6H, 2OCH_3_), 5.13 (d, *J* = 4.0 Hz, 1H, 10b-H), 5.25 (s, 1H, 5-H), 6.45 (t, *J* = 2.4 Hz, 1H, 4′-H), 6.74 (d, 2H, 2′-H, 6′-H), 6.88 (d, *J* = 8.4 Hz, 1H, 7-H), 6.97 (t, *J* = 7.6 Hz, 1H, 9-H), 7.19 (m, 1H, 8-H), 7.42 (d, *J* = 7.6 Hz, 1H, 10-H); ^13^C-NMR (100 MHz, acetone-*d*_6_) δ: 46.3 (C-3), 54.9 (C-4a), 55.6 (2 × C-OCH_3_), 61.1 (C-2), 70.4 (C-10b), 79.0 (C-5), 100.2 (C-4′), 105.0 (C-2′, C-6′), 117.0 (C-7), 121.7 (C-9), 122.3 (C-10a), 128.2 (C-10), 129.3 (C-8), 141.7 (C-1′), 156.1 (C-6′), 161.9 (C-3′, C-5′); HRMS (ESI) calcd. for C_19_H_21_NO_4_ [M + H]^+^ 328.1543; found 328.1543.

*(±)-(4aR*,5S*,10bS*)-5-(3,4,5-trimethoxyphenyl)-2,3,4,4a,5,10b-hexahydrochromeno[4,3-b][1,4]oxazine [rac-(4aR*,5S*,10aS*)-**2e**]*: Colorless oil, yield 46%, ^1^H-NMR (360 MHz, CDCl_3_) δ: 1.72 (bs, 1H, NH), 2.91 (m, 3H, 3-Ha, 3-Hb, 4a-H), 3.87 (m, 10 H, 2-Ha, 3 × OCH_3_), 4.06 (dd, *J* = 11.2, 1.8 Hz, 1H, 2-Hb), 4.56 (d, *J* = 9.4 Hz, 1H, 10b-H), 4.88 (d, *J* = 10.1Hz, 1H, 5-H), 6.70 (s, 2H, 2′-H, 6′-H), 6.89 (dd, *J* = 8.3, 1.1Hz, 1H, 7-H), 6.98 (m, 1H, 9-H), 7.20 (m, 1H, 8-H), 7.44 (d, *J* = 7.6 Hz, 1H, 10-H); ^13^C-NMR (90 MHz, CDCl_3_) δ: 46.4 (C-3), 56.3 (2 × C-OCH_3_), 59.2 (C-4a), 60.9 (C-OCH_3_), 67.5 (C-2), 76.1 (C-10b), 80.6 (C-5), 104.2 (C-2′, C-6′), 116.2 (C-7), 121.1 (C-9), 122.2 (C-10a), 125.5 (C-10), 129.1 (C-8), 132.5 (C-1′), 138.4 (C-4′), 153.5 (C-6a), 153.7 (C-3′, C-5′); HRMS (ESI) calcd. for C_20_H_23_NO_5_ [M + H]^+^ 358.1649; found 358.1648.

*(±)-(4aR*,5S*,10bR*)-5-(3,4,5-trimethoxyphenyl)-2,3,4,4a,5,10b-hexahydrochromeno[4,3-b][1,4]oxazine [rac-(4aR*,5S*,10aR*)-**2e**]*: Colorless oil, yield 33%, ^1^H-NMR (400 MHz, CDCl_3_) δ: 2.13 (bs, 1H, NH), 2.91 (m, 3H, 2-Ha, 3-Ha, 3-Hb), 3.58 (d, *J* = 12.0 Hz, 1H, 2-Hb), 3.83 (s, 3H, OCH_3_), 3.91 (s, 6 H, 2 × OCH_3_), 4.30 (s, 1H, 5-H), 4.90 (d, *J* = 7.6 Hz, 1H, 4a-H), 5.43 (d, *J* = 7.6 Hz, 1H, 10b-H), 6.75 (s, 2H, 2′-H, 6′-H), 6.95 (d, *J* = 8.0 Hz, 1H, 7-H), 6.99 (t, *J* = 7.6, 7.2 Hz, 1H, 9-H), 7.23 (t, *J* = 8.0, 7.2 Hz, 1H, 8-H), 7.40 (d, *J* = 7.6 Hz, 1H, 10-H); ^13^C-NMR (100 MHz, CDCl_3_) δ: 50.5 (C-3), 56.3 (2 × C-OCH_3_), 60.9 (C-OCH_3_), 61.7 (C-5), 68.2 (C-2), 79.0 (C-10b), 90.2 (C-4a), 104.2 (C-2′, C-6′), 110.6 (C-7), 121.6 (C-9), 124.8 (C-10a), 126.5 (C-10), 131.1 (C-8), 137.2 (C-4′), 139.3 (C-1′), 153.4 (C-3′, C-5′), 160.2 (C-6a); HRMS (ESI) calcd. for C_20_H_23_NO_5_ [M + H]^+^ 358.1649; found 358.1649

(±)-(4aR*,5R*,10bR*)-5-(3,4,5-trimethoxyphenyl)-2,3,4,4a,5,10b-hexahydrochromeno[4,3-b][1,4]oxazine [rac-(4aR*,5R*,10aR*)-**2e**]: White crystals, yield 7%, mp 164–166 °C; ^1^H-NMR (400 MHz, acetone-*d*_6_) δ: 2.66 (d, *J* = 12.0 Hz, 1H, 3-Ha), 2.77 (m, 1H, 3-Hb), 3.37 (m, 4 H, 2-Ha, 2-Hb, 4a-H, NH), 3.74 (s, 3H, OCH_3_), 3.85 (s, 6H, 2 × OCH_3_), 5.10 (d, *J* = 4.0 Hz, 1H, 10b-H), 5.22 (s, 1H, 5-H), 6.87 (s, 3H, 2′-H, 6′-H, 7-H), 6.96 (m, 1H, 9-H), 7.18 (m, 1H, 8-H), 7.42 (d, *J* = 7.6 Hz, 1H, 10-H); ^13^C-NMR (100 MHz, acetone-*d*_6_) δ: 46.4 (C-3), 55.0 (C-4a), 56.4 (2 × C-OCH_3_), 60.5 (C-OCH_3_), 61.2 (C-2), 70.6 (C-10b), 79.2 (C-5), 104.5 (C-2′, C-6′), 117.0 (C-7), 121.7 (C-9), 122.5 (C-10a), 128.2 (C-10), 129.2 (C-8), 134.9 (C-1′), 154.3 (C-3′, C-5′), 156.2 (C-6a); HRMS (ESI) calcd. for C_20_H_23_NO_5_ [M + H]^+^ 358.1649; found 358.1648.

(±)-(4aR*,5S*,10bS*)-5-(naphthalen-1-yl)-2,3,4,4a,5,10b-hexahydrochromeno[4,3-b][1,4]oxazine hydrochloride [rac-(4aR*,5S*,10aS*)-**2f**]: White crystals, yield 53%, mp 241–244 °C; ^1^H-NMR (400 MHz, DMSO-*d*_6_) δ: 3.22 (s, 2H, 3-Ha, 3-H_b_), 4.19 (m, 3H, 2-Ha, 2-Hb, 4a-H), 5.52 (s, 1H, 10b-H), 6.43 (bs, 1H, 5-H), 6.87 (s, 1H, 7-H), 7.05 (t, *J* =7.6 Hz, 1H, 9-H), 7.27 (s, 1H, 8-H), 7.45 (d, *J* = 7.6 Hz, 1H, 10-H), 7.60 (s, 3H, 2′-H, 3′-H, 7′-H), 7.94 (s, 1H, 6′-H), 8.04 (m, 2H, 4′-H, 5′-H), 8.53 (s, 2H, 8′-H, NH_2_-Ha), 11.06 (bs, 1H, NH_2_-Hb); ^13^C-NMR (100 MHz, DMSO-*d*_6_) δ: 44.0 (C-3), 55.6 (C-4a), 63.4 (C-2), 70.8 (C-5), 71.9 (C-10b), 116.0 (C-7), 120.6 (C-10a), 121.3 (C-9), 124.2 (C-8′), 125.5 (C-10), 126.0 (C-2′, C-3′, C-7′), 126.7 (C-6′), 128.8 (C-5′), 129.6 (C-8), 130.7 (C-4′); HRMS (ESI) calcd. for C_21_H_19_NO_2_ [M + H]^+^ 318.1489; found 318.1486.

*(±)-(4aR*,5S*,10bS*)-5-naphthalen-2-yl-2,3,4,4a,5,10b-hexahydrochromeno[4,3-b][1,4]oxazine [rac-(4aR*,5S*,10aS*)-**2g**]:* White crystals, yield 51%, mp 102–104 °C; ^1^H-NMR (360 MHz, CDCl_3_) δ: 1.55 (bs, 1H, NH), 2.82 (m, 2H, 3-Ha, 3-Hb), 3.02 (t, *J* = 9.7, 9.4 Hz,1H, 4a-H), 3.86 (m, 1H, 2-Ha), 4.03 (dd, *J* = 11.5, 2.2 Hz, 1H, 2-Hb), 4.62 (d, *J* = 9.4 Hz, 1H, 10b-H), 5.11 (d, *J* = 9.7 Hz, 1H, 5-H), 6.90 (d, *J* = 8.3 Hz, 1H, 7-H), 6.98 (m, 1H, 9-H), 7.15 (m, 1H, 8-H), 7.46 (m, 3H, 10-H, 6′-H, 7′-H), 7.59 (dd, *J* = 8.3, 1.4 Hz, 1H, 3′-H), 7.85 (m, 4 H, 1′-H, 4′-H, 5′-H, 8′-H); ^13^C-NMR (90 MHz, CDCl_3_) δ: 46.3 (C-3), 59.0 (C-4a), 67.5 (C-2), 76.1 (C-10b), 80.6 (C-5), 116.2 (C-7), 121.1 (C-8), 122.3 (C-10a), 124.3 (C-1′), 125.5 (C-8′), 126.6 (C-3′), 126.7 (C-7′), 127.4 (C-9), 127.9 (C-6′), 128.2 (C-5′), 129.1 (C-10, C-4′), 133.2 (C-4a’), 133.8 (C-8a’), 134.3 (C-2′), 153.6 (C-6a); HRMS (ESI) calcd. for C_21_H_19_NO_2_ [M + H]^+^ 318.1489; found 318.1486.

*(±)-(4aR,5S,10bR)-5-(naphthalen-2-yl)-2,3,4,4a,5,10b-hexahydrochromeno[4,3-b][1,4]oxazine [rac-(4aR*,5S*,10aR*)-**2g**]*: White crystals, yield 44%, mp 154–156 °C; ^1^H-NMR (360 MHz, CDCl_3_) δ: 2.26 (bs, 1H, NH), 3.05 (m, 3H, 2-Ha, 3-Ha, 3-Hb), 3.70 (m, 1H, 2-Hb), 4.64 (s, 1H, 5-H), 5.11 (d, *J* = 7.9 Hz, 1H, 4a-H), 5.56 (d, *J* = 7.6 Hz, 1H, 10b-H), 7.04 (m, 2H, 7-H, 9-H), 7.39 (m, 1H, 8-H), 7.51 (m, 3H, 6′-H, 7′-H, 10-H), 7.70 (dd, *J* = 8.6, 1.8 Hz, 1H, 3′-H), 7.90 (m, 3H, 4′-H, 5′-H, 8′-H), 8.06 (s, 1H, 1′-H); ^13^C-NMR (90 MHz, CDCl_3_) δ: 50.7 (C-3), 61.4 (C-5), 68.5 (C-2), 79.1 (C-10b), 90.2 (C-4a), 110.6 (C-7), 121.5 (C-8), 124.9 (C-10a), 125.5 (C-1′), 125.6 (C-8′), 125.8 (C-3′), 126.2 (C-7′), 126.6 (C-9), 127.7 (C-6′), 128.2 (C-5′), 128.4 (C-4′), 131.1 (C-10), 132.7 (C-4a’), 133.5 (C-8a’), 140.8 (C-2′), 160.3 (C-6a); HRMS (ESI) calcd. for C_21_H_19_NO_2_ [M + H]^+^ 318.1489; found 318.1488.

*(±)-(4aR*,5R*,10bR*)-5-(naphthalen-2-yl)-2,3,4,4a,5,10b-hexahydrochromeno[4,3-b][1,4]oxazine [rac-(4aR*,5R*,10aR*)-**2g**]:* White crystals, yield 11%, mp 117–119 °C; ^1^H-NMR (360 MHz, acetone-*d*_6_) δ: 2.63 (d, *J* = 11.9 Hz, 1H, 3-Ha), 2.75 (m, 2H, 3-Hb, NH), 3.38 (m, 2H, 2-Ha, 2-Hb), 3.52 (d, *J* = 4.7 Hz, 1H, 4a-H), 5.19 (d, *J* = 4.0 Hz, 1H, 10b-H), 5.48 (s, 1H, 5-H), 6.95 (d, *J* = 8.3 Hz, 1H, 7-H), 7.00 (t, *J* = 7.6 Hz, 1H, 9-H), 7.22 (t, *J* = 7.2 Hz, 1H, 8-H), 7.46 (d, *J* = 7.6 Hz, 1H, 10-H), 7.52 (m, 2H, 6′-H, 7′-H), 7.66 (d, *J* = 8.6 Hz, 1H, 3′-H), 7.94 (m, 3H, 4′-H, 5′-H, 8′-H), 8.09 (s, 1H, 1′-H); ^13^C-NMR (90 MHz, acetone-*d*_6_) δ: 45.4 (C-3), 53.9 (C-4a), 60.3 (C-2), 69.6 (C-10b), 78.3 (C-5), 116.1 (C-7), 120.9 (C-8), 121.6 (C-10a), 124.2 (C-1′), 125.0 (C-8′), 125.9 (C-3′), 126.1 (C-7′), 127.4 (C-9), 127.6 (C-6′), 127.8 (C-5′), 128.0 (C-4′), 128.4 (C-10), 133.1 (C-4a’), 133.2 (C-8a’), 136.1 (C-2′), 155.3 (C-6a); HRMS (ESI) calcd. for C_21_H_19_NO_2_ [M + H]^+^ 318.1489; found 318.1488.

### 2.8. General Procedure for the Synthesis of Acetamide Derivatives rac-cis-***24a***-***e***,***g*** and rac-trans-***24a***-***g***

3-Aminoflavanone hydrochloride salts *rac-cis*-**1a-e**, **g** or *rac-trans*-**1a-g** (0.655 mmol) were suspended in anhydrous THF (5 mL) under inert atmosphere. Under stirring, Et_3_N (230 µL, 1.64 mmol) was added to the suspension at room temperature or at 0 °C. After 10 min, acetyl chloride (56 µL, 0.786 mmol) was added dropwise to the reaction mixture and stirred for additional 10 min. Extraction with ethyl acetate and water, drying over MgSO_4_, and concentration under reduced pressure provided the crude product, which was purified by column chromatography using hexane/ethyl acetate 1:1 as eluent.

*(±)-N-[(2S*,3R*)-4-oxo-2-phenyl-3,4-dihydro-2H-chromen-3-yl]acetamide (*rac*-*cis*-**24a**)*: White crystals, yield 69%, mp 169–171 °C; ^1^H-NMR (400 MHz, CDCl_3_) δ: 2.02 (s, 3H, CH_3_), 5.42 (t, *J* = 6.0 Hz, 1H, 3-H), 6.10 (d, *J* = 6.8 Hz, 1H, 2-H), 6.30 (d, *J* = 5.2 Hz, 1H, NH), 7.03 (m, 2H, 3′-H, 5′-H), 7.13 (m, 2H, 6-H, 8-H), 7.25 (m, 3H, 2′-H, 4′-H, 6′-H), 7.54 (m, 1H, 7-H), 7.82 (dd, *J* = 8.0, 1.6 Hz, 1H, 5-H); ^13^C-NMR (100 MHz, CDCl_3_) δ: 23.2 (CH_3_), 56.5 (C-3), 79.7 (C-2), 118.1 (C-8), 120.1 (C-4a), 121.6 (C-6), 126.9 (C-5), 127.2 (C-2′, C-6′), 128.7 (C-3′, C-5′), 128.9 (C-4′), 135.1 (C-1′), 137.2 (C-7), 160.5 (C-8a), 170.5 (amide carbonyl), 189.4 (C-4); HRMS (ESI) calcd. for C_17_H_15_NaNO_3_ [M+Na]^+^ 304.095; found 304.096.

*(±)-N-[(2S*,3R*)-4-oxo-2-(4-methoxyphenyl)-3,4-dihydro-2H-chromen-3-yl]acetamide (*rac*-*cis*-**24b**)*: White crystals, yield 72%, mp 147–149 °C; ^1^H-NMR (400 MHz, CDCl_3_) δ: 2.02 (s, 3H, CH_3_), 3.74 (s, 3H, OCH_3_), 5.40 (t, *J* = 6.4 Hz, 1H, 3-H), 6.06 (d, *J* = 6.4 Hz, 1H, 2-H), 6.28 (d, *J* = 5.2 Hz, 1H, NH), 6.75 (d, 2H, 3′-H, 5′-H), 6.99 (m, 4 H, 6-H, 8-H, 2′-H, 6′-H), 7.50 (m, 1H, 7-H), 7.82 (dd, *J* = 7.6, 1.6 Hz, 1H, 5-H); ^13^C-NMR (100 MHz, CDCl_3_) δ: 23.5 (C-CH_3_), 55.6 (C-OCH_3_), 56.9 (C-3), 79.9 (C-2), 114.4 (C-3′, C-5′), 118.5 (C-8), 120.4 (C-4a), 121.8 (C-6), 127.2 (C-5), 127.4 (C-1′), 129.0 (C-2′, C-6′), 137.4 (C-4), 160.3 (C-4′), 160.7 (C-8a), 170.7 (amide carbonyl), 190.0 (C-4); HRMS (ESI) calcd. for C_18_H_17_NaNO_4_ [M+Na]^+^ 334.105; found 334.107.

*(±)-N-[(2S*,3R*)-4-oxo-2-(3,4-dimethoxyphenyl)-3,4-dihydro-2H-chromen-3-yl]acetamide (*rac*-*cis*-**24c**):* White crystals, yield 75%, mp 183–185 °C; ^1^H-NMR (400 MHz, CDCl_3_) δ: 2.03 (s, 3H, CH_3_), 3.72 (s, 3H, OCH_3_), 3.81 (s, 3H, OCH_3_), 5.41 (t, *J* = 6.4 Hz, 1H, 3-H), 6.06 (d, *J* = 6.4 Hz, 1H, 2-H), 6.30 (bs, 1H, NH), 6.68 (m, 3H, 2′-H, 5′-H, 6′-H), 7.01 (m, 2H, 6-H, 8-H), 7.51 (m, 1H, 7-H), 7.83 (dd, *J* = 7.6, 1.6 Hz, 1H, 5-H); ^13^C-NMR (100 MHz, CDCl_3_) δ: 23.1 (C-CH_3_), 55.7 (C-OCH_3_), 55.8 (C-OCH_3_), 56.5 (C-3), 79.7 (C-2), 110.8 (C-5′), 111.0 (C-2′), 118.1 (C-8), 119,4 (C-6′), 120.0 (C-4a), 121.5 (C-6), 126.7 (C-5), 127.3 (C-1′), 137.1 (C-7), 148.9 (C-4′), 149.4 (C-3′), 160.2 (C-8a), 170.4 (amide carbonyl), 189.5 (C-4); HRMS (ESI) calcd. for C_19_H_19_NaNO_5_ [M+Na]^+^ 364.116; found 364.118.

*(±)-N-[(2S*,3R*)-4-oxo-2-(3,5-dimethoxyphenyl)-3,4-dihydro-2H-chromen-3-yl]acetamide (*rac*-*cis*-**24d**)*: White crystals, yield 67%, mp 144–147 °C; ^1^H-NMR (400 MHz, CDCl_3_) δ: 2.03 (s, 3H, CH_3_), 3.65 (s, 6H, 2 × OCH_3_), 5.39 (t, *J* = 6.4 Hz, 1H, 3-H), 6.00 (d, *J* = 6.4 Hz, 1H, 2-H), 6.30 (d, 2H, 2′-H, 6′-H), 6.34 (d, *J* = 2.0 Hz, 1H, 4′-H), 6.43 (d, *J* = 5.6 Hz, 1H, NH), 7.01 (m, 2H, 6-H, 8-H), 7.50 (m, 1H, 7-H), 7.80 (dd, *J* = 8.0, 1.6 Hz, 1H, 5-H); ^13^C-NMR (100 MHz, CDCl_3_) δ: 23.1 (C-CH_3_), 55.3 (2 × C-OCH_3_), 56.3 (C-3), 79.6 (C-2), 100.4 (C-4′), 105.4 (C-2′, C-6′), 118.0 (C-8), 120.1 (C-4a), 121.6 (C-6), 126.8 (C-5), 137.0 (C-1′), 137.2 (C-7), 160.3 (C-8a), 160.8 (C-3′, C-5′), 170.4 (amide carbonyl), 189.2 (C-4); HRMS (ESI) calcd. for C_19_H_19_NO_5_ [M + H]^+^ 342.134; found 342.134.

*(±)-N-[(2S*,3R*)-4-oxo-(3,4,5-trimethoxyphenyl)-3,4-dihydro-2H-chromen-3-yl]acetamide (*rac*-*cis*-**24e**)*: White crystals, yield 64%, mp 170–172 °C; ^1^H-NMR (400 MHz, CDCl_3_) δ: 2.04 (s, 3H, CH_3_), 3.67 (s, 6H, 2 × OCH_3_), 3.79 (s, 3H, OCH_3_), 5.41 (t, *J* = 6.0 Hz, 1H, 3-H), 6.03 (d, *J* = 6.4 Hz, 1H, 2-H), 6.38 (s, 2H, 2′-H, 6′-H), 6.42 (d, *J* = 5.6 Hz, 1H, NH), 7.04 (m, 2H, 6-H, 8-H), 7.53 (m, 1H, 7-H), 7.83 (dd, *J* = 8.0, 1.2 Hz, 1H, 5-H); ^13^C-NMR (100 MHz, CDCl_3_) δ: 23.1 (C-CH_3_), 56.0 (2 × C-OCH_3_), 56.4 (C-3), 60.8 (C-OCH_3_), 79.9 (C-2), 104.4 (C-2′, C-6′), 118.0 (C-8), 120.0 (C-4a), 121.6 (C-6), 126.7 (C-5), 130.4 (C-1′, C-4′), 137.1 (C-7), 153.2 (C-3′, C-5′), 160.3 (C-8a), 170.4 (amide carbonyl), 189.3 (C-4); HRMS (ESI) calcd. for C_20_H_21_NaNO_6_ [M+Na]^+^ 394.126; found 394.128.

*(±)-N-[(2S*,3S*)-4-oxo-2-phenyl-3,4-dihydro-2H-chromen-3-yl]acetamide (*rac*-*trans*-**24a***): White crystals, yield 75%, mp 192–194 °C; ^1^H-NMR (400 MHz, CDCl_3_) δ: 1.86 (s, 3H, CH_3_), 5.02 (dd, *J* = 12.0, 8.4 Hz, 1H, 3-H), 5.38 (d, *J* = 12.4 Hz, 1H, 2-H), 6.03 (d, *J* = 8.4 Hz, 1H, NH), 7.02 (m, 2H, 6-H, 8-H), 7.39 (m, 3H, 3′-H, 5′-H, 7-H), 7.49 (m, 3H, 2′-H, 4′-H, 6′-H), 7.88 (dd, *J* = 7.6, 0.8 Hz, 1H, 5-H); ^13^C-NMR (100 MHz, CDCl_3_) δ: 22.9 (C-CH_3_), 58.3 (C-3), 83.2 (C-2), 118.1 (C-8), 120.2 (C-4a), 122.1 (C-6), 127.7 (C-5), 127.8 (C-2′, C-6′), 128.6 (C-3′, C-5′), 129.4 (C-4′), 136.2 (C-1′), 136.6 (C-7), 161.4 (C-8a), 170.3 (amide carbonyl), 191.0 (C-4); HRMS (ESI) calcd. for C_17_H_15_NaNO_3_ [M+Na]^+^ 304.095; found 304.096.

*(±)-N-[(2S*,3S*)-4-oxo-2-(4-methoxyphenyl)-3,4-dihydro-2H-chromen-3-yl]acetamide (*rac*-*trans*-**24b**)*: White crystals, yield 71%, mp 189–191 °C; ^1^H-NMR (400 MHz, CDCl_3_) δ: 1.88 (s, 3H, CH_3_), 3.82 (s, 3H, OCH_3_), 5.08 (dd, *J* = 12.4, 8.4 Hz, 1H, 3-H), 5.31 (d, *J* = 12.4 Hz, 1H, 2-H), 5.90 (d, *J* = 8.4 Hz, 1H, NH), 6.92 (d, 2H, 3′-H, 5′-H), 7.00 (d, *J* = 8.4 Hz, 1H, 8-H), 7.05 (t, *J* = 7.2 Hz, 1H, 6-H), 7.42 (d, 2H, 2′-H, 6′-H), 7.49 (m, 1H, 7-H), 7.88 (dd, *J* = 7.2, 1.2 Hz, 1H, 5-H); ^13^C-NMR (100 MHz, CDCl_3_) δ: 23.0 (C-CH_3_), 55.4 (C-OCH_3_), 58.0 (C-3), 83.1 (C-2), 114.0 (C-3′, C-5′), 118.1 (C-8), 120.2 (C-4a), 122.0 (C-6), 127.6 (C-5), 128.2 (C-1′), 129.3 (C-2′, C-6′), 136.6 (C-7), 160.4 (C-4′), 161.4 (C-8a), 170.2 (amide carbonyl), 191.3 (C-4); HRMS (ESI) calcd. for C_18_H_17_NaNO_4_ [M+Na]^+^ 334.105; found 334.107.

*(±)-N-[(2S*,3S*)-4-oxo-2-(3,4-dimethoxyphenyl)-3,4-dihydro-2H-chromen-3-yl]acetamide (*rac*-*trans*-**24c**)*: White crystals, yield 82%, mp 180–181 °C; ^1^H-NMR (400 MHz, DMSO-*d*_6_) δ: 1.72 (s, 3H, CH_3_), 3.76 (d, 6 H, 2 × OCH_3_), 4.92 (dd, *J* = 12.4, 8.4 Hz, 1H, 3-H), 5.49 (d, *J* = 12.4 Hz, 1H, 2-H), 6.94 (m, 2H, 2′-H, 5′-H), 7.06 (d, *J* = 8.4 Hz, 1H, 8-H), 7.11 (m, 2H, 6-H, 6′-H), 7.58 (m, 1H, 7-H), 7.79 (dd, *J* = 7.6, 1.2 Hz, 1H, 5-H), 8.14 (d, *J* = 8.4 Hz, 1H, NH); ^13^C-NMR (100 MHz, DMSO-*d*_6_) δ: 22.3 (C-CH_3_), 55.4 (C-OCH_3_), 55.5 (C-OCH_3_), 57.4 (C-3), 81.5 (C-2), 111.1 (C-5′), 111.2 (C-2′), 118.0 (C-8), 120.0 (C-4a), 120.7 (C-6′), 121.8 (C-6), 126.9 (C-5), 129.3 (C-1′), 136.3 (C-7), 148.4 (C-4′), 149.2 (C-3′), 160.8 (C-8a), 169.1 (amide carbonyl), 190.5 (C-4); HRMS (ESI) calcd. for C_19_H_19_NaNO_5_ [M+Na]^+^ 364.116; found 364.118.

*(±)-N-[(2S*,3S*)-4-oxo-2-(3,5-dimethoxyphenyl)-3,4-dihydro-2H-chromen-3-yl]acetamide (*rac*-*trans*-**24d**):* White crystals, yield 72%, mp 192–193 °C; ^1^H-NMR (400 MHz, DMSO-*d*_6_) δ: 1.74 (s, 3H, CH_3_), 3.75 (s, 6H, 2 × OCH_3_), 4.85 (dd, *J* = 12.0, 8.4 Hz, 1H, 3-H), 5.51 (d, *J* = 12.0 Hz, 1H, 2-H), 6.51 (t, *J* = 2.0 Hz, 1H, 4′-H), 6.66 (d, 2H, 2′-H, 6′-H), 7.07 (d, *J* = 8.4 Hz, 1H, 8-H), 7.12 (t, *J* = 7.6 Hz, 1H, 6-H), 7.59 (m, 1H, 7-H), 7.79 (dd, *J* = 8.0, 1.6 Hz, 1H, 5-H), 8.18 (d, *J* = 8.4 Hz, 1H, NH); ^13^C-NMR (100 MHz, DMSO-*d*_6_) δ: 22.3 (C-CH_3_), 55.3 (2 × C-OCH_3_), 57.5 (C-3), 81.3 (C-2), 100.4 (C-4′), 105.8 (C-2′, C-6′), 118.0 (C-8), 120.0 (C-4a), 121.9 (C-6), 126.9 (C-5), 136.3 (C-7), 139.2 (C-1′), 160.2 (C-3′, C-5′), 160.7 (C-8a), 169.2 (amide carbonyl), 190.2 (C-4); HRMS (ESI) calcd. for C_19_H_19_NO_5_ [M + H]^+^ 342.134; found 342.134.

*(±)-N-[(2S*,3S*)-4-oxo-2-(3,4,5-trimethoxyphenyl)-3,4-dihydro-2H-chromen-3-yl]acetamide (*rac*-*trans*-**24e**)*: White crystals, yield 74%, mp 147–149 °C; ^1^H-NMR (400 MHz, CD_3_OD) δ: 1.89 (s, 3H, CH_3_), 3.80 (s, 3H, OCH_3_), 3.87 (s, 6H, 2 × OCH_3_), 5.04 (d, *J* = 12.4 Hz, 1H, 3-H), 5.42 (d, *J* = 12.4 Hz, 1H, 2-H), 6.84 (s, 2H, 2′-H, 6′-H), 7.07 (d, *J* = 8.4 Hz, 1H, 8-H), 7.13 (m, 1H, 6-H), 7.58 (m, 1H, 7-H), 7.89 (dd, *J* = 8.0, 1.6 Hz, 1H, 5-H); ^13^C-NMR (100 MHz, CD_3_OD) δ: 22.3 (C-CH_3_), 56.7 (2 × C-OCH_3_), 59.3 (C-3), 61.1 (C-OCH_3_), 83.9 (C-2), 106.3 (C-2′, C-6′), 119.1 (C-8), 121.3 (C-4a), 123.1 (C-6), 128.3 (C-5), 133.8 (C-1′), 137.7 (C-7), 139.6 (C-4′) 154.4 (C-3′, C-5′), 162.7 (C-8a), 173.2 (amide carbonyl), 192.1 (C-4); HRMS (ESI) calcd. for C_20_H_21_NaNO_6_ [M+Na]^+^ 394.126; found 394.128.

*(±)-N-[(2S*,3S*)-4-oxo-2-(naphthalen-1-yl)-3,4-dihydro-2H-chromen-3-yl]acetamide (*rac*-*trans*-**24f**)*: White crystals, yield 81%, mp 248–250 °C; ^1^H-NMR (400 MHz, DMSO-*d*_6_) δ: 1.58 (s, 3H, CH_3_), 5.12 (t, *J* = 11.6 Hz, 1H, 3-H), 6.42 (d, *J* = 12.0 Hz, 1H, 2-H), 7.07 (d, *J* = 8.0 Hz, 1H, 8-H), 7.17 (t, *J* = 8.0 Hz, 1H, 6-H), 7.21 (m, 4 H, 7-H, 2′-H, 3′-H, 7′-H), 7.81 (d, *J* = 5.6 Hz, 1H, 6′-H), 7.87 (dd, *J* = 8.0, 1.6 Hz, 1H, 5-H), 7.98 (m, 2H, 4′-H, 5′-H), 8.20 (m, 2H, 8′-H, NH); ^13^C-NMR (100 MHz, DMSO-*d*_6_) δ: 22.1 (C-CH_3_), 57.5 (C-3), 78.3 (C-2), 118.0 (C-6), 120.3 (C-4a), 122.1 (C-8), 123.5 (C-8′), 125.2 (C-5), 125.8 (C-2′, C-3′), 126.4 (C-7′), 127.0 (C-6′), 128.7 (C-5′), 129.4 (C-4′), 131.3 (C-1′), 132.8 (C-4a’), 133.3 (C-8a’), 136.4 (C-7), 160.9 (C-8a), 169.4 (amide carbonyl), 190.2 (C-4); HRMS (ESI) calcd. for C_21_H_17_NaNO_3_ [M+Na]^+^ 354.110; found 354.113.

*(±)-N-[(2S*,3S*)-4-oxo-2-(naphthalen-2-yl)-3,4-dihydro-2H-chromen-3-yl]acetamide* (*rac-trans-***24g**): White crystals, yield 73%, mp 200–202 °C; ^1^H-NMR (400 MHz, DMSO-*d*_6_) δ: 1.68 (s, 3H, CH_3_), 5.00 (dd, *J* = 12.4, 8.4 Hz, 1H, 3-H), 5.76 (d, *J* = 12.4 Hz, 1H, 2-H), 7.11 (d, *J* = 8.4 Hz, 1H, 8-H), 7.15 (m, 1H, 6-H), 7.55 (m, 2H, 3′-H, 7′-H), 7.61 (m, 1H, 7-H), 7.68 (dd, *J* = 8.4, 1.6 Hz, 1H, 5-H), 7.84 (dd, *J* = 8.0, 1.6 Hz, 1H, 6′-H), 7.95 (m, 3H, 4′-H, 5′-H, 8′), 8.02 (s, 1H, 1′-H), 8.21 (d, *J* = 8.4 Hz, 1H, NH); ^13^C-NMR (100 MHz, DMSO-*d*_6_) δ: 22.2 (C-CH_3_), 57.7 (C-3), 81.7 (C-2), 118.0 (C-8), 120.1 (C-4a), 122.0 (C-6), 125.1 (C-5), 126.4 (C-1′), 126.5 (C-8′), 126.9 (C-3′), 127.2 (C-7′), 127.6 (C-6′), 127.8 (C-5′), 128.0 (C-4′), 132.4 (C-1′), 133.1 (C-4a’), 134.5 (C-8a’), 136.4 (C-7), 160.8 (C-8a), 169.2 (amide carbonyl), 190.2 (C-4); HRMS (ESI) calcd. for C_21_H_17_NO_3_ [M + H]^+^ 332.128; found 332.128.

### 2.9. General Procedure for the Synthesis of Condensed Thiazole Derivatives ***3a***-***g***

Under inert atmosphere, acetamide derivatives *rac-cis-***24a-e**, **g** or *rac-trans-***24a-g** (0.355 mmol) and Lawesson’s reagent (0.355 mmol) were dissolved in anhydrous toluene (5 mL). The mixture was stirred for 4 h at 70 °C. After cooling, toluene was evaporated and the crude product was purified by column chromatography using hexane/ethyl acetate 8:1 or toluene/ethyl acetate 8:1 as eluent, which provided the pure product.

*(±)-2-methyl-4-phenyl-4H-chromeno[3,4-d][1,3]**thiazole* (**3a**): White crystals, yield 82%, mp 101–103 °C; ^1^H-NMR (400 MHz, CDCl_3_) δ: 2.67 (s, 3H, CH_3_), 6.54 (s, 1H, 4-H), 6.89 (m, 2H, 6-H, 8-H), 7.12 (m, 2H, 7-H, 9-H), 7.28 (m, 3H, 2′-H, 4′-H, 6′-H), 7.37 (m, 2H, 3′-H, 5′-H); ^13^C-NMR (100 MHz, CDCl_3_) δ: 19.6 (CH_3_), 78.7 (C-4), 117.0 (C-6), 118.2 (C-5a), 121.9 (C-8), 124.3 (C-9), 126.2 (C-9b), 127.3 (C-3′, C-5′), 128.6 (C-4′), 128.7 (C-2′, C-6′), 129.3 (C-7), 139.8 (C-1′), 147.4 (C-3a), 151.2 (C-9a), 165.2 (C-2); HRMS (ESI) calcd. for C_17_H_13_NaNOS [M+Na]^+^ 362.061; found 362.067.

(4*R*)-**3a**: *t*_R_ = 3.20 min on Chiralpak IA column (hexane/2-propanol 80:20), HPLC-ECD {λ [nm] (ϕ)}: 326sh (−3.68), 293 (−9.35), 280sh (−8.29), 243 (40.45), 218 (−169.42).

(4*S*)-**3a**: *t*_R_ = 3.78 min on Chiralpak IA column (hexane/2-propanol 80:20), HPLC-ECD {λ [nm] (ϕ)}: 326sh (4.65), 293 (10.30), 280 (9.23), 243 (−43.81), 218 (173.68).

*(±)-2-methyl-4-(4-methoxyphenyl)-4H-chromeno[3,4-d][1,3]**thiazole* (**3b**): White crystals, yield 52%. mp 108–109 °C; ^1^H-NMR (400 MHz, CDCl_3_) δ: 2.69 (s, 3H, CH_3_), 3.74 (s, 3H, OCH_3_), 6.49 (s, 1H, 4-H), 6.82 (d, 2H, 3′-H, 5′-H), 6.89 (m, 2H, 6-H, 8-H), 7.09 (m, 2H, 7-H, 9-H), 7.26 (d, 2H, 2′-H, 6′-H); ^13^C-NMR (100 MHz, CDCl_3_) δ: 19.6 (C-CH_3_), 55.3 (C-OCH_3_), 78.5 (C-4), 114.1 (C-3′, C-5′), 117.1 (C-6), 118.3 (C-5a), 121.8 (C-8), 124.3 (C-9), 126.3 (C-9b), 128.9 (C-2′, C-6′), 129.2 (C-7), 131.9 (C-1′), 147.7 (C-4′), 151.2 (C-3a), 159.9 (C-9a), 165.2 (C-2); HRMS (ESI) calcd. for C_18_H_15_NaNO_2_S [M+Na]^+^ 332.072; found 332.070.

(4*R*)-**3b**: *t*_R_ = 4.36 min on Chiralpak IA column (hexane/2-propanol 80:20), HPLC-ECD {λ [nm] (ϕ)}: 326sh (−1.96), 292 (−11.31), 280sh (−9.34), 257 (32.56), 223 (−86.35).

(4*S*)-**3b**: *t*_R_ = 5.02 min on Chiralpak IA column (hexane/2-propanol 80:20), HPLC-ECD {λ [nm] (ϕ)}: 326sh (1.87), 292 (7.45), 280sh (5.86), 257 (−17.30), 223 (50.79).

*(±)-2-methyl-4-(3,4-dimethoxyphenyl)-4H-chromeno[3,4-d][1,3]**thiazole* (**3c**): White crystals, yield 66%, mp 119–121 °C; ^1^H-NMR (400 MHz, CDCl_3_) δ: 2.71 (s, 3H, CH_3_), 3.81 (d, 6H, 2xOCH_3_), 6.48 (s, 1H, 4-H), 6.77 (d, *J* = 8.0 Hz, 1H, 6-H), 6.84 (d, *J* = 8.4 Hz, 1H, 8-H), 6.92 (m, 3H, 2′-H, 5′-H, 6′-H), 7.11 (m, 2H, 7-H, 9-H); ^13^C-NMR (100 MHz, CDCl_3_) δ: 19.6 (C-CH_3_), 55.9 (2xC-OCH_3_), 78.8 (C-4), 110.7 (C-5′), 111.1 (C-2′), 117.1 (C-6), 118.3 (C-5a), 120.0 (C-6′), 121.9 (C-8), 124.3 (C-9), 126.3 (C-9b), 129.3 (C-7), 132.2 (C-1′), 147.7 (C-3a), 149.1 (C-4′), 149.4 (C-3′), 151.2 (C-9a), 165.2 (C-2); HRMS (ESI) calcd. for C_19_H_17_NaNO_3_S [M+Na]^+^ 362.082; found 362.080.

(4*S*)-**3c**: *t*_R_ = 7.25 min on Chiralpak IA column (hexane/2-propanol 80:20), HPLC-ECD {λ [nm] (ϕ)}: 326sh (5.55), 294 (15.61), 263sh (−8.66), 241 (−22.57), 217 (103.93).

(4*R*)-**3c**: *t*_R_ = 7.58 min on Chiralpak IA column (hexane/2-propanol 80:20), HPLC-ECD {λ [nm] (ϕ)}: 326sh (−4.29), 294 (−14.08), 263sh (10.85), 241 (25.19), 217 (−98.22).

*(±)-2-methyl-4-(3,5-dimethoxyphenyl)-4H-chromeno[3,4-d][1,3]**thiazole* (**3d**): White crystals, yield 70%, mp 75–76 °C; ^1^H-NMR (360 MHz, CDCl_3_) δ: 2.67 (s, 3H, CH_3_), 3.69 (s, 6H, 2xOCH_3_), 6.36 (s, 1H, 4′-H), 6.47 (s, 1H, 4-H), 6.54 (d, 2H, 2′-H, 6′-H), 6.87 (m, 2H, 6-H, 8-H), 7.09 (m, 2H, 7-H, 9-H); ^13^C-NMR (90 MHz, CDCl_3_) δ: 19.5 (C-CH_3_), 55.3 (2 × C-OCH_3_), 78.4 (C-4), 100.2 (C-4′), 105.3 (C-2′, C-6′), 116.9 (C-6), 118.1 (C-5a), 121.9 (C-8), 124.3 (C-9), 126.2 (C-9b), 129.2 (C-7), 141.9 (C-1′), 147.1 (C-3a), 151.1 (C-9a), 160.8 (C-3′, C-5′), 165.2 (C-2); HRMS (ESI) calcd. for C_19_H_17_NaNO_3_S [M+Na]^+^ 362.082; found 362.081.

(4*R*)-**3d**: *t*_R_ = 4.80 min on Chiralpak IA column (hexane/2-propanol 80:20), HPLC-ECD {λ [nm] (ϕ)}: 326sh (−6.17), 294 (−10.80), 283sh (−3.78), 244 (29.05), 218 (−143.69).

(4*S*)-**3d**: *t*_R_ = 5.78 min on Chiralpak IA column (hexane/2-propanol 80:20), HPLC-ECD {λ [nm] (ϕ)}: 326sh (5.29), 294 (8.26), 283sh (2.36), 244 (−24.94), 218 (114.38).

*(±)-2-methyl-4-(3,4,5-trimethoxyphenyl)-4H-chromeno[3,4-d][1,3]**thiazole* (**3e**): White crystals, yield 84%, mp 113–115 °C; ^1^H-NMR (400 MHz, CDCl_3_) δ: 2.72 (s, 3H, CH_3_), 3.77 (m, 9H, 3 × OCH_3_), 6.46 (s, 1H, 4-H), 6.62 (s, 2H, 2′-H, 6′-H), 6.93 (m, 2H, 6-H, 8-H), 7.16 (m, 2H, 7-H, 9-H); ^13^C-NMR (101 MHz, CDCl_3_) δ: 20.0 (C-CH_3_), 56.5 (2 × C-OCH_3_), 61.2 (C-OCH_3_), 79.4 (C-4), 104.9 (C-2′, C-6′), 117.3 (C-6), 118.5 (C-5a), 122.4 (C-8), 124.7 (C-9), 126.7 (C-9b), 129.7 (C-7), 135.5 (C-1′, C-4′), 147.8 (C-3a), 151.6 (C-9a), 153.7 (C-3′, C-5′), 165.6 (C-2); HRMS (ESI) calcd. for C_20_H_19_NaNO_4_S [M+Na]^+^ 392.093; found 392.095.

(4*R*)-**3e**: *t*_R_ = 6.51 min on Chiralpak IA column (hexane/2-propanol 80:20), HPLC-ECD {λ [nm] (ϕ)}: 326sh (−4.54), 293 (−9.61), 285sh (−4.80), 263sh (7.01), 240 (13.85), 218 (−75.26).

(4*S*)-**3e**: *t*_R_ = 6.67 min on Chiralpak IA column (hexane/2-propanol 80:20), HPLC-ECD {λ [nm] (ϕ)}: 326sh (4.66), 293 (8.98), 285sh (4.36), 263sh (−6.88), 240 (−14.89), 218 (67.60).

*(±)-2-methyl-4-(naphthalen-1-yl)-4H-chromeno[3,4-d][1,3]**thiazole* (**3f**): White crystals, yield 60%, mp 173–175 °C; ^1^H-NMR (400 MHz, CDCl_3_) δ: 2.69 (s, 3H, CH_3_), 6.77 (d, *J* = 8.0 Hz, 1H, 6-H), 6.90 (t, *J* = 7.2 Hz, 1H, 8-H), 7.04 (s, 2H, 4-H, 7-H), 7.20 (m, 3H, 9-H, 2′-H, 3′-H), 7.50 (t, *J* = 7.6 Hz, 1H, 6′-H), 7.55 (t, *J* = 7.6 Hz, 1H, 7′-H), 7.77 (d, *J* = 8.0 Hz, 1H, 4′-H), 7.84 (d, *J* = 7.6 Hz, 1H, 5′-H), 8.45 (d, *J* = 8.4 Hz, 1H, 8′-H); ^13^C-NMR (100 MHz, CDCl_3_) δ: 19.7 (C-CH_3_), 76.2 (C-4), 117.3 (C-6), 118.5 (C-5a), 122.0 (C-8), 124.4 (C-9), 124.5 (C-8′), 125.1 (C-3′), 125.9 (C-6′), 126.6 (C-2′), 126.8 (C-7′), 127.4 (C-9b), 128.7 (C-5′), 129.2 (C-4′), 129.8 (C-7), 131.7 (C-8a’), 134.1 (C-1′), 134.3 (C-4a’), 147.1 (C-3a), 151.1 (C-9a), 165.2 (C-2); HRMS (ESI) calcd. for C_21_H_15_NaNOS [M+Na]^+^ 352.077; found 352.076.

(4*R*)-**3f**: *t*_R_ = 4.10 min on Chiralpak IA column (hexane/2-propanol 80:20), HPLC-ECD {λ [nm] (ϕ)}: 316 (−1.29), 274sh (−5.17), 239sh (−6.11), 223 (−8.99), 216 (9.89).

(4*S*)-**3f**: *t*_R_ = 5.62 min on Chiralpak IA column (hexane/2-propanol 80:20), HPLC-ECD {λ [nm] (ϕ)}: 316 (0.40), 274sh (3.19), 239sh (2.89), 223 (4.81), 216 (−6.90).

*(±)-2-methyl-4-(naphthalen-2-yl)-4H-chromeno[3,4-d][1,3]**thiazole* (**3g**): White crystals, yield 55%, mp 129–130 °C; ^1^H-NMR (400 MHz, CDCl_3_) δ: 2.67 (s, 3H, CH_3_), 6.70 (s, 1H, 4-H), 6.89 (m, 2H, 7-H, 9-H), 7.10 (m, 2H, 6-H, 8-H), 7.40 (m, 2H, 6′-H, 7′-H), 7.52 (d, *J* = 8.8 Hz, 1H, 3′-H), 7.74 (m, 4 H, 1′-H, 4′-H, 5′-H, 8′-H); ^13^C-NMR (100 MHz, CDCl_3_) δ: 19.6 (C-CH_3_), 79.0 (C-4), 117.0 (C-6), 118.2 (C-5a), 122.0 (C-8), 124.4 (C-9), 125.0 (C-8′), 126.2 (C-3′), 126.4 (C-6′), 126.4 (C-9b), 126.5 (C-1′), 127.7 (C-7′), 128.4 (C-5′), 128.6 (C-4′), 129.3 (C-7), 133.2 (C-4a’), 133.5 (C-8a’), 137.1 (C-2′), 147.4 (C-3a), 151.3 (C-9a), 165.3 (C-2); HRMS (ESI) calcd. for C_21_H_15_NaNOS [M+Na]^+^ 352.077; found 352.077.

(4*R*)-**3g**: *t*_R_ = 3.79 min on Chiralpak IA column (hexane/2-propanol 80:20), HPLC-ECD {λ [nm] (ϕ)}: 239 (6.99), 225 (−28.42), 214 (7.25).

(4*S*)-**3g**: *t*_R_ = 4.52 min on Chiralpak IA column (hexane/2-propanol 80:20), HPLC-ECD {λ [nm] (ϕϕ)}:239 (−5.03), 225 (12.80), 214 (−1.62).

### 2.10. General Procedure for the Knorr Reaction Affording the Pyrrole-Condensed Derivatives ***4a***-***g***

3-Aminoflavanone derivatives *rac-cis*-**1a-e**, **g** or *rac-trans*-**1a-g** (0.363 mmol) were dissolved in a mixture of 96% ethanol (4 mL) and water (2 mL). Under stirring, ethyl acetoacetate (55 µL, 0.436 mmol) and NaOAc x 3 H_2_O (300 mg, 2.178 mmol) were added to the reaction. The mixture was refluxed for 3 h and then water was added. Extraction with CH_2_Cl_2_, drying of the combined organic phases over MgSO_4_, and concentration under reduced pressure provided the crude product as orange oil. Column chromatography using toluene/ethyl acetate 4:1 with 0.1% Et_3_N as eluent and subsequent trituration in cold Et_2_O afforded the pure product.

*(±)-ethyl 2-methyl-4-phenyl-3,4-dihydrochromeno[3,4-b]pyrrole-1-carboxylate* (**4a**): Off-white crystals, yield 53%, mp 133–136 °C; ^1^H-NMR (400 MHz, CDCl_3_) δ: 1.37 (t, 3H, CH_3_), 2.42 (s, 3H, 2-CH_3_), 4.31 (m, 2H, CH_2_), 6.10 (s, 1H, 4-H), 6.91 (dd, *J* = 8.0, 1.6 Hz, 1H, 6-H), 6.97 (m, 1H, 8-H), 7.04 (m, 1H, 7-H), 7.40 (m, 5H, 2′-H, 3′-H, 4′-H, 5′-H, 6′-H), 7.73 (bs, 1H, NH), 8.27 (dd, *J* = 8.0, 2.0 Hz, 1H, 9-H); ^13^C-NMR (100 MHz, CDCl_3_) δ: 14.5 (2xCH_3_), 60.0 (C-CH_2_), 75.8 (C-4), 109.0 (C-5), 114.9 (C-9b), 116.8 (C-6), 121.6 (C-9a), 122.0 (C-8), 125.7 (C-3a), 126.1 (C-9), 126.9 (C-4), 128.0 (C-2′, C-6′), 129.1 (C-3′, C-5′), 129.4 (C-7), 136.7 (C-2), 138.2 (C-1′), 152.0 (C-5a), 165.9 (ester carbonyl); HRMS (ESI) calcd. for C_21_H_19_NO_3_ [M + H]^+^ 334.1438; found 334.1438.

(4*R*)-**4a**: *t*_R_ = 3.20 min on Chiralpak IA column (hexane/2-propanol 80:20), HPLC-ECD {λ [nm] (ϕ)}: 300 (−3.47), 250 (−1.88), 235 (8.76), 214 (−28.73).

(4*S*)-**4a**: *t*_R_ = 3.43 min on Chiralpak IA column (hexane/2-propanol 80:20), HPLC-ECD {λ [nm] (ϕϕ)}: 300 (3.06), 250 (2.22), 235 (−10.08), 214 (30.02).

*(±)-ethyl 2-methyl-4-(4-methoxyphenyl)-3,4-dihydrochromeno[3,4-b]pyrrole-1-carboxylate* (**4b**): White crystals, yield 32%, mp 169–171 °C; ^1^H-NMR (400 MHz, CDCl_3_) δ: 1.36 (t, 3H, CH_3_), 2.40 (s, 3H, 4-CH_3_), 3.76 (s, 3H, OCH_3_), 4.26 (m, 2H, CH_2_), 6.01 (s, 1H, 4-H), 6.86 (m, 3H, 6-H, 3′-H, 5′-H), 6.94 (t, *J* = 7.2 Hz, 1H, 7-H), 7.01 (t, *J* = 6.8 Hz, 1H, 8-H), 7.30 (d, 2H, 2′-H, 6′-H), 7.86 (bs 1H, NH), 8.25 (d, *J* = 6.8 Hz, 1H, 9-H); ^13^C-NMR (100 MHz, CDCl_3_) δ: 14.4 (C-CH_3_), 14.5 (C-CH_3_), 55.4 (C-OCH_3_), 60.0 (C-CH_2_), 75.4 (C-4), 108.9 (C-1), 114.4 (C-3′, C-5′), 114.9 (C-9b), 116.8 (C-6), 121.6 (C-9a), 121.9 (C-8), 125.9 (C-3a), 126.0 (C-9), 126.8 (C-7), 129.5 (C-2′, C-6′), 130.2 (C-1′), 136.7 (C-2), 152.0 (C-5a), 160.4 (C-4′), 165.9 (ester carbonyl); HRMS (ESI) calcd. for C_22_H_21_NaNO_4_ [M+Na]^+^ 364.1543; found 364.1546.

(4*R*)-**4b**: *t*_R_ = 7.13 min on Chiralpak IA column (hexane/2-propanol 90:10), HPLC-ECD {λ [nm] (ϕ)}: 310sh (−4.28), 277 (−5.65), 237 (33.61), 219 (−31.69), 203 (25.66).

(4*S*)-**4b**: *t*_R_ = 7.68 min on Chiralpak IA column (hexane/2-propanol 90:10), HPLC-ECD {λ [nm] (ϕ)}: 310 (3.55), 277 (4.50), 237 (−27.21), 219 (23.67), 203 (−26.07).

*(±)-ethyl 2-methyl-4-(3,4-dimethoxyphenyl)-3,4-dihydrochromeno[3,4-b]pyrrole-1-carboxylate* (**4c**): White crystals, yield 48%, mp 214–216 °C; ^1^H-NMR (400 MHz, CDCl_3_) δ: 1.40 (t, 3H, CH_3_), 2.45 (s, 3H, 4-CH_3_), 3.78 (m, 6H, 2 × OCH_3_), 4.35 (m, 2H, CH_2_), 6.02 (s, 1H, 4-H), 6.80 (d, *J* = 6.8 Hz, 1H, 5′-H), 6.93 (m, 5H, 6-H, 7-H, 8-H 2′-H, 6′-H), 8.04 (bs, 1H, NH), 8.28 (d, *J* = 5.6 Hz, 1H, 9-H); ^13^C-NMR (100 MHz, CDCl_3_) δ: 14.4 (C-CH_3_), 14.5 (C-CH_3_), 55.9 (2 × C-OCH_3_), 60.2 (C-CH_2_), 75.9 (C-4), 109.0 (C-1), 110.8 (C-2′), 111.0 (C-5′), 115.0 (C-9b), 116.8 (C-6′), 120.8 (C-6), 121.7 (C-9a), 122.0 (C-8), 126.0 (C-3a), 126.1 (C-9), 126.8 (C-7), 130.5 (C-1′), 136.8 (C-2), 149.4 (C-4′,), 149.8 (C-3′), 152.2 (C-5a), 165.9 (ester carbonyl); HRMS (ESI) calcd. for C_23_H_23_NaNO_5_ [M+Na]^+^ 416.1468; found 416.1466.

(4*R*)-**4c**: *t*_R_ = 9.60 min on Chiralpak IA column (hexane/2-propanol 90:10), HPLC-ECD {λ [nm] (ϕ)}: 308 (−2.89), 238 (10.00), 215 (−15.99), 206 (−9.40).

(4*S*)-**4c**: *t*_R_ = 9.96 min on Chiralpak IA column (hexane/2-propanol 90:10), HPLC-ECD {λ [nm] (ϕ)}: 308 (2.20), 238 (−8.41), 215 (10.48), 206 (−12.66).

*(±)-ethyl 2-methyl-4-(3,5-dimethoxyphenyl)-3,4-dihydrochromeno[3,4-b]pyrrole-1-carboxylate* (**4d**): White crystals, yield 35%, mp 156–158 °C; ^1^H-NMR (400 MHz, CDCl_3_) δ: 1.38 (t, 3H, CH_3_), 2.45 (s, 3H, 4-CH_3_), 3.79 (s, 6H, 2 × OCH_3_), 4.30 (m, 2H, CH_2_), 6.04 (s, 1H, 4-H), 6.49 (s, 1H, 4′-H), 6.63 (d, 2H, 2′-H, 6′-H), 6.96 (m, 3H, 6-H, 7-H, 8-H), 7.60 (bs, 1H, NH), 8.27 (dd, *J* = 7.6, 1.2 Hz, 1H, 9-H); ^13^C-NMR (100 MHz, CDCl_3_): δ: 14.6 (C-CH_3_), 55.6 (2 × C-OCH_3_), 60.1 (C-CH_2_), 75.8 (C-4), 101.3 (C-4′), 105.6 (C-2′, C-6′), 109.2 (C-1), 114.9 (C-9a), 116.9 (C-6), 121.6 (C-9a), 122.2 (C-8), 125.6 (C-3a), 126.2 (C-9), 126.9 (C-7), 136.6 (C-2), 140.4 (C-1′), 152.2 (C-5a), 161.5 (C-3′, C-5′), 165.8 (ester carbonyl); HRMS (ESI) calcd. for C_23_H_23_NaNO_5_ [M+Na]^+^ 416.1468; found 416.1464.

(4*R*)-**4d**: *t*_R_ = 4.53 min on Chiralpak IA column (hexane/2-propanol 80:20), HPLC-ECD {λ [nm] (ϕ)}: 310 (−4.70), 254 (−2.22), 237 (10.34), 216 (−20.58), 205 (27.90).

(4*S*)-**4d**: *t*_R_ = 4.80 min on Chiralpak IA column (hexane/2-propanol 80:20), HPLC-ECD {λ [nm] (ϕ)}: 310 (4.06), 254 (2.07), 237 (−9.85), 216 (19.44), 20 5(−21.99).

*(±)-ethyl 2-methyl-4-(3,4,5-trimethoxyphenyl)-3,4-dihydrochromeno[3,4-b]pyrrole-1-carboxylate* (**4e**): Pale yellow crystals, yield 55%, mp 199–201 °C; ^1^H-NMR (400 MHz, CDCl_3_): δ: 1.40 (t, 3H, CH_3_), 2.51 (s, 3H, 4-CH_3_), 3.77 (m, 9H, 3 × OCH_3_), 4.31 (m, 2H, CH_2_), 6.02 (s, 1H, 4-H), 6.66 (s, 2H, 2′-H, 6′-H), 6.93 (d, *J* = 7.6 Hz, 1H, 6-H), 6.99 (t, *J* = 7.6 Hz, 1H, 7-H), 7.05 (t, *J* = 7.2 Hz, 1H, 8-H), 8.28 (d, *J* = 7.2 Hz, 1H, 9-H), 8.44 (bs, 1H, NH); ^13^C-NMR (100 MHz, CDCl_3_) δ: 14.4 (C-CH_3_), 14.6 (C-CH_3_), 56.1 (2 × C-OCH_3_), 60.0 (C-CH_2_), 60.7 (C-OCH_3_), 76.6 (C-4), 105.1 (C-2′, C-6′), 108.9 (C-1), 115.0 (C-9b), 116.7 (C-6), 121.7 (C-9a), 122.1 (C-8), 125.8 (C-3a), 126.1 (C-9), 126.8 (C-7), 134.0 (C-1′), 136.8 (C-2), 138.1 (C-4′),152.2 (C-5a), 153.4 (C-3′, C-5′), 166.0 (ester carbonyl); HRMS (ESI) calcd. for C_24_H_25_NaNO_6_ [M+Na]^+^ 446.1574; found 446.1571.

(4*R*)-**4e**: *t*_R_ = 11.57 min on Chiralpak IC column (hexane/2-propanol 70:30), HPLC-ECD {λ [nm] (ϕ)}: 311 (−6.94), 291sh (−6.42), 240 (21.81), 217 (−36.71).

(4*S*)-**4e**: *t*_R_ = 16.28 min on Chiralpak IC column (hexane/2-propanol 70:30), HPLC-ECD {λ [nm] (ϕ)}: 311 (9.81), 291sh (9.30), 240 (−30.91), 217 (51.74).

*(±)-ethyl 2-methyl-4-(naphthalen-1-yl)-3,4-dihydrochromeno[3,4-b]pyrrole-1-carboxylate* (**4f**): White crystals, yield 51%, mp 204–207 °C; ^1^H-NMR (400 MHz, DMSO-*d*_6_) δ: 1.37 (t, 3H, CH_3_), 2.50 (s, 3H, 4-CH_3_), 4.32 (q, 2H, CH_2_), 6.68 (m, 1H, 6-H), 6.94 (m, 2H, 7-H, 8-H), 7.07 (d, *J* = 6.8 Hz, 1H, 2′-H), 7.15 (s, 1H, 4-H), 7.41 (t, *J* = 7.6 Hz, 1H, 3′-H), 7.59 (t, *J* = 7.2 Hz, 1H, 6′-H), 7.66 (t, *J* = 7.2 Hz, 1H, 7′-H), 7.95 (d, *J* = 8.0 Hz, 1H, 4′-H), 8.01 (d, *J* = 8.0 Hz, 1H, 5′-H), 8.38 (m, 1H, 9-H), 8.55 (d, *J* = 8.0 Hz, 1H, 8′-H), 11.50 (bs, 1H, NH); ^13^C-NMR (100 MHz, DMSO-*d*_6_) δ: 14.07 (C-CH_3_), 14.4 (C-CH_3_), 59.3 (C-CH_2_), 72.0 (C-4), 107.5 (C-1), 114.0 (C-9b), 116.6 (C-6), 121.4 (C-8), 121.7 (C-9a), 124.4 (C-8′), 125.2 (C-3′), 125.2 (C-3a), 125.6 (C-9), 126.0 (C-6′), 126.2 (C-7), 126.6 (C-2′), 126.8 (C-7′), 128.6 (C-5′), 129.6 (C-2), 131.2 (C-4a’), 133.7 (C-8a’), 133.8 (C-1′), 137.2 (C-4), 150.6 (C-5a), 165.2 (ester carbonyl); HRMS (ESI) calcd. for C_25_H_21_NaNO_3_ [M+Na]^+^ 406.1414; found 406.1411.

(4*R*)-**4f**: *t*_R_ = 3.66 min on Chiralpak IA column (hexane/2-propanol 80:20), HPLC-ECD {λ [nm] (ϕ)}: 323 (0.91), 271 (−6.26), 238sh (−10.92), 225 (−73.35), 212 (47.44).

(4*S*)-**4f**: *t*_R_ = 4.38 min on Chiralpak IA column (hexane/2-propanol 80:20), HPLC-ECD {λ [nm] (ϕ)}: 323 (−1.22), 271 (7.61), 238sh (12.80), 225 (78.13), 212 (−57.70).

*(±)-ethyl 2-methyl-4-(naphthalen-2-yl)-3,4-dihydrochromeno[3,4-b]pyrrole-1-carboxylate* (**4g**): White crystals, yield 60%, mp 149–151 °C; ^1^H-NMR (400 MHz, CDCl_3_) δ: 1.35 (s, 3H, CH_3_), 2.34 (s, 3H, 4-CH_3_), 4.29 (s, 2H, CH_2_), 6.21 (s, 1H, 4-H), 6.93 (m, 3H, 6-H, 8-H, 3′-H), 7.48 (m, 3H, 7-H, 6′-H, 7′-H), 7.80 (m, 5H, 9-H, 4′-H, 5′-H, 8′-H, NH), 8.28 (d, *J* = 6.0 Hz, 1H, 1′-H); ^13^C-NMR (100 MHz, CDCl_3_) δ: 14.4 (C-CH_3_), 14.5 (C-CH_3_), 60.0 (C-CH_2_), 75.9 (C-4), 109.0 (C-1), 114.9 (C-9b), 116.8 (C-6), 121.6 (C-9a), 122.2 (C-8), 125.2 (C-8′), 125.6 (C-3a), 126.1 (C-3′), 126.6 (C-9), 126.8 (C-6′), 126.9 (C-7), 127.4 (C-1′), 127.9 (C-7′), 128.3 (C-5′), 129.2 (C-4′), 133.2 (C-4a’), 133.8 (C-8a’), 135.5 (C-2′), 136.9 (C-2), 152.0 (C-5a), 165.9 (ester carbonyl); HRMS (ESI) calcd. for C_25_H_21_NaNO_3_ [M+Na]^+^ 406.1414; found 406.1414.

(4*R*)-**4g**: *t*_R_ = 3.86 min on Chiralpak IA column (hexane/2-propanol 80:20), HPLC-ECD {λ [nm] (ϕ)}: 317 (−6.27), 273 (−6.98), 236 (17.06), 222 (−37.60), 207 (55.29).

(4*S*)-**4g**: *t*_R_ = 4.20 min on Chiralpak IA column (hexane/2-propanol 80:20), HPLC-ECD {λ [nm] (ϕ)}: 317 (5.90), 273 (6.95), 236 (−18.81), 222 (35.02), 207 (−37.51).

### 2.11. X-Ray Diffraction Analysis

Single crystals of **17e** have been obtained with the slow evaporation of its solution in chloroform. Data collection was carried out at 298 K using Mo-Kα radiation (λ = 0.71073 Å) with a Burker-Nonius MACH3 diffractometer equipped with point detector. The structure could be solved by SIR-92 program [20] and refined by full-matrix least-squares method on F^2^ using the SHELX program [21]. Non-hydrogen atoms were refined anisotropically, hydrogen atoms were placed into geometric positions and methyl protons were fixed using the riding model. Publication material was prepared with the WINGX-suite [22] and publCIF software [23]. ORTEP view of the structure with selected geometric parameters are shown in Appendix A, other data are in the expected range. Further crystallographic information is compiled in Appendix A. The structure is deposited in the Cambridge Crystallographic Data Centre under CCDC 2016874.

### 2.12. MTT Assay

The number of viable cells was indirectly determined by measuring the conversion of the tetrazolium salt MTT (3-{4,5-dimethilthiasol-2-il}-2,5-diphenyltetrasolium bromide, Sigma-Aldrich) to formazan by mitochondrial dehydrogenases. Cells were plated in 96-well multi-titer plates (10,000 cells per well density) in quadruplicates and were cultured for 3 days and treated by the compounds daily. Negative control group was treated with equal amount of vehicle solvent (DMSO) and positive control group was treated with 1 µg/mL doxorubicin. Cells were then incubated with 5 mg/mL MTT for 3 h, precipitated formazan crystals were dissolved in acidic isopropanol (10% 1 M HCl in isopropanol supplemented with 10% Triton X 100), and concentration of formazan was assessed colorimetrical way measuring absorbance at 565 nm.

#### Determination of IC_50_

Logistic dose-response curves were fitted using the equation y = A2 + (A1-A2)/(1 + (x/x0)^p) where the parameters are: A1: initial value (ymin), A2: final value (ymax), x0: center (EC/IC_50_), and p is the calculated power. Fittings were carried out and parameters were calculated using Origin 8.6 (OriginLab Corporation, Northampton, MA, USA).

### 2.13. MitoProbe^TM^ DilC_1_(5) Assay and SYTOX Green Labeling

Decrease in mitochondrial membrane potential is an early hallmark of apoptosis and the disruption of the plasma membrane integrity is characteristic for cellular necrosis. These events were investigated simultaneously using MitoProbe^TM^ DilC_1_(5) assay kit and SYTOX green (both from Molecular Probes/ThermoiFisher) staining, respectively. DilC_1_(5) (1,1′,3,3,3′,3′-hexamethylindodicarbo - cyanine iodide) is a fluorescent cyanine dye which penetrates cytoplasm with intact membrane and accumulates primarily in mitochondria depending on the mitochondrial membrane potential. Since decrease in mitochondrial membrane potential is an early marker of apoptosis, the DilC_1_(5) staining intensity is typically decreased in apoptotic cells. SYTOX Green is a nucleic acid stain impermeant to live cells with intact plasma membrane, but it penetrates the compromised membrane of necrotic, dead cells resulting in a bright green fluorescent nuclear staining.

Cells were plated in 96-well multi-titer plates (10,000 cells per well density) in quadruplicates and were incubated for 24 h with various concentrations of *rac*-**19g**. Negative control group was treated with equal amount of vehicle solvent (DMSO) and positive control groups were shortly treated with 50 µM carbonyl cyanide 3- chlorophenylhydrazone (CCCP) or lysis buffer (20 mM Tris HCl, 5 mM EDTA in H_2_O) to disrupt mitochondrial membrane potential or to lyse cells and disrupt membrane integrity, respectively. Then supernatant were removed and cells were incubated with DilC_1_(5) and SYTOX Green following the manufacturer’s protocol. Finally, the excess of the dyes was removed, cells were gently washed in PBS, and fluorescence of DilC_1_(5) and SYTOX Green was measured at 630/680 nm and 490/520 nm (excitation/emission), respectively, using a FlexStation 3 (Molecular Devices) multimodal microplate reader.

### 2.14. CyQUANT^®^ Cell Proliferation Assay

CyQUANT assay assesses the cellular proliferation indirectly by directly determining the DNA content in a cell population using CyQUANT fluorescent dye. The dye exhibits strong fluorescence enhancement when bound to cellular DNA and the fluorescent intensity is proportional to the amount of bound DNA in a high dynamic range. Therefore, it is suitable to asses DNA synthesis associated with cellular proliferation. Cells were plated in 96-well multi-titer plates (10,000 cells per well density) in quadruplicates and were incubated for 24 h with various concentrations of *rac*-**19g**, then assayed for cellular proliferation following the manufacturer’s protocol. Briefly, supernatants were gently removed and the plate was snap frozen and stored at −70 °C. Then plate was thawed, cells were lysed and incubated with CyQUANT dye. The excess of the dye was removed and fluorescence was measured at 490/520 nm (excitation/emission) using a FlexStation 3 (Molecular Devices) multimodal microplate reader.

### 2.15. Computational Methods

Mixed torsional/low-frequency mode conformational searches were carried out by means of the Macromodel 10.8.011 software using the Merck Molecular Force Field (MMFF) with an implicit solvent model for CHCl_3_ [24]. Geometry reoptimizations were carried out at the B3LYP/6-31+G(d,p) level in vacuo, the CAM-B3LYP/TZVP [25] and the ωB97X/TZVP [26] levels with the PCM solvent model for CHCl_3_. TDDFT ECD calculations were run with various functionals (B3LYP, BH&HLYP, CAM-B3LYP, PBE0) and the TZVP basis set as implemented in the Gaussian 09 package with the same or no solvent model as in the preceding DFT optimization step [27]. ECD spectra were generated as sums of Gaussians with 1500–3000 cm^−1^ widths at half-height, using dipole-velocity-computed rotational strength values [28]. Ball-and-stick representations of the conformers were generated by using the Molekel software [29].

## 3. Results and Discussion

### 3.1. Synthesis

The tosyl oxime derivatives **5a**-**g**, starting materials of the Neber rearrangement for the synthesis of 3-aminoflavanones **1a**-**g** (Scheme 1), were prepared from 2′-hydroxyacetophenone (**13**) in four steps (Scheme 2).

2′-Hydroxyacetophenone (**13**) was reacted with seven different arenecarbaldehydes in a Claisen-Schmidt condensation reaction to produce the corresponding chalcones **15a**-**g**, which were transformed to racemic flavanone analogues **6a**-**g** in a biomimetic intramolecular oxa-Michael cyclization. The flavanones **6a**-**g** were converted to the oximes **16a**-**g** with NH_2_OH⋅HCl, which were tosylated to afford the oxime tosylates **5a**-**g**.

The oxime tosylates of cyclic ketones are common starting materials of the Neber rearrangement reaction, in which NaOEt or KOEt is generally used as a base in dry EtOH or benzene, followed by acidic hydrolysis to produce the hydrochloric salt of the α-aminoketone [2]. In enantioselective organocatalytic Neber rearrangements, in which the isolation of the optically active 2*H*-azirine derivative is needed, reactive oxime tosylates were reacted with thiourea [7,9] or cinchona organocatalysts [30] in the presence of an inorganic base and there was no subsequent acidic hydrolysis. In our experiments, we treated the oxime tosylates **5a**-**g** with NaOEt base in dry toluene for one day at room temperature and performed the acidic hydrolysis with 3N HCl solution for two hours on the CH_2_Cl_2_ solution of the concentrated filtrate. This condition resulted in the formation of both the *cis*- (*rac-cis*-**1a-e**, **g**) and *trans*-3-aminoflavanone (*rac-trans*-**1a-g**) derivatives, which could be readily separated and isolated in the work-up procedure (Table 1).

After the acidic hydrolysis, the crude product was dissolved in CH_2_Cl_2_ and 3 N HCl solution (3 mL) was added to it and the resultant orange suspension was filtered and the solid was washed with acetone that afforded the hydrochloride salt of the pure *rac-cis*-**1a-e**, **g** as white powder. The filtrate was concentrated and triturated with acetone to produce the hydrochloride salt of *rac-trans*-**1a-g** as off-white powder. The values of the ^3^*J*_2-H,3-H_ coupling constants were found in the range of 5.2–5.6 Hz for *rac-cis*-**1a-e**, **g**, while in the range of 12.4–12.6 Hz for *trans*-**1a-g**. The residue was concentrated and purification by column chromatography provided the 2-styrylbenzoxazole side-products **17a**-**g** with 10–20% yield, which were obtained by ring-opening of the γ-pyrone ring and intramolecular cyclization of the phenolic hydroxyl group on the intermediate of the Beckmann rearrangement. The planar structure of **17e** was also confirmed by single crystal X-ray diffraction analysis (see Appendix A for details). The formation of similar 2-styrylbenzoxazoles was also reported in the Beckmann reaction of flavanones through the *trans*-chalcone oximes [31]. The *cis*- (*rac-cis*-**1a-e**, **g**) and *trans*-3-aminoflavanones (*rac-trans*-**1a-g**) were isolated in 1:1 ratio with a C-2 phenyl substituent (**1a**) and with an approximate two-fold excess of the *trans* isomer with other C-2 aryl groups except for **1f**, where only the *rac-trans-***1f** was obtained (Table 1). In the reported Neber rearrangements of flavanones, the acidic hydrolysis step was performed for a longer period of time and surprisingly the *trans* isomer of 2-aminoflavanone could be isolated as the single product [5,6]. The formation of both *cis*- and *trans*-3-amino-2-methylchroman-4-one was described in the Neber reaction of the *O*-(*p*-tolylsulfonyl)oxime of 2-methylchroman-4-one [32], and *cis* diastereoselectivity was reported in the Neber reaction of a 2-aryl-4-piperidone derivative [33]. Our finding suggested that the *cis*-3-aminoflavanone derivatives *rac-cis*-**1a-e**, **g** formed initially through the corresponding 2*H*-azirine (**F** → *cis*-**G** → *cis*-**H**) either diastereoselectively or together with the *trans* isomer (**F** → *trans*-**G** → *trans*-**H**) and then the acidic hydrolysis promoted the conversion of the *cis* isomer to the *trans* one by enolization-induced epimerization of the α-aminoketones **1a**-**g** (Scheme 3).

If the acidic treatment was maintained for a long time, all the *cis-*3-aminoflavanones were transformed to the lower energy *trans* isomer by enolization at C-2 as reported in literature examples [5,6]. With bulky C-2 aryl group such as the 1-naphthyl one in **1f**, only the *trans* product was isolated even with our procedure (Table 1).

Since the diastereomeric 2-aminoflavanone derivatives *rac-cis*-**1a-g** and *rac-trans*-**1a-g** could be obtained in pure form by simple filtration and trituration, we could use this asset to synthesize different stereoisomers of morpholine-condensed target molecules **1a**-**g** in a four-step sequence. In the first step, *rac-trans*-**1a-g** were acylated with chloroacetyl chloride, which was followed by the diastereoselective reduction of the carbonyl group with NaBH_4_ affording the *sec*-alcohols *rac*-**19a-g** with the all-*trans* relative configuration (Scheme 4).

The all *trans* relative configuration of *rac*-**19a-g** was confirmed by the ^3^*J*_2-H,3-H_ and ^3^*J*_3-H,4-H_ coupling constants, the values of which were found in the range of 9.2–10.4 Hz indicating the *trans-diaxial* relationship of 2-H/3-H and 3-H/4-H. The cyclization was carried out with NaH (*rac*-**19a-g** → *rac*-**20a-g**) and the resultant lactams were reduced with LiAlH_4_ to the target molecules *rac*-(4a*S**,5*R**,10b*R**)-**2a**-**g** by preserving the all-*trans* relative configuration. Most of the all-*trans*-*rac*-(4a*S**,5*R**,10b*R**)-**2a**-**g** were isolated as the hydrochloride salt except for the *rac*-(4a*S**,5*R**,10b*R**)-**2c** and -**2d**, since they had good solubility in the organic solvent.

The same synthetic scheme was also utilized for the preparation of stereoisomeric *rac*-**2a-e**, **g** target molecules starting from the *rac-cis*-**1a-e**, **g**. When the acylation reaction *rac-cis*-**1a-e**, **g** → *rac-cis*-2**1a-e**, **g** was performed at room temperature (Scheme 5), partial epimerization occurred at C-3 and the thermodynamically more stable *rac-trans*-**18a-e**, **g** formed as the main product instead of the expected *rac-cis*-2**1a-e**, **g**.

During the separation of the diastereomers by column chromatography, we observed that the whole amount of the *cis* isomer was converted to the *trans* one under the slightly acidic condition of the silica gel. In order to avoid the epimerization at C-3, the acylation was carried out at 0 °C and the crude product of *rac-cis*-2**1a-e**, **g** was reduced directly with NaBH_4_ (*rac-cis*-**21a-e**, **g** → *rac*-**22a-e**, **g**) without purification on column chromatography. The reduction of the ketone carbonyl group occurred diastereoselectively (dr ≥ 95:5) and it provided the all-*cis* stereoisomer of the alcohols *rac*-2**2a-e**, **g**. The *cis* orientation of 2-H, 3-H, and 4-H was determined by the NOE correlations 2-H/3-H, 2-H/4-H, and 3-H/4-H as well as by the small values of ^3^*J*_2-H,3-H_ and ^3^*J*_3-H,4-H_ coupling constants. The ^3^*J*_3-H,4-H_ coupling constant was measured 5.2 Hz for *rac*-**22a**, while the ^3^*J*_2-H,3-H_ was so small that it could not be resolved, since the 2-H had a sharp singlet in the ^1^H NMR spectrum. Cyclization of *rac*-**22a-e**, **g** with NaH afforded the lactam derivatives *rac*-**23a-e**, **g**, which were reduced with LiAlH_4_ in refluxing dioxane to produce surprisingly the *rac*-(4a*R**,5*S**,10b*R**)-**2a-e**, **g** as the major product (33–60%) and *rac*-(4a*R**,5*R**,10b*R**)-**2a-e**, **g** (5–20%) as the minor one (Scheme 5). In the major products *rac*-(4a*R**,5*S**,10a*R**)-**2a-e**, **g**, the C-5 chirality center was inverted to decrease the steric crowding of the *cis* substituents. The (4a*R**,5*S**,10b*R**) relative configuration was confirmed by NOE correlations of 4a-H with 10b-H and 5-H, the 7.6 Hz value for the ^3^*J*_4a-H,10b-H_ coupling constant and sharp singlet of ^1^H NMR signal for 5-H, which suggested *axial* orientation of the C-5 aryl and C-4a NH group and *equatorial* one of the C-10b oxygen. In contrast, with the all-*cis* relative configuration of the minor product *rac*-(4a*R**,5*R**,10b*R**)-**2a-e**, **g**, 5-H/10b-H, 5-H/4a-H, and 10b-H/4a-H NOE correlations, ^3^*J*_4a-H,10b-H_ = 4.0 Hz and a sharp ^1^H NMR singlet for 5-H were measured, which derived from *equatorial* orientation of the C-5 aryl and C-10b oxygen and *axial* one of the C-4a NH group. Three of the four possible diastereomers of 5-substituted chromeno[4,3-*b*][1,4]oxazines **2a**-**e**, **g** were synthesized, which may enable to study stereochemistry–activity relationship.

The α-amino-ketone moiety of *rac*-**1a**-**g** was also used to build condensed thiazole and pyrrole units at the C-3−C-4 bond, and the resultant heterocycles were tested for antiproliferative activities on human cancer cell lines. For the synthesis of thiazole-condensed heterocycles, the amino group of *rac-cis*-**1a-e**, **g** or *rac-trans*-**1a-g** was acetylated with acetyl chloride and the Lawesson reagent was utilized for the cyclization (Scheme 6). In the case of *cis*-**1a-e**, **g**, the acetylation reaction was carried out at 0 °C for 15 min, which could preserve the *cis* relative configuration of *rac-cis*-**24a-e**, **g**. Although the C-3 chirality center was lost during the cyclization, both the *cis*-**1a-e**, **g** and *rac-trans*-**1a-g** were cyclized separately to the thiazole-condensed derivatives *rac*-**3a-g**.

The cyclization of the *trans*-*N*-acetyl derivatives *rac-trans-***24a-g** provided consistently higher yields (Appendix A) than those of the corresponding *rac-cis-***24a-e**, **g**.

The Knorr condensation [34] of *α*-amino ketones *rac-trans*-**1a**-**g** with ethyl acetoacetate produced the pyrrole-condensed derivatives *rac*-**4a-g** after trituration with cold diethyl ether with moderate yield (Scheme 7).

### 3.2. Antiproliferative Activity

The antiproliferative activity of the morpholine-, thiazole-, and pyrrole-condensed derivatives and their precursors were evaluated initially against A2780 ovarian and WM35 melanoma cancer cell lines at 50 μM concentration by monitoring at 24 and 72 h (Appendix A). While morpholine and pyrrole units are common structural elements in cytotoxic compounds of synthetic or natural origin [35,36,37,38,39], there are fewer reports available on cytotoxic condensed thiazole derivatives [40,41]. The *N*-chloroacetyl-3-amino-flavan-4-ol derivatives *rac*-**19a**-**g** and **22a-g** exhibited strong antiproliferative activity regardless the stereochemistry against both cell lines at 50 μM concentration, while the related *N*-acetyl derivatives *rac-cis*-**24a**-**e**, **g** or *rac-trans*-**24a**-**g** were inactive or they had much weaker activity. This suggested that the *N*-chloroacetyl derivatives act as alkylating agents and the chloroacetyl moiety is essential for the activity. The *N*-chloroacetyl-3-amino-flavanone derivatives *rac-trans*-**18a**-**g** and *rac-trans*-**21a**-**g** had usually weaker activity than the corresponding flavan-4-ol derivatives *rac*-**19a**-**g** and *rac*-**22a**-**g**, which suggested that the reduction of the C-4 carbonyl group to hydroxyl improved the antiproliferative activity. The IC_50_ value was determined for the most active *N*-chloroacetyl derivative *rac*-**19g** using the MTT assay, which was found 0.15 and 3.5 µM against A2780 and WM35 cancer cell lines (Appendix A), respectively (Table 2). Against the non-cancerous HaCaT human keratinocytes, 6.06 µM IC_50_ value was measured, which implies a remarkable 50-fold selectivity.

Following 24 hrs incubation, *rac*-**19g** decreased the mitochondrial membrane potential [DilC_1_(5) staining], which is an early hallmark of apoptosis (Figure 2a), and inhibited proliferation-associated DNA synthesis (CyQUANT Assay) of A2780 ovarian carcinoma cells (Figure 2c). However, the integrity of the plasma membrane was found to be intact (negative SYTOX Green staining) arguing against necrotic cytotoxic effect of the compound (Figure 2b).

Interestingly, *rac*-**19g**, containing a 1,3-oxygenated-2-*N*-chloroacetylaminopropane subunit as part of the flavanol scaffold, may be considered a cyclic analogue of irreversible acid ceramidase (AC) inhibitors such as SACLAC and SOCLAC (Figure 3) [42].

SACLAC and SABRAC were found to inhibit the growth of chemoresistant forms of prostate cancer [43] and to reduce the viability of acute myeloid leukemia cells with an EC_50_ of approximately 3 µM across 30 human cell lines [44].

From the morpholine-condensed derivatives, the *rac*-(4a*S**,5*R**,10b*R**)-**2b** and *rac*-(4a*S**,5*R**,10b*S**)-**2d** seemed to be the most active at 50 μM concentration (Appendix A) but their IC_50_ values were found larger than 10 μM against both A2780 and WM35 cell lines. The thiazole-condensed derivative *rac*-**3e** containing a C-4 3,4,5-trimethoxyphenyl substituent had the best activities (2.74 and 2.14 μM IC_50_) among the prepared condensed *O,N*-heterocycles (Table 2). The type of the C-4 aryl substituent had significant effect on the antiproliferative activity, since the thiazole derivatives *rac*-**3a**,**b**,**c**,**g** had much weaker activities at 50 μM concentration. All the pyrrole-condensed derivatives *rac*-**4a**-**g** had distinct antiproliferative activity at 50 μM concentration and low micromolar IC_50_ values were measured for **4b**, **4c,** and **4g** in the range of 2.95–9.37 μM (Table 2). Similarly to compound **11** (Figure 1) [14], **4b**, **4c,** and **4g** may be viewed as simplified analogues of cytotoxic lemallarins such as lemallarin C [17], in which there is a substituted pyrrole-condensed 2*H*-chromene unit instead of the pyrrole-condensed coumarine heterocyclic core of lemallarins. Although the condensed derivatives were effective in decreasing the viability of the A2780 and WM35 cancer cell lines, they did not display remarkable selectivity toward the cancer cell lines when compared their potency to non-cancerous HaCaT human keratinocytes (Table 2).

### 3.3. Stereochemical Analysis

The antiproliferative activity of our condensed chiral *O,N*-heterocycles prompted us to separate the enantiomers with HPLC using chiral stationary phase, measure the online HPLC-ECD spectra, and determine the absolute configuration (AC) by TDDFT-ECD calculations. The online HPLC-ECD approach aided with ECD calculations was proven an efficient method for the stereochemical analysis of scalemic or racemic mixtures of bioactive natural products [45,46] or synthetic derivatives [47,48]. The enantiomers of *rac*-**3e** and *rac*-**4g** were separated on Chiralpak IA column using hexane/2-propanol 80:20 as eluent and even partial separation of the enantiomers was sufficient to record mirror-image online HPLC-ECD spectra, which were not optimized further (Figure 4).

Except for **4e** (Chiralpak IC, hexane/2-propanol 70:30), the same HPLC condition was utilized to separate the enantiomers of the related **3a**-**d**, **f**, **g**, and **4a**-**f**, which afforded a base-line separation of the enantiomers for **3a**, **b**, **d**, **f**, **g**, and **4e**, **f**. Mirror-image HPLC-ECD spectra were recorded in all the cases, which could be used to characterize the enantiomers and determine the AC.

The ACs of the separated enantiomers were deduced by the solution TDDFT-ECD protocol [49], which also revealed the low-energy solution conformers of the studied molecules. The initial MMFF conformational isomers of the arbitrarily chosen (*R*)-**3e** and (*R*)-**4g** were re-optimized separately at the B3LYP/6-31+G(d,p), CAM-B3LYP/TZVP PCM/CHCl_3_, and ωB97X/TZVP PCM/CHCl_3_ levels and ECD spectra were computed at four different levels for the resulting conformational ensembles (Figure 5).

The DFT re-optimization of the initial 35 MMFF conformers of (*R*)-**3e** resulted in 8, 6, and 9 low-energy conformers over 1% Boltzmann population at the B3LYP/6-31+G(d,p), CAM-B3LYP/TZVP PCM/CHCl_3_, and ωB97X/TZVP PCM/CHCl_3_ levels, respectively (Appendix A). In the four lowest energy ωB97X/TZVP PCM/CHCl_3_ conformers of (*R*)-**3e** (Figure 5a), the C-4 aryl group adopted axial orientation and the plane of the benzene ring of the 3,4,5-trimethoxyphenyl group was either near co-planar (ω_C-2′,C-1′,C-4, 4-H_ = −23.3° in conformer A) or perpendicular (ω_C-2′,C-1′,C-4, 4-H_ = −109.0° in conformer C) to the plane determined by the atoms 4-H, C-4, and C-1′. The computed ECD spectra of the four conformers showed only minor variations and all of them reproduced well the negative Cotton effects (CEs) at 326, 293, and 218 nm and the positive ones at 263 and 240 nm of the experimental HPLC-ECD spectrum of the first-eluting enantiomer of **3e**. The Boltzmann-weighted B3LYP/TZVP PCM/CHCl_3_ ECD spectrum of (*R*)-**3e** had the best agreement (Figure 5c) and thus (*R*) AC was determined for the first-eluting enantiomer of **3e**. The HPLC-ECD spectra of the thiazole-condensed derivatives **3a**-**e** had the same ECD profile (Appendix A), on the basis of which the AC of the separated enantiomers could be assigned. The configurational assignment was also confirmed by the TDDFT-ECD calculation of **3a** (Appendix A), which determined (*R*) AC for the first-eluting enantiomer of **3a**. Interestingly, the enantiomers of **3c**, containing a C-4 3,4-dimethoxyphenyl substituent, showed reversed elution order under the same HPLC condition, which was evident from the mirror-image HPLC-ECD spectrum of the first-eluting enantiomer (Appendix A). TDDFT-ECD calculations were performed to determine the AC for the enantiomers of the thiazole-condensed derivatives **3f** and **3g** with 1- and 2-naphthyl substituents, which were expected to influence both the ECD spectra and the chiral separation. The 2-naphthyl group of (*R*)-**3g** adopted an *axial* orientation in all the three low-energy ωB97X/TZVP PCM/CHCl_3_ conformers (Figure 5b) and the computed ECD spectra reproduced well the experimental HPLC-ECD spectrum of the first-eluting enantiomer, for which (*R*) AC was assigned (Figure 5d). The first-eluting enantiomer of **3f** had completely different HPC-ECD spectrum from those of **3a**-**e, g** with overlapping negative CEs and broad transitions in the range of 350–210 nm (Appendix A). Although ECD calculations of (*R*)-**3f** could not produce a perfect agreement because of the improper estimation of the conformational ensemble (Appendix A), the AC of the first-eluting enantiomer was determined as (*R*).

The DFT re-optimization of the initial 31 MMFF conformers of (*R*)-**4g** afforded 18 ωB97X/TZVP PCM/CHCl_3_ low-energy conformers above 1% population, which differed in the orientation of the C-1 ethyloxycarbonyl and the C-4 2-naphthyl substituents (Figure 6a and Appendix A). The 2-naphthyl group had an *equatorial* arrangement in 13 computed conformers with a total population of 79.2%, while the *axial* conformer was represented by 5 conformers with 20.3% sum population (Appendix A).

The computed ECD spectra of the *equatorial* and *axial* conformers were markedly different and the intense negative CE at 222 nm and the positive one at 207 nm derived from the *axial* conformers, since the *equatorial* conformers had different transitions in this region. The Boltzmann-weighted ECD spectra of (*R*)-**4g** reproduced well the experimental HPLC-ECD spectrum of the first-eluting enantiomer with negative CEs above 250 nm (Figure 6c), on the basis of which (*R*) AC was assigned to it. Similarly, (*R*) AC was deduced for the first-eluting enantiomer of **4f** containing a 1-naphthyl group by the ECD calculation, although it could not reproduce well the 225 nm negative CE, which possibly derived from one of the *equatorial* conformers with underestimated population (Appendix A). The HPLC-ECD spectra of the first-eluting enantiomers of **4a**-**e** were quite similar (Appendix A) and **4a** was used as a reference compound to determine the AC for the separated enantiomers with aryl substitution pattern. In the ωB97X/TZVP PCM/CHCl_3_ low-energy conformers of (*R*)-**4a** (Figure 6b and Appendix A), the equatorial conformers were the dominant with 83.2% total population (7 conformers), while the axial conformers had 16.7% sum population (3 conformers). The good agreement of the Boltzmann-weighted ECD spectrum of (*R*)-**4a** with that of the first-eluting enantiomer allowed assigning the AC of first-eluting enantiomer with negative CEs above 260 nm as (*R*) (Figure 6d and Appendix A). Since the first-eluting enantiomers of **4b**-**e** had same HPLC-ECD profile (Appendix A), their ACs were determined as (*R*), which means that the HPLC elution order did not change with the different substitution patterns of **4a**-**g**.

Enantiomers of *rac*-**20a**-**g** and *rac*-**23a**-**g**, containing a morpholin-3-one residue condensed with a flavan moiety, were also separated on Chiralpak IA column using hexane/2-propanol as eluent, which provided base-line separation for most of the molecules (Appendix A). Mirror-image HPLC-ECD spectra were recorded for the separated enantiomers, the long-wavelength ^1^*L*_b_ transition of which could be correlated with the absolute configuration and it could be used to check the validity of the chroman [50] or flavan helicity rule [51,52] on conformationally rigid condensed flavan derivatives with three chirality centers. The (4a*R*,5*S*,10b*S*)-**20a** was selected as a reference compound for the **20a**-**g** series and the solution TDDFT-ECD protocol was performed on it to determine its absolute configuration independently from the helicity rule. The CAM-B3LYP/TZVP PCM/CHCl_3_ re-optimization of the initial single MMFF conformer of (4a*R*,5*S*,10b*S*)-**20a** provided one low-energy conformer, in which the 4a-H, 5-H and 10b-H protons adopted axial orientation in agreement with the 9.2 and 10.4 Hz values for the ^3^*J*_4a-H,10b-H_ and ^3^*J*_4a-H,5-H_ coupling constants (Figure 7a).

The condensed 2*H*-3,4-dihydropyran ring had a half-chair conformation with *M*-helicity as defined by the negative value (−41.0°) of the torsional angle ω_C-6a, O-6, C-5,C-4a_ (Figure 7a,c). Positive CE was found for the highest wavelength ECD band in the CAM-B3LYP/TZVP PCM/CHCl_3_ ECD spectrum of (4a*R*,5*S*,10b*S*)-**20a**, which agreed well with the positive ^1^*L*_b_ band [282sh (4.70, 274 (5.26)] observed in the HPLC-ECD spectrum of the second-eluting enantiomer (Figure 7b). Thus the positive ^1^*L*_b_ band CE of (4a*R*,5*S*,10b*S*)-**20a** derives from *M*-helicity of the condensed flavan chromophore, which corroborates well the flavan semi-empirical helicity rule. For the configurational assignment of the separated enantiomers of **20b**-**g**, the sign of the long-wavelength ^1^*L*_b_ band CE was considered, since in contrast to the higher wavelength benzene transitions such as the ^1^*L*_a_, this is not expected to change with the different substitution of the C-5 aryl group [51]. The first-eluting enantiomer of **20a**,**b**,**f,** and **g** had negative ^1^*L*_b_ band CE, on the basis of which their ACs were assigned as (4a*S*,5*R*,10b*R*), while the positive ^1^*L*_b_ band CE of the first-eluting enantiomers of **20c**-**e** derived from (4a*R*,5*S*,10b*S*) AC (Appendix A). The presence of a C-3′ methoxy substituent, which is missing from **20a**,**b** changed the elution order of the enantiomers on the Chiralpak IA column for **20c**-**e**.

The separated enantiomers of *rac*-**23a**-**g** had all *cis* relative configuration and hence they differed in the AC of C-4a from the corresponding **20a**-**g** derivatives. Compound (4a*R*,5*R*,10b*R*)-**23a** was selected for TDDFT-ECD calculation. The CAM-B3LYP/TZVP PCM/CHCl_3_ re-optimization of the initial MMFF conformer of (4a*R*,5*R*,10b*R*)-**23a** provided only one low-energy conformer, in which the 5-H and 10b-H had *axial* orientation, while the 4a-H adopted *equatorial* one (Figure 8a).

The geometry of this conformer was in accordance with the 5.6 Hz value of the ^3^*J*_4a-H,10b-H_ coupling constant (ω_4a-H,C-4a,C-10b,10b-H_ = 39.1°) and the broad unresolved singlet of 5-H (ω_4a-H,C-4a,C-5,5-H_ = 60.6°). In the computed conformer, the condensed 2*H*-3,4-dihydropyran ring had half-chair conformation with *P*-helicity as defined by the positive value (+49.0°) of the torsional angle ω_C-6a, O-6, C-5,C-4a_ (Figure 8a,c). The CAM-B3LYP/TZVP PCM/CHCl_3_ ECD spectrum of (4a*R*,5*R*,10b*R*)-**23a** showed negative CE for the long-wavelength ^1^*L*_b_ band, which reproduced well the negative CEs of the second-eluting enantiomer of **23a** at 283 and 276 nm (Figure 8b). The flavan helicity rule was found valid for **23a** as well, since *P*-helicity of the heteroring resulted in negative ^1^*L*_b_ band CE. The (4a*R*,5*R*,10b*R*)-**23a** was the second-eluting enantiomer on the Chiralpak IA column, while the C-4a epimeric (4a*S*,5*R*,10b*R*)-**20a** was found the first-eluting one under the same conditions. The sign of the ^1^*L*_b_ band CE was used to determine the AC for the separated enantiomers of *rac*-**23b**-**g**. Similarly to **23a**, the first-eluting enantiomer had (4a*S*, 5*S*, 10b*S*) AC for **23d**, containing a C-5 3,5-dimethoxyphenyl substituent, while (4a*R*,5*R*,10b*R*) AC was determined for the first-eluting enantiomers of **23b**,**c** and **23e**,**g** (Appendix A).

## 4. Conclusions

The Neber rearrangement of seven oxime tosylates of flavanone analogues, containing a C-2 aryl substituent with different substitution pattern or a 1- or 2-naphthyl group, resulted in *trans*-3-aminoflavanones as the major product and the *cis* diastereomer as the minor one. The *cis* diastereomers could be obtained by simple filtration from the reaction mixture, while the *trans* isomers were isolated in pure form by trituration with acetone. The formation of 2-styrylbenzoxazol side-products was also observed, which were produced by ring-opening of the γ-pyrone ring and intramolecular cyclization of the Beckmann rearrangement intermediate. The *cis*- and *trans*-2-aminoflavanones were utilized for cyclization reactions to condense the 2-aryl-chroman or -2*H*-chromene subunit with morpholine, thiazole, or pyrrole moieties at the C-3-C-4 bond. Three diastereomers of morpholine-condensed 2-aryl-chromans, containing three chirality centers, were prepared through the *N*-chloroacetyl derivatives in four steps. Seven thiazole-condensed derivatives with different C-2 substituents were produced by the cyclization of the *N*-acetyl derivatives with Lawesson’s reagent and seven pyrrole-condensed one in the Knorr cyclization. Antiproliferative activities of condensed heterocycles and precursors were evaluated against A2780 and WM35 cancer cell lines at 50 μM concentration and IC_50_ values were determined for the best ones with MTT assay. One of the 3-(*N*-chloroacetylamino)-flavan-4-ol derivatives, containing a C-2 2-naphthyl substituent and showing analogy with acid ceramidase inhibitors, had 0.15 μM IC_50_ value against the A2780 cell line. This decrease in the viability is associated with an increase in apoptotic markers and decreased proliferation (DNA synthesis) but not with cellular necrosis. The IC_50_ values against HaCat and WM35 cell lines were found to be 6.06 and 3.50 μM, respectively, which implies 50- and 20-fold selectivities compared to that against A2780. From the condensed heterocycles, the thiazole derivative containing a 3,4,5-trimethoxy substituent had the best activity with 2.72 and 2.14 μM IC_50_ values against the A2780 and WM35 cell lines. Four pyrrole-condensed derivatives, which may be viewed as simplified analogues of natural lamellarins, had IC_50_ values below 10 μM down to 2.95 μM. Enantiomers of the condensed heterocycles were separated by chiral HPLC, HPLC-ECD spectra were recorded and TDDFT-ECD calculations were carried out to determine the AC of the enantiomers. The configurational assignment may aid future stereoselective synthesis and exploration of stereochemistry–activity relationships.

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
