# Peer review of "Synthesis and HPLC-ECD Study of Cytostatic Condensed O,N-Heterocycles Obtained from 3-Aminoflavanones"

_biomolecules, 2020, doi:10.3390/biom10101462_

Round 1
Reviewer 1 Report
This manuscript represents a very nice piece of work, carefully designed and elegantly executed. I recommend its publication as it stands.
The only minor remark is to lane 168. The sentence is not clear to me. Usually, we prepare anhydrous solution of alkali metal alkoxides applying metals (like sodium) because commercial alkoxides are hygroscopic. If you indeed used NaOEt, the sentence is OK. If, sodium, please indicate the amount.
Reviewer 2 Report
This manuscript described studies on synthesis, purification, and bioassay of condensed O,N-heterocycles. The authors emphasized synthesis and evaluation of antiproliferative compounds having heterocyclic structure. They also emphasized separation of enantiomers by chiral HPLC and determination of stereochemistries by ECD spectra.
I feel this manuscript focused on separation and confirmation of enantiomers by chiral HPLC and HPLC-ECD study rather than synthesis and biological evaluation of chemical compounds. Studies on stereochemical analysis of separated enantiomers are delineated. However, it seems hard to find improvement in terms of synthetic method or biological meaningfulness. Synthetic method shows low selectivity and explanation on antiproliferative activities against cell lines is relatively weak.
Considering scope of Biomolecules, which focused on natural and bio-inspired molecules, this manuscript seems suitable for publication in Molecules after an appropriate revision.
My comments for revision are as follow;
- Comment for Title
‘Cytostatic Condensed O,N-Heterocyles’ is broad and ambiguous term to describe compounds shown in manuscript. More specific key word will be helpful to understand this work.
- Comment for Introduction
It will be helpful to add figures showing summary of previous studies and improvement or difference of this work compared with reported work. (Page 1. Line 44 ~ Page 2 Line 54)
- Comment for Materials and Methods
Jasco (Line 85), JASCO (Line 88), and JASCO (Line 94) need to be consistent.
- Comment for Scheme 2.
It is hard to understand whether naphthyls of f and g are substituents at R2 or entire aryl groups are naphthyl group in Table in Scheme 2 and Table 1
- Comment for Scheme 3.
It would be helpful to explain the mechanism of epimerization with drawing of specific intermediate.
Reviewer 3 Report
The submitted manuscript "biomolecules-924992" presents a very rich research material, in which the attention is mainly focused on the chemical part and the structural analysis of the obtained library of new compounds.The synthesis and structure of the new derivatives was very well and meticulously prepared.
Unfortunately, the biological part is very poor. In vitro cytostatic studies have been performed only on the two cancer cell lines W35 and A2780.
It is necessary to report the activity against normal cell lines in order to be able to determine the cytotoxicity of the tested compounds.
I consider it necessary to investigate the mechanism of the apoptotic pathway induced by the most active compound rac-19g.
The biological research needs to be developed and improved. This is a necessary factor for the manuscript to be published in the journal Biomolecules.
Reviewer 4 Report
This manuscript reports an impressive amount of excellent work, with a perfect documentation in all its sections, synthesis, biological activity and computational chemistry.
I warmly congratulate the authors for the great work done. Despite the fact that the reported syntheses are leading to the racemic compounds, and the biological activities have been measured on the racemic mixtures, the paper is of very high interest. The full description and discussion of diastereoselectivity, as well as the assignment of stereochemistry by CD and computations will be fundamental in further developments of the reported synthesis towards asymmetry.
I have only a very minor comment: the way the cis and trans isomers, and side product has been isolated, reported on lines 1202-1208 is quite unclear. The cis isomer seems to precipitate from the first amount of acetone added. Then the trans is obtained after concentration of the acetone mother solution, and finally the side product remains in the acetone after the precipitation of the trans product. Is this correct? As this separation step is clearly critical, some more details could be given in the results section, as the volume of acetone used, before and after the concentration.
Round 2
Reviewer 2 Report
The manuscript is revised properly to be suitable for publication in Biomolecules.
Reviewer 3 Report
After applying all corrections, I recommend the presented material to be published in the Biomolecules.